

# Quantifying the global atmospheric power budget

Anastassia M. Makarieva[1,2], Victor G. Gorshkov[1,2], Andrei V. Nefiodov[1], Douglas Sheil[3], Antonio Donato Nobre[4], and Bai-Lian Li[2]

[1]Theoretical Physics Division, Petersburg Nuclear Physics Institute, 188300 Gatchina, St. Petersburg, Russia
[2]USDA-China MOST Joint Research Center for AgroEcology and Sustainability, University of California, Riverside 92521-0124, USA
[3]Faculty of Environmental Sciences and Natural Resource Management, Norwegian University of Life Sciences, Ås, Norway
[4]Centro de Ciência do Sistema Terrestre INPE, São José dos Campos SP 12227-010, Brazil

*Correspondence to:* Anastassia Makarieva (ammakarieva@gmail.com), Victor Gorshkov (vigorshk@thd.pnpi.spb.ru), Andrei Nefiodov (anef@thd.pnpi.spb.ru), Douglas Sheil (douglas.sheil@nmbu.no), Antonio Donato Nobre (anobre27@gmail.com), Bai-Lian Li (bai-lian.li@ucr.edu)

**Abstract.** The power of atmospheric circulation is a key measure of the Earth's climate system. The mismatch between predictions and observations under a warming climate calls for a reassessment of how atmospheric power $W$ is defined, estimated and constrained. Here we review published formulations for $W$ and show how they differ when applied to a moist atmosphere. Three factors, a non-zero source/sink in the continuity equation, the difference between velocities of gaseous air and conden-
sate, and interaction between the gas and condensate modifying the equations of motion, affect the formulation of $W$. Starting from the thermodynamic definition of mechanical work, we derive an expression for $W$ from an explicit consideration of the equations of motion and continuity. Our analyses clarify how some past formulations are incomplete or invalid. Three caveats are identified. First, $W$ critically depends on the boundary condition for gaseous air velocity at the Earth's surface. Second, confusion between gaseous air velocity and mean velocity of air and condensate in the expression for $W$ results in gross errors
despite the observed magnitudes of these velocities are very close. Third, $W$ expressed in terms of measurable atmospheric parameters, air pressure and velocity, is scale-specific; this must be taken into account when adding contributions to $W$ from different processes. We further present a formulation of the atmospheric power budget, which distinguishes three components of $W$: the kinetic power associated with horizontal pressure gradients ($W_K$), the gravitational power of precipitation ($W_P$) and the condensate loading ($W_c$). This formulation is valid with an accuracy of the squared ratio of the vertical to horizontal
air velocities. Unlike previous approaches, it allows evaluation of $W_P + W_c$ without knowledge of atmospheric moisture or precipitation. This formulation also highlights that $W_P$ and $W_c$ are the least certain terms in the power budget as they depend on vertical velocity; $W_K$ depending on horizontal velocity is more robust. We use MERRA and NCAR/NCEP re-analyses to evaluate the atmospheric power budget at different scales. Estimates of $W_K$ are found to be consistent across the re-analyses, while estimates for $W$ and $W_P$ drastically differ. We then estimate independent precipitation-based values of $W_P$ and discuss
how such estimates could reduce uncertainties. Our analyses indicate that $W_K$ increases with temporal resolution approaching our theoretical estimate for condensation-induced circulation when all convective motion is resolved. Implications of these findings for constraining global atmospheric power are discussed.





# 1 Introduction

Energy from the sun maintains atmospheric circulation which redistributes energy from warmer to colder regions and deter-
mines many aspects of global and local climate, including the terrestrial water cycle. How much power does our atmosphere's
circulation generate and why? These questions have long challenged theorists (Lorenz, 1967) and have gained renewed signif-
icance given how our planet's climate is affected by changes in atmospheric circulation (e.g., Bates, 2012; Shepherd, 2014).

A particular problem with understanding global circulation is the various mismatches that have arisen between model pre-
dictions and observed trends (e.g., Kociuba and Power, 2015). While models suggest general circulation should slow as global
temperatures rise, independent observations indicate circulation is intensifying (e.g., de Boisséson et al., 2014). Global wind
power appears to be rising as well (Huang and McElroy, 2015). On the other hand, global circulation models tend to overesti-
mate wind power (Boer and Lambert, 2008). For example, using the CAM3.5 model Marvel et al. (2013) estimated the global
kinetic power of the atmosphere at $3.4 \, \text{W m}^{-2}$, while published observational estimates range from 2 to $2.5 \, \text{W m}^{-2}$ (Kim and
Kim, 2013; Schubert and Mitchell, 2013; Huang and McElroy, 2015).

What then is meant by *atmospheric power* and how can it be estimated from observations? One approach is to consider
the rate at which the kinetic energy of winds dissipates to heat. Then *atmospheric power* can be defined as the *rate at which
new kinetic energy must be produced to offset the dissipative effects of friction* (Lorenz, 1967, p. 97). Following Lorenz (1967,
Eq. 102), atmospheric power ($\text{W m}^{-2}$) in a steady state should be defined as

$$W_I \equiv -\frac{1}{\mathcal{S}} \int_{\mathcal{M}} \mathbf{v} \cdot \nabla p \alpha d\mathcal{M} = -\frac{1}{\mathcal{S}} \int_{\mathcal{V}} \mathbf{v} \cdot \nabla p d\mathcal{V}, \tag{1}$$

Here $p$ is air pressure, $\mathbf{v}$ is air velocity, $\alpha \equiv 1/\rho$ is the specific air volume, $\rho$ is air density, $\mathcal{S}$, $\mathcal{M}$ and $\mathcal{V}$ are, respectively, Earth's
surface area, total mass and total volume of the atmosphere, $d\mathcal{M} = \rho d\mathcal{V}$. Sometimes atmospheric power is also referred to as
*global wind power* (e.g., Marvel et al., 2013) or *kinetic energy dissipation* (Boville and Bretherton, 2003). Laliberté et al.
(2015) termed $W_I$ atmospheric *work output*, while Robertson et al. (2011) and Pauluis (2015) referred to $W_I$ as, respectively,
the *generation of kinetic energy* and *kinetic energy production by atmospheric motions*.

Lorenz (1967, Eq. 102) proposed an additional formulation for kinetic energy production,

$$W_{II} \equiv -\frac{1}{\mathcal{S}} \int_{\mathcal{V}} \mathbf{u} \cdot \nabla p d\mathcal{V}, \tag{2}$$

where $\mathbf{u}$ is horizontal velocity of air. For a recent application see, e.g., Huang and McElroy (2015). According to Lorenz
(1967), $W_I = W_{II}$.

An alternative approach is to formulate atmospheric power from a thermodynamic viewpoint as *mechanical work* per unit
time performed by the air parcels as they change their volume. Accordingly, in the first law of thermodynamics as it is used
in the atmospheric sciences mechanical work per unit air volume per unit time is formulated as $p\nabla \cdot \mathbf{v}$ (e.g., Fiedler, 2000;
Ooyama, 2001; Pauluis and Held, 2002). With this reasoning global atmospheric power is

$$W_{III} \equiv \frac{1}{\mathcal{S}} \int_{\mathcal{V}} p\nabla \cdot \mathbf{v} d\mathcal{V}. \tag{3}$$





Meanwhile, according to Vallis (2006, Eq. 1.65), work done per unit mass is $pd\alpha$. Then total work performed by the atmosphere per unit time is[1]

$$W_{IV} \equiv \frac{1}{S} \int_{\mathcal{M}} p \frac{d\alpha}{dt} d\mathcal{M}. \tag{4}$$

Here

$$\frac{dX}{dt} \equiv \frac{\partial X}{\partial t} + (\mathbf{v} \cdot \nabla)X \tag{5}$$

is the material derivative of $X$.

As we discuss below, for a dry hydrostatic atmosphere obeying the continuity equation

$$\frac{\partial \rho}{\partial t} + \nabla \cdot (\rho \mathbf{v}) = \dot{\rho} \tag{6}$$

with $\dot{\rho} = 0$, all four definitions of atmospheric power are equal, $W_I = W_{II} = W_{III} = W_{IV}$. In an atmosphere with a water cycle, the source term $\dot{\rho}$ is not zero. Here gas (water vapor) is created by evaporation and destroyed by condensation with a local rate $\dot{\rho} \neq 0$ (kg m$^{-3}$ s$^{-1}$). As we will show, in this case each of the four candidate expressions $W_I$, $W_{II}$, $W_{III}$ and $W_{IV}$ are distinct.

In a moist atmosphere the velocity notation in Eqs. (1)-(4) becomes ambiguous: is it the velocity of gaseous air alone or the mean velocity of gaseous air and condensate particles? Furthermore, in the presence of phase transitions, the atmospheric circulation, as it performs mechanical work, does two things: it not only generates kinetic energy of macroscopic air motions, but it also lifts water generating the gravitational power of precipitation (Gorshkov, 1995; Pauluis et al., 2000). In a dry atmosphere the second component of mechanical work is absent. Thus, in a moist atmosphere each of the four definitions (1)-(4) can either refer to the generation of kinetic energy alone, to the mechanical work as a whole or to none of them. Which definition, if any, represents the "true power" of a moist atmosphere and is consistent with the thermodynamic interpretation of work?

In Section 2 we explore how the derivation of an expression for global power of atmospheric circulation is affected by phase transitions. In Section 3 we discuss how global atmospheric power can be represented as a sum of three distinct physical components. Two components dominate in the atmosphere of Earth: the kinetic power of the wind generated by horizontal pressure gradients and the gravitational power of precipitation generated by the ascending air. We compare our results with the previous formulations of the atmospheric power budget by Pauluis et al. (2000). In Section 4 we illustrate how the relationships derived in Section 3 require a revision of the recent estimates of atmospheric power by Laliberté et al. (2015). In Section 5 we illustrate our formulations by evaluating the atmospheric power budget from the MERRA (Rienecker et al., 2011) and NCAR/NCEP (Kalnay et al., 1996) re-analyses at different temporal resolutions. In Section 6 we discuss constraints on atmospheric power. There are two problems: to explain why wind power on Earth differs from zero (minimal threshold) and to determine what constrains this power (maximum threshold). We discuss the opportunities provided by consideration of the dynamic effects of condensation in combination with a conventional thermodynamic approach. Section 7 provides a list of the obtained results.

---

[1]Definition (4) for atmospheric power was endorsed by two referees of this work, see doi 10.5194/acp-2016-203-RC2 and 10.5194/acp-2016-203-RC4.





## 2 Atmospheric power in the presence of phase transitions

When going from a dry to a moist atmosphere, where, besides dry air, there is also water vapor and the non-gaseous water (condensate) present, we need to accurately define velocity and density (Pelkowski and Frisius, 2011). We consider an atmosphere of total volume $\mathcal{V}$ as composed of $n$ macroscopic air parcels each of volume $\tilde{V}_i$ (m³) such that $\mathcal{V} \equiv \sum_{i=1}^{n} \tilde{V}_i = \int_{\mathcal{V}} d\mathcal{V}$. Here $\mathcal{V}$ can be defined as the volume bounded by the Earth's surface and the surface corresponding to some fixed pressure level $p_t$ at the top of the atmosphere, e.g. to $p_t = 0.1$ hPa. This is the uppermost level in many atmospheric datasets including those in the MERRA re-analysis. With $\tilde{m}_d$, $\tilde{m}_v$ and $\tilde{m}_c$ being mass of, respectively, dry air, water vapor and condensate in a considered parcel, we define $\rho \equiv \tilde{m}/\tilde{V}$ to be the air density, $\tilde{m} \equiv \tilde{m}_d + \tilde{m}_v$, $\rho_d \equiv \tilde{m}_d/\tilde{V}$, $\rho_v \equiv \tilde{m}_v/\tilde{V}$, and $\rho_c \equiv \tilde{m}_c/\tilde{V}$ to be the condensate density.

We consider work performed by the atmospheric gases. We assume the thermodynamic notion that work is the product of pressure and volume change, such that work $W_a$ performed by an air parcel per unit time per unit volume is

$$W_a \equiv \frac{p}{\tilde{V}} \frac{d\tilde{V}}{dt}. \tag{7}$$

Considering that the parcel volume $\tilde{V}$ changes as a result of movement of each element $\mathbf{n}d\tilde{S}$ of the bounding material surface with velocity $\mathbf{v}$ (Batchelor, 2000, p. 74), we have

$$W_a = \frac{p}{\tilde{V}} \int_{\tilde{S}} \mathbf{v} \cdot \mathbf{n} d\tilde{S} = \frac{p}{\tilde{V}} \int_{\tilde{V}} \nabla \cdot \mathbf{v} d\tilde{V} = p\nabla \cdot \mathbf{v}, \tag{8}$$

where $\mathbf{n}$ is the outward normal unit vector. The latter equality is valid in the limit of sufficiently small $\tilde{V}$. Then, global atmospheric power per unit surface area can be defined and evaluated from the observed pressure and velocity of air as

$$W \equiv \frac{1}{\mathcal{S}} \sum_{i=1}^{n} W_{ai}\tilde{V}_i = \frac{1}{\mathcal{S}} \int_{\mathcal{V}} W_a d\mathcal{V} = \frac{1}{\mathcal{S}} \int_{\mathcal{V}} p\nabla \cdot \mathbf{v} d\mathcal{V}. \tag{9}$$

Equation (9) is equivalent to $W = W_{III}$, see Eq. (3). Its derivation requires several caveats. First, Eq. (9) assumes, as does the thermodynamic definition of work (7), that pressure is uniform within each parcel but varies among parcels. Based on this assumption, $W$ (9) represents a *definition* of total macroscopic mechanical work per unit time (global atmospheric power) that is consistent with the thermodynamic definition of work (7). As such, $W$ (9) is a function of the temporal and spatial scale at which the macroscopic velocity $\mathbf{v}$ is determined. This statement equally applies to dry as well as to moist atmospheres.

(Considering that pressure, too, varies across the parcel as velocity does, one could define parcel's work in Eq. (8) as $(1/\tilde{V})\int_{\tilde{S}} p\mathbf{v} \cdot \mathbf{n}d\tilde{S}$, i.e. placing pressure $p$ under the integral. For such an approach, see, for example, Fig. 2.7 of Holton (2004). However, in this case total atmospheric power $W$ would invariably be zero, which does not make sense. Indeed, $\int_{\mathcal{V}} \nabla \cdot (p\mathbf{v})d\mathcal{V} = 0$, see Eqs. (12)-(14) below.)

Second, since pressure $p$ in Eq. (8) is the total pressure of all gases in the parcel, velocity $\mathbf{v}$, the divergence of which governs how the parcel's volume changes, is assumed in Eq. (8) to be equal for all gases (dry air and water vapor). This statement also applies equally to dry and moist atmospheres.





Third, the derivation of Eq. (9) considers the work of the expanding *gas*. Hence, $\mathbf{v}$ in (9) is the velocity of the gaseous constituents of the atmosphere alone and does not include condensate.

Forth, the derivation of (9) is invariant with respect to phase transitions that change the amount of gas. That is, when deriving Eq. (9) no use was made of the equality

$$W_a = \frac{p}{\tilde{V}} \left( \tilde{m} \frac{d\alpha}{dt} + \alpha \frac{d\tilde{m}}{dt} \right), \tag{10}$$

where $\alpha \equiv 1/\rho = \tilde{V}/\tilde{m}$ is the volume occupied by unit air mass. The continuity equation or the equation of state were not used either. This is because Eq. (8) assumes that the volume of the air parcel can change *only* when $\nabla \cdot \mathbf{v} \neq 0$, i.e. when the parcel boundaries move at different velocities *at the considered scale*. Indeed, the standard thermodynamic interpretation of Eq. (9) is that if a certain parcel expands (positive work), the rest of the atmosphere contracts by the same amount (negative work). Thus, the expanding air parcels perform work on the compressing air parcels. When expansion and compression occur at different

pressures, the resulting difference can be converted to mechanical work.

The situation is different in the presence of phase transitions. Consider an atmospheric parcel in a still atmosphere composed of pure water vapor. Let it condense into a droplet. Now the parcel's reduction in volume $d\tilde{V}/dt < 0$ is due to the work of the intermolecular forces driving condensation. It is *not* due to some other air parcel expanding. Furthermore, condensation occurs rapidly governed by molecular velocities. Therefore, the condensation-induced volume changes are generally *not described* by

the velocity divergence $\nabla \cdot \mathbf{v}$, since the latter is defined at an arbitrary macroscopic scale. The question therefore arises whether the above derivation of $W = W_{III}$ (9) can be reconciled with Eq. (10) for $W_a$ in the presence of phase transitions.

We show in Appendix A that if we use the continuity equation and the ideal gas equation of state, the integration of $W_a$ (7) yields Eq. (9). This is because the requirement of continuity postulates that any void space produced by condensation must be filled by the expanding adjacent air parcels. For ideal gas, this additional positive work of rapid non-equilibrium expansion of

air parcels (which is absent in a dry atmosphere) cancels the negative work of the intermolecular forces that are responsible for the condensation-induced volume reduction. As discussed in Appendix A, this cancellation is a consequence of the ideal gas equation of state. As a result, the expression for global atmospheric power does not explicitly depend on condensation rate.

However, it is during such condensation-induced rapid expansion of the neighboring air parcels that the macroscopic pressure gradients can form to drive atmospheric circulation and *determine* the magnitude of atmospheric power $W$ (9). The conven-

tional view is that the circulation arises when some air parcels receive more heat than others and thus begin to expand. The cause of condensation-driven circulation is different. Here air parcels expand after condensation has reduced the concentration of gas in the adjacent space. Notably, Eq. (9) does not carry information about the causes of circulation. It defines macroscopic mechanical work per unit time (power) in a form compatible with the thermodynamic definition (7).

Fifth, Eq. (10) makes it clear that in the presence of phase transitions work done per unit mass is not equal to $p d\alpha/dt$ (cf.

Vallis, 2006, Eq. 1.65) but to

$$W_a \frac{\tilde{V}}{\tilde{m}} = W_a \alpha = p \left( \frac{d\alpha}{dt} + \frac{\alpha}{\tilde{m}} \frac{d\tilde{m}}{dt} \right) = p \left( \frac{d\alpha}{dt} + \alpha^2 \dot{\rho} \right), \tag{11}$$



where $\dot{\rho} \equiv (d\tilde{m}/dt)/\tilde{V}$ (kg m$^{-3}$ s$^{-1}$) is the source term from the continuity equation (6). It describes the local rate of phase transitions. The global integral of this additional term is not zero, $\int_{\mathcal{M}} p\alpha^2 \dot{\rho} d\mathcal{M} = \int_{\mathcal{V}} p\alpha \dot{\rho} d\mathcal{V} \neq 0$. Therefore, expression $W_{IV}$ (4) that neglects this term is incorrect, $W_{IV} \neq W_{III} = W$. It cannot be used for evaluation of atmospheric power when the atmosphere has a water cycle.

Finally, we note that Eq. (9) does not assume stationarity. Nor does it assume hydrostatic equilibrium.

In the next section we consider how $W$ can be decomposed into several terms with different physical meaning. This will clarify how $W = W_{III}$ relates to $W_I$ (1) and $W_{II}$ (2).

## 3 Revisiting current understanding of the atmospheric power budget

### 3.1 The boundary condition for vertical air velocity at the Earth's surface

Noting that $p\nabla \cdot \mathbf{v} = \nabla \cdot (p\mathbf{v}) - \mathbf{v} \cdot \nabla p$ and using the divergence theorem (Gauss-Ostrogradsky theorem) we can see that $W = W_{III}$ (9) coincides with $W_I$ (1),

$$W = W_{III} \equiv \frac{1}{\mathcal{S}} \int_{\mathcal{V}} p\nabla \cdot \mathbf{v} d\mathcal{V} = -\frac{1}{\mathcal{S}} \int_{\mathcal{V}} \mathbf{v} \cdot \nabla p d\mathcal{V} + I_t + I_s = W_I + I_t + I_s, \tag{12}$$

if the following integrals are zero:

$$I_t \equiv \frac{p_t}{\mathcal{S}} \int_{z=z(p_t)} \mathbf{v} \cdot \mathbf{n} d\mathcal{S} = 0, \tag{13}$$

$$I_s \equiv \frac{1}{\mathcal{S}} \int_{\mathcal{S}} p_s \mathbf{v} \cdot \mathbf{n} d\mathcal{S} = 0. \tag{14}$$

Integral (13) is taken over the upper boundary $z = z(p_t)$, where $z(p_t)$ is the altitude of the pressure level $p = p_t$ defining the top of the atmosphere. At the upper boundary $\mathbf{v}$ is zero in the steady state only. Generally for $z = z(p_T)$ we have $\mathbf{v} \cdot \mathbf{n} \neq 0$ (a similar condition of non-zero velocity at the upper boundary (the oceanic surface) is commonly used in oceanic science, e.g. Tailleux (2015)). However, since the distribution of pressure versus altitude is exponential and $I_t$ is proportional to $p_t$, by choosing a sufficiently small $p_t$ it is possible to ensure that $I_t$ (13) is arbitrarily small compared to $W$. For $p_t = 0.1$ hPa we estimate $I_t \sim 10^{-4} W$ (see Fig. 13d in Appendix D). So it is safe to assume that $I_t = 0$.

Integral (14) is taken over the Earth's surface ($p_s$ is surface pressure). It is zero when

$$\mathbf{v} \cdot \mathbf{n}|_{z=0} \equiv w_s = 0, \tag{15}$$

where $w_s$ is the surface value of the vertical velocity of air $\mathbf{w}$. In a dry atmosphere Eq. (15) always holds. As we discuss below, for a moist atmosphere Eq. (15) also holds, such that $W = W_{III} = W_I$.

In a dry atmosphere the ideal gas molecules collide elastically with the Earth's surface. At $z = 0$ there are as many molecules going upwards as there are going downwards. When water evaporates from the Earth's surface, there are more water vapor





molecules going upwards than downwards. The mean vertical velocity of the water vapor molecules at $z = 0$ is positive. It differs from the mean vertical velocity of dry air, which remains zero. The formulation for $W$ (9), which assumes equal velocity for all gases, is therefore not applicable at $z = 0$.

However, as the colliding molecules exchange momentum, already at a distance of the order of a few free path lengths $l_f \sim 10^{-7}$ m from the surface, all air molecules have one and the same mean velocity relative to the Earth's surface. The

vertical component of this velocity averaged over any macroscopic horizontal scale $l \gg l_f$ must be zero. This is because there is no source of dry air at $z = 0$. Indeed, suppose that at $z \sim l_f$ vertical velocity $w_s$ is positive over an area of the order of $l^2$. There is an upward flux of dry air from this area equal to $\rho_d w_s l^2$ (kg s$^{-1}$). In the absence of a source of dry air at $z < l_f$ mass conservation requires a horizontal inflow of dry air to the considered area of the order $\rho_d u_s l_f l$, where $u_s$ is the mean horizontal velocity at $z \leq l_f$. Equating the horizontal and vertical fluxes of dry air we find $w_s \sim u_s(l_f/l)$. Since under no-slip condition

the horizontal velocity at the surface is zero, it follows that $w_s = 0$. (Even if we take $u_s \sim 1$ m s$^{-1}$, for a horizontal scale of $l \sim 1$ km we find $w_s \sim 10^{-10} u_s$ and $p_s w_s \sim 10^{-5}$ W m$^{-2}$, which is less than $\sim 10^{-5}$ of the global atmospheric power $W$.)

We emphasize that the boundary condition (15) is vital for the equality between $W = W_{III}$ (9), (3) derived from the thermodynamic definition of work and $W_I$ (1) of Lorenz (1967). Moreover, using $w_s \neq 0$ when analysing atmospheric power budget yields significant errors. For example, if one defined $w_s > 0$ for $z = 0$ from the upward flux of water vapor as $\rho_{vs} w_s = E$,

where $\rho_{vs}$ is water vapor density at the surface and $E$ is evaporation (see, e.g., Eq. 3 of Pauluis et al., 2000, to be discussed in Section 3.5), one would obtain an estimate for atmospheric power exceeding the incoming flux of solar radiation. Indeed, with $E \sim 10^3$ kg m$^{-2}$ yr$^{-1}$ and $\rho_{vs} \sim 10^{-2}$ kg m$^{-3}$ we would have $w_s \sim 3 \times 10^{-3}$ m s$^{-1}$ and $I_s = p_s w_s = 3 \times 10^2$ W m$^{-2}$. Then from Eq. (12) we would obtain $W = W_I + I_s > I_s \sim 300$ W m$^{-2}$.

The reason for this contradiction is that the notions of atmospheric work and power are scale-specific: they are defined for a

given macroscopic scale (see Section 2). Meanwhile the non-zero mean vertical velocity of evaporating water molecules at $z = 0$ exists on the molecular scale only, where no macroscopic mechanical work is done. Mixing up molecular and macroscopic scales yields unphysical results. Thus, in any evaluation of mechanical work output of the atmospheric air parcels one should put $w_s = 0$. Equation (15) is not in contradiction with the existence of an inflow of water vapor into the atmosphere at $z = 0$. It just means that water vapor – treated as an ideal gas – should be considered as arising by evaporation within the surface air

parcels, the latter having zero vertical velocity at their lower boundary. Mathematically this is achieved by introducing a source of water vapor at $z = l_f \approx 0$ in the form of Dirac's delta function (see Eq. (59) below).

For $z \lesssim l_f$ mean velocities of water vapor and dry air do not coincide, such that Eq. (7) is not directly applicable in this region. It is interesting to see how this equation could be modified to make sense within this narrow layer. A certain share of water vapor molecules with density $\rho_E$ possess mean vertical velocity $w_E > 0$ at $z = 0$. This velocity is related to evaporation

$E$ as $E = \rho_E w_E$. On the other hand, as we have established, at $z \sim l_f$ all air molecules have vertical velocity $w_s = 0$. Thus for the water vapor molecules with $w = w_E > 0$ we have $\int_0^{l_f} p_E \nabla w \, dz = -p_E w_E$. Here $p_E = (\rho_E/M_v)RT_s$ is the partial pressure of these molecules (treated as ideal gas), $M_v$ is molar mass of water vapor, $R = 8.3$ J mol$^{-1}$ K$^{-1}$ is the universal gas





constant. Using $w_E = E/\rho_E$ by analogy with Eq. (9) we have

$$W_s \equiv \int_0^{l_f} p_E \nabla w \, dz = -p_E w_E = -(E/M_v)RT_s. \tag{16}$$

Note that the resulting expression for power $W_s$ associated with surface evaporation does not depend on $w_E$, $\rho_E$ or $p_E$ – vertical velocity, density or partial pressure of the evaporating molecules. It depends solely on the evaporation rate $E/M_v$ expressed in moles of gas per unit time per unit area. In contrast to condensation, which frees space from water vapor thus allowing the neighboring air parcels to expand filling void space, evaporation adds water vapor molecules to the atmosphere thus compressing the water vapor that is already there. This compression explains why $W_s$ (16) is negative: the partial pressure of water vapor grows in the result of evaporation. The water vapor is worked upon.

Water vapor compressed at the surface expands as it moves towards the condensation area, where the intermolecular forces will compress it again into a liquid droplet. If condensation and evaporation are spatially separated on a macroscopic scale, this expansion will generate kinetic energy at this scale. Since potential energy associated with gas pressure can be fully converted to kinetic energy, one can expect that an atmospheric circulation driven by phase transitions of water vapor will have global power $W$ close to $|W_s|$. As we will discuss in Section 6 this happens indeed to be the case: with $M_v = 18$ g mol$^{-1}$, $E \approx 10^3$ kg m$^{-2}$ yr$^{-1}$ and $T_s \approx 300$ K from Eq. (16) we have $|W_s| = 4.4$ W m$^{-2}$, which is fairly close to the observed global atmospheric power $W$.

## 3.2 Kinetic power and the gravitational power of precipitation

We have established that in a moist atmosphere in the view of $w_s = 0$ total power equals $W = W_I$. This result does not assume stationarity. We will now analyze the steady-state atmospheric power budget by decomposing $W$ into distinct terms. We will use the following steady-state continuity equations for air and condensate particles (e.g., Ooyama, 2001):

$$\nabla \cdot (\rho \mathbf{v}) = \dot{\rho}, \tag{17}$$

$$\nabla \cdot (\rho_c \mathbf{v}_c) = -\dot{\rho}, \tag{18}$$

and the steady-state equation of air motion (cf. Lorenz, 1967, Eq. 1):

$$\rho \frac{d\mathbf{v}}{dt} = -\nabla p + \rho \mathbf{g} + \mathbf{F} + \mathbf{F}_c. \tag{19}$$

Here $\mathbf{v}_c$ is velocity of condensate particles, $\rho_c$ is their density (total mass per unit air volume), $\mathbf{v}$ and $\rho$ is gas velocity and density, $d\mathbf{v}/dt = (\mathbf{v} \cdot \nabla)\mathbf{v}$, $\mathbf{g}$ is acceleration of gravity, $\mathbf{F}$ is the turbulent friction force and $\mathbf{F}_c$ is the force exerted on the gas by condensate particles.

Taking the scalar product of Eq. (19) with air velocity $\mathbf{v}$, integrating the resulting equation over atmospheric volume $\mathcal{V}$ and dividing by Earth's surface area $\mathcal{S}$, we note that the first term in the right-hand side of Eq. (19) equals $W_I$ (1):

$$W = W_I = W_F - \frac{1}{\mathcal{S}} \int_{\mathcal{V}} \left( \rho \mathbf{g} \cdot \mathbf{w} - \rho \frac{dK}{dt} + \mathbf{F}_c \cdot \mathbf{v} \right) d\mathcal{V}, \quad W_F \equiv -\frac{1}{\mathcal{S}} \int_{\mathcal{V}} \mathbf{F} \cdot \mathbf{v} \, d\mathcal{V}. \tag{20}$$



Here $K \equiv v^2/2$ is kinetic energy of the gas per unit mass; $W_F$ represents work per unit time of the turbulent friction force. This term comprises dissipation of kinetic energy to heat, which is positive definite, and export of kinetic energy from the atmosphere via surface stress (see, e.g., Fiedler, 2000; Landau and Lifshitz, 1987, § 16).

To clarify the meaning of the second term in Eq. (20) we recall that

$$\mathbf{g} = -g\nabla z, \tag{21}$$

where $g \equiv |\mathbf{g}|$ and use the divergence theorem together with Eq. (17) to obtain[2]

$$W_P \equiv -\frac{1}{\mathcal{S}}\int_{\mathcal{V}} \rho\mathbf{w}\cdot\mathbf{g}\,d\mathcal{V} = \frac{1}{\mathcal{S}}\int_{\mathcal{V}} \rho g\mathbf{v}\cdot\nabla z\,d\mathcal{V} = \frac{1}{\mathcal{S}}\int_{\mathcal{S}} g\mathbf{n}\cdot(\mathbf{v}\rho z)d\mathcal{S} - \frac{1}{\mathcal{S}}\int_{\mathcal{V}} \dot{\rho}gz\,d\mathcal{V} = -\frac{1}{\mathcal{S}}\int_{\mathcal{V}} gz\dot{\rho}\,d\mathcal{V}. \tag{22}$$

The surface integral in (22) is taken at the Earth's surface (here it is zero because $z = 0$) and $z = z(p_t)$ (here it is also zero,

because $\rho\mathbf{n}\cdot\mathbf{v} = 0$). In the last integral in Eq. (22) $gz$ represents potential energy of a unit mass in the Earth's gravitational field. Thus, $W_P > 0$ represents the rate at which the potential energy of condensate particles is produced during condensation.

Defining precipitation path length $\mathcal{H}_P$ as

$$\mathcal{H}_P \equiv -\frac{1}{\mathcal{S}}\frac{\int_{z>0}\dot{\rho}z\,d\mathcal{V}}{P}, \quad P \equiv -\frac{1}{\mathcal{S}}\int_{z>0} \dot{\rho}\,d\mathcal{V}, \tag{23}$$

where $P \geq 0$ is precipitation at the ground $z = 0$, we find from Eq. (22) that

$$W_P = g\mathcal{H}_P P. \tag{24}$$

It is natural to call $W_P$ the "gravitational power of precipitation" (Gorshkov, 1995, p. 30).

For any quantity $X$ noting that $\rho(\mathbf{v}\cdot\nabla)X = (\nabla\cdot\mathbf{v})X\rho - X\nabla\cdot(\rho\mathbf{v})$ and using the definiton of material derivative (5) and the continuity equation (17) we can apply the divergence theorem with the boundary conditions (13), (14) to obtain in a steady state

$$\int_{\mathcal{V}} \rho\frac{dX}{dt}\,d\mathcal{V} = -\int_{\mathcal{V}} \dot{\rho}X\,d\mathcal{V}. \tag{25}$$

In the view of Eq. (25) the third term in Eq. (20) equals

$$\dot{K} \equiv \frac{1}{\mathcal{S}}\int_{\mathcal{V}} \rho\frac{dK}{dt}\,d\mathcal{V} = -\frac{1}{\mathcal{S}}\int_{\mathcal{V}} \dot{\rho}K\,d\mathcal{V}. \tag{26}$$

It describes how the kinetic energy $\rho K$ of air (gas) in a unit volume is affected by phase transitions that change the amount of gas.

In a steady state we have $\int_{\mathcal{V}}\dot{\rho}\,d\mathcal{V} = 0$. Since a significant part of evaporation $\dot{\rho} > 0$ is located at the surface $z = 0$, for $z > 0$ condensation $\dot{\rho} < 0$ dominates. Thus, $W_P$ (22) is always positive. The sign of $\dot{K}$ (26) depends on whether the kinetic

---

[2]Equation (22) can be also obtained using the definition (5) of material derivative, $\rho\mathbf{g}\cdot\mathbf{w} = \rho\mathbf{g}\cdot d\mathbf{z}/dt = \rho(\mathbf{v}\cdot\nabla)(\mathbf{g}\cdot\mathbf{z}) = -(\nabla\cdot\mathbf{v})\rho gz + gz\nabla\cdot(\rho\mathbf{v})$.





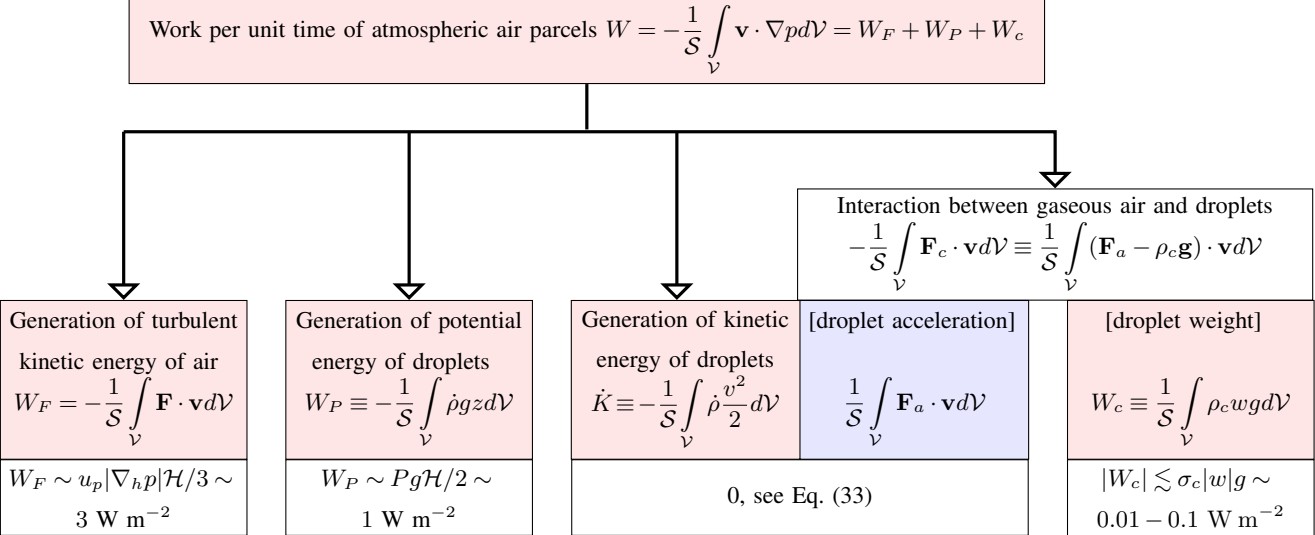

**Figure 1.** Atmospheric power budget, Eq. (37), with $\mathbf{F}_c$ given by Eq. (31). Red and blue rectangles denote positive and negative terms, respectively. Characteristic values of the budget terms are given for $w = 10^{-2}$ m s$^{-1}$, $u_p = 1$ m s$^{-1}$, $|\nabla_h p| = 1$ Pa km$^{-1}$, $P = \rho_v w/4 = 2.5 \times 10^{-5}$ kg m$^{-2}$ s$^{-1}$ (0.8 m yr$^{-1}$), $\rho_v = 10^{-2}$ kg m$^{-3}$, $\mathcal{H} = 10$ km, $\sigma_c = 0.1 - 1$ kg m$^{-2}$, see Section 3.4 for details. Note that $\dot{K} \sim P v^2/2 \sim 10^{-3}$ W m$^{-2}$ for $v = 10$ m s$^{-1}$.

energy $K$ of air is larger at the surface (where evaporation dominates and $\dot{\rho} > 0$) than at the mean condensation height (where condensation dominates and $\dot{\rho} < 0$). Under no-slip condition $K|_{z=0} \equiv K_s = 0$ and $\dot{K}$ is positive.

When the condensate particles and air do not interact ($\mathbf{F}_c = 0$), the atmospheric power budget (20) becomes

$$W_F = W - W_P - \dot{K}. \tag{27}$$

Expanding air parcels perform work $W$ per unit time. In the absence of phase transitions ($\dot{\rho} = 0$) this work goes to generate kinetic energy of air at the considered scale. This energy dissipates by turbulent friction, such that $W_F = W$. In the presence of phase transitions, $W$ is additionally spent to create kinetic energy $K$ and potential energy $gz$ of condensate particles at a rate $-\int_{\mathcal{V}} \dot{\rho}(K + gz)d\mathcal{V} > 0$ (Fig. 1). As indicated by Eq. (27), production of kinetic energy of air is then reduced by the corresponding amount $W_P + \dot{K}$. This gravitational and kinetic energy of condensate particles dissipates outside the atmosphere. A certain part of kinetic energy of air also dissipates outside the atmosphere, as it goes to generate the kinetic energy of waves and potential energy of oceanic stratification to drive oceanic circulation (Wang and Huang, 2004; Ferrari and Wunsch, 2009; Tailleux, 2010).

In the general case, condensate particles which, at the moment of their formation, have kinetic energy $K$ and potential energy $gz$, can use (some part of) this energy to interact with the air – either generating additional kinetic energy of air or impeding its generation at the considered spatial scale. This interaction introduces extra terms to the atmospheric power budget which we discuss below.





### 3.3 Interaction between air and condensate particles

For the brevity sake, we will refer to condensate particles as "droplets". The equation of motion for one droplet of mass $m$ is

$$m\mathbf{a} = m\mathbf{g} + \mathbf{f}_c, \tag{28}$$

where $\mathbf{a}$ is droplet acceleration, $\mathbf{f}_c$ is the force exerted by the air on the droplet and $-\mathbf{f}_c$ is the force exerted by the droplet on the air.

Total force $\mathbf{F}_c$ exerted on the air by all droplets contained in a unit air volume is

$$\mathbf{F}_c = -\frac{\sum_i \mathbf{f}_{ci}}{\tilde{V}} = \rho_c \mathbf{g} - \mathbf{F}_a, \quad \mathbf{F}_a \equiv \frac{1}{\tilde{V}} \sum_i m_i \mathbf{a}_i, \quad \rho_c \equiv \frac{\sum_i m_i}{\tilde{V}}. \tag{29}$$

Here the summation is over all droplets in the considered air parcel of volume $\tilde{V}$. If the droplets do not accelerate, then $\mathbf{F}_a = 0$ and the interaction between air and droplets in Eq. (19) is reduced to $\mathbf{F}_c = \rho_c \mathbf{g}$.

It is commonly assumed that horizontal velocity $\mathbf{u}_c$ of the condensate coincides with that of air, $\mathbf{u}_c = \mathbf{u}$, while vertical velocity $\mathbf{w}_c$ relates to the vertical velocity of air $\mathbf{w}$ as $\mathbf{w}_c = \mathbf{w} + \mathbf{w}_T$, where velocity $\mathbf{w}_T$ does not change along the droplet path: $\partial \mathbf{w}_T / \partial t + (\mathbf{v}_c \cdot \nabla)\mathbf{w}_T = 0$ (Ooyama, 2001; Satoh, 2003, 2014). In this case acceleration $\mathbf{a}$ of one droplet with velocity 
$\mathbf{v}_c = \mathbf{u}_c + \mathbf{w}_c = \mathbf{v} + \mathbf{w}_T$ is not zero:

$$\mathbf{a} \equiv \frac{d\mathbf{v}_c}{dt_c} \equiv \frac{\partial \mathbf{v}_c}{\partial t} + (\mathbf{v}_c \cdot \nabla)\mathbf{v}_c = \frac{\partial \mathbf{v}}{\partial t} + (\mathbf{v} \cdot \nabla)\mathbf{v} + (\mathbf{w}_T \cdot \nabla)\mathbf{v} = \frac{d\mathbf{v}}{dt} + w_T \frac{\partial \mathbf{v}}{\partial z}. \tag{30}$$

If all droplets in the considered volume have equal velocity[3], using Eqs. (29) and (30) we have

$$\mathbf{F}_a = \rho_c \frac{d\mathbf{v}}{dt} + \rho_c w_T \frac{\partial \mathbf{v}}{\partial z}, \quad \mathbf{F}_c = \rho_c \mathbf{g} - \mathbf{F}_a = \rho_c \mathbf{g} - \rho_c \frac{d\mathbf{v}}{dt} - \rho_c w_T \frac{\partial \mathbf{v}}{\partial z}. \tag{31}$$

This formulation of $\mathbf{F}_c$ was proposed by Ooyama (2001) using a different logic than we did in Eqs. (28)-(31), namely by
295 considering change of momentum $\rho \mathbf{v} + \rho_c \mathbf{v}_c$ (thus combining the equations of motion with the continuity equations for air and condensate). Equations of motions (19) with $\mathbf{F}_c$ given by Eq. (31)

$$(\rho + \rho_c)\frac{d\mathbf{v}}{dt} = -\nabla p + (\rho + \rho_c)\mathbf{g} + \mathbf{F} - \rho_c w_T \frac{\partial \mathbf{v}}{\partial z} \tag{32}$$

are used in general circulation models (see, e.g., Satoh, 2014, Chapter 26).

Using $\mathbf{v}_c = \mathbf{v} + \mathbf{w}_T$ and noting that, in the view of Eqs. (17), (18), (25) and the no-slip condition $K|_{z=0} = 0$, the following
integral is zero:

$$\int_{\mathcal{V}} \left( \rho \frac{dK}{dt} + \mathbf{F}_a \cdot \mathbf{v} \right) d\mathcal{V} = \int_{\mathcal{V}} \left( \rho \frac{dK}{dt} + \rho_c \frac{dK}{dt} + \rho_c w_T \frac{\partial K}{\partial z} \right) d\mathcal{V} = \int_{z=0} \rho_c w_T K d\mathcal{S} = 0, \tag{33}$$

putting $\mathbf{F}_c$ (31) in Eq. (20) yields

$$W_F = W - W_P - W_c, \quad W_c \equiv -\frac{1}{\mathcal{S}} \int_{\mathcal{V}} \rho_c \mathbf{g} \cdot \mathbf{w} d\mathcal{V}. \tag{34}$$

---

[3] For droplets with different velocities Eq. (31) does not hold (Makarieva et al., 2013a).





Comparing Eq. (34) with Eq. (27) we note that the negative term $-\dot{K}$ has disappeared from the power budget (Fig. 1). This

means that kinetic energy imparted by the expanding air parcels to condensate particles at a rate $\dot{K}$ has been converted back

to the kinetic energy of air as the accelerating droplets exerted drag on the air governed by their acceleration $\mathbf{F}_a \neq 0$. We also

note the appearance of the condensate loading term $W_c$, which we will discuss below.

     Before proceeding to quantitative estimates of the atmospheric power budget, we note one implicit inconsistency in Eq. (34).

As the droplets hit the ground, they possess kinetic energy $w_T^2/2$, which dissipates at a rate $-\rho_{cs}w_T^3/2$. Equation (34) does not

appear to account for this dissipative process. But, if not produced by work $W$ of the air parcels, where does this energy come

from? This inconsistency is inherited from the continuity equation (18) for the condensate. This equation assumes that a droplet

with velocity $\mathbf{v}_c = \mathbf{v} + \mathbf{w}_T$ different from that of water vapor appears exactly at the point where water vapor condenses. The

real process is different: immediately upon its formation the droplet has the same velocity as the water vapor molecules from

which it forms. Then the droplet is accelerated downward by gravity to acquire velocity $\mathbf{v} + \mathbf{w}_T$ *slightly below the point of*

*condensation* in a moving air parcel. This means that using Eq. (18) overestimates potential energy $gz$ of droplets with velocity

$\mathbf{v}_c$, and, hence, $W_P$ (22) in Eq. (34), by an amount equal to the difference between their kinetic energy and that of local air

at the point of condensation. Quantitatively this inconsistency is small: with $|w_T| \sim 5$ m s$^{-1}$, we have $-\rho_c w_T^3/2 = P w_T^2/2 \sim$

$4 \times 10^{-4}$ W m$^{-2}$, which is less than one tenth of per cent of the observed global atmospheric power.

### 3.4    Scale analysis of the atmospheric power budget

Equation (20) cannot be readily used to assess the global atmospheric power budget from observations, because no general

formulations exist for turbulent friction $\mathbf{F}$. A more convenient formulation for $W$ can be obtained considering the vertical

equation of motion (19). Multiplying Eq. (19), where $\mathbf{F}_c$ is given by Eq. (31), by vertical velocity $\mathbf{w}$ and taking into account

Eq. (33) we obtain:

$$\int_{\mathcal{V}} \left( \rho \frac{dK_w}{dt} + \rho_c \frac{dK_w}{dt} + \rho_c w_T \frac{\partial K_w}{\partial z} \right) d\mathcal{V} = \int_{\mathcal{V}} \left( -w\frac{\partial p}{\partial z} - \rho g w - \rho_c g w + F_w w \right) d\mathcal{V} = 0. \tag{35}$$

Here $K_w \equiv w^2/2$ is the vertical velocity contribution to the kinetic energy of air $K \equiv v^2/2 = (u^2 + w^2)/2$ and $F_w$ is the

vertical component of $\mathbf{F}$. By analogy with Eq. (33), the right-hand side of Eq. (35) equals zero since according to Eq. (14) we

have $K_w|_{z=0} = 0$.

     We will now estimate the relative magnitude of the third term, $F_w w$, in the right-hand part of Eq. (35). Consider a circulation

pattern of horizontal size $L$, vertical size $\mathcal{H} \sim 10$ km (height of the troposphere), horizontal velocity $u$ and vertical velocity

$w$, $w/u \sim H_w/L < \mathcal{H}/L$, where $H_w$ is the scale height for vertical velocity. If $H_w \sim 1$ km is the boundary layer, we have

$w/u = 0.1\mathcal{H}/L$ (see, e.g., Holton, 2004, p. 39). The horizontal $F_u$ and vertical $F_w$ components of $\mathbf{F}$ in Eq. (19) are of the

order of $F_u \sim \nu_e u/(H_w l_e)$ and $F_w \sim \nu_e w/(H_w l_e)$, respectively, where $\nu_e$ is eddy viscosity and $l_e$ is the eddy spatial scale

(e.g., Holton, 2004, Chapter 5). The leading term in $W_F$ is thus of the order of $uF_u \sim \nu_e u^2/(H_w l_e)$ (see also Landau and

Lifshitz, 1987, § 16), while $|F_w w|$ is proportional to $w^2 \ll u^2$. Neglecting $F_w w$ we can write Eq. (35) as

$$-\frac{1}{\mathcal{S}} \int_{\mathcal{V}} w\frac{\partial p}{\partial z} d\mathcal{V} = \frac{1}{\mathcal{S}} \int_{\mathcal{V}} (\rho g w + \rho_c g w) \, d\mathcal{V} \equiv W_P + W_c. \tag{36}$$





Using Eq. (36) and $W = W_I$, the steady-state atmospheric budget consistent with the continuity equations, (17) and (18), and the equations of motion, (19) and (31), can be formulated as follows:

$$W = -\frac{1}{\mathcal{S}} \int_{\mathcal{V}} \mathbf{v} \cdot \nabla p d\mathcal{V} \approx W_K + W_P + W_c, \qquad (37)$$

$$W_K \equiv -\frac{1}{\mathcal{S}} \int_{\mathcal{V}} \mathbf{u} \cdot \nabla p d\mathcal{V}, \qquad (38)$$

$$W_P \equiv \frac{1}{\mathcal{S}} \int_{\mathcal{V}} \rho g w d\mathcal{V} = -\frac{1}{\mathcal{S}} \int_{\mathcal{V}} g z \dot{\rho} d\mathcal{V} = P g \mathcal{H}_P, \qquad (39)$$

$$W_c \equiv \frac{1}{\mathcal{S}} \int_{\mathcal{V}} \rho_c g w d\mathcal{V}. \qquad (40)$$

The same result could be obtained assuming hydrostatic equilibrium and writing the vertical equation of motion in Eq. (19) as

$$\nabla_z p = (\rho + \rho_c)\mathbf{g}. \qquad (41)$$

However, while hydrostatic equilibrium is valid with an accuracy of $w/u$ (Wedi and Smolarkiewicz, 2009), approximation in Eq. (37) based on Eq. (36) is valid with an accuracy of $(w/u)^2$. For example, it will be valid with an accuracy of 1% for a circulation pattern with $L \sim \mathcal{H} \sim 10$ km, $H_w \sim 1$ km and $w/u \sim 10^{-1}$. Note that with the same accuracy $W_F = W_K$, cf. Eqs. (34) and (37).

Equations (37)-(40) clarify the relationship between the two formulations of atmospheric power $W_I$ (1) and $W_{II}$ (2): $W_{II} = W_K$ coincides with $W = W_I$ in the absence of phase transitions only, i.e. when $W_P = W_c = 0$. This resolves some confusion in the literature, whereby in some publications it is total atmospheric power $W = W_I$ that is referred to as *generation of kinetic energy* (e.g., Robertson et al., 2011, their Eq. 1), while in others the same term is applied to $W_K$, which is estimated from horizontal velocities (see, e.g., Boville and Bretherton, 2003; Huang and McElroy, 2015). At the same time, in such studies $W_K$ is confused with the total atmospheric power $W$: i.e. in the total power budget the gravitational power of precipitation, $W_P$, is overlooked (e.g., Huang and McElroy, 2015, their Fig. 10). We also note that the gravitational power of precipitation $W_P$ has not been explicitly identified in past studies assessing the Lorenz energy cycle (see, e.g., Kim and Kim, 2013, and references therein).

The gravitational power of precipitation $W_P$ (39) does not depend on air-condensate interactions. (For example, this term would be present in the atmospheric power budget even if the condensate disappeared immediately upon condensation or experienced free fall not interacting with the air at all.) This is because $W_P$ reflects the net work of expanding and contracting air parcels as they travel from the level where evaporation occurs (where water vapor arises) to the level where condensation occurs (where water vapor disappears). When condensation occurs above where evaporation occurs, the air parcels expand as they move upwards towards condensation, and the work is positive irrespective of what happens to the condensate.

Term $W_c$ (40) in Eq. (37) describes the impact of condensate loading. Since condensate makes the air heavier (Pelkowski and Frisius, 2011), it impedes acceleration of the ascending air but promotes acceleration of the descending air. Thus, if the condensate is predominantly located in areas with $w > 0$ (ascending air), $W_c > 0$ reduces kinetic power $W_K$ compared to



total power $W$. If the condensate is located where $w < 0$, $W_K$ increases compared to $W$ as far as $W_c < 0$: in this case part of potential energy of the condensate is returned to the air motions that have originally generated it. Since in the terrestrial atmosphere condensate particles are predominantly generated in the ascending air flows, $W_c$ should be positive.

Characteristic magnitudes of $W_K$, $W_P$ and $W_c$ for air motions corresponding to the horizontal scale of $L \sim 100$ km can
be estimated as follows (Fig. 1). Consider a circulation pattern with a typical horizontal pressure gradient of the order of $|\nabla_h p| \sim 1$ Pa km$^{-1}$ and horizontal velocity component parallel to the pressure gradient $u_p \sim 1$ m s$^{-1}$. (Note that $u_p$, which determines air motion across isobars towards the area of lower pressure, is usually smaller than the geostrophic or cyclostrophic velocity component that is parallel to isobars.) Using these values we obtain for kinetic energy generation in the boundary layer $H_w |\nabla_h p| u_p \sim 1$ W m$^{-2}$. Above the boundary layer significantly less kinetic power is generated per unit air volume (see
Section 5 below), such that the actual value of $W_K$ we estimate from MERRA turns out to be about three times less than $\mathcal{H} |\nabla_h p| u_p \sim 10$ W m$^{-2}$.

For $L \sim 100$ km from the continuity equation we have $w \sim u_p H_w / L \sim 10^{-2}$ m s$^{-1}$. Assuming that at $z = H_w$ water vapor density is $\rho_v \sim 10^{-2}$ kg m$^{-3}$ and that about half of all ascending water vapor condenses and precipitates and that the air ascends over one half of the planet area, we find that precipitation due to the considered air motions would be of the order of
380 $P \sim 0.25 \rho_v w \sim 2.5 \times 10^{-5}$ kg m$^{-2}$ s$^{-1}$, which is equivalent to $0.8$ m yr$^{-1}$. Since the observed global precipitation is $1$ m yr$^{-1}$, this estimate indicates that a major part of global precipitation and, hence, $W_P$ can be accounted for by air motions resolved at the horizontal scale of the order of 100 km. Assuming that precipitation path length $\mathcal{H}_P \sim \mathcal{H}/2$ (water precipitates from the mid troposphere) we obtain $W_P \sim Pg\mathcal{H}/2 \sim \rho_v wg\mathcal{H}/8 \sim 1$ W m$^{-2}$, i.e. $W_P$ is somewhat less than $W_K$ but of the same order of magnitude.

Total amount $\sigma_c \equiv \int_0^{z(p_t)} \rho_c dz$ of condensate (including liquid and ice) per unit area of the ground surface ranges between $\sigma_c \sim 0.1 - 1$ kg m$^{-2}$ (Bauer and Schluessel, 1993). For $w \sim 10^{-2}$ m s$^{-1}$ we have $|W_c| \lesssim \sigma_c wg \sim 0.01 - 0.1$ W m$^{-2}$. A major part of total condensate content is represented by cloud water, i.e. by very small condensate particles having practically the same vertical velocity as the air parcels carrying them. Such condensate particles are common in non-raining clouds, which account for 90% of all observed cloudiness (O'Dell et al., 2008). Condensate particles travelling together with the air are
present in equal amounts in descending and ascending air flows such that their contribution to $W_c$ should be close to zero. These considerations suggest that at the spatial scale where $w \sim 10^{-1}$ m s$^{-1}$ the contribution of $W_c$ to global atmospheric power is of the order of one per cent of $W_P$.

The fact that $|W_c| \ll W_P$ indicates that the condensate particles spend a negligible portion of their potential energy $gz$ they possess at the point of their formation to impact generation of kinetic energy $W_K$ at the considered spatial scale. Since for
$w \sim 10^{-2}$ m s$^{-1}$ the estimated precipitation $P \sim \rho_v w/4$ approximately accounts for observed global rainfall, consideration of smaller scale circulation patterns with $w > 10^{-2}$ m s$^{-1}$ will increase the long-term global mean value of $W_c$ if only there exists a strong positive correlation between $\rho_c$ and $w$.

Equations (37) and (38) show that the sum of the two terms depending on phase transitions, $W_P + W_c$, can be estimated from air velocity and pressure gradient alone as $W_P + W_c \approx W_P \approx W - W_K$ without any knowledge of atmospheric moisture



content $\rho_v$, local condensation rate $\dot{\rho}$ or precipitation. This allows global $W_P$ to be estimated from re-analyses data as done in Section 5.

### 3.5   Comparison to Pauluis et al. 2000

Our assessment of the atmospheric power budget started from the thermodynamic definition of work (7). Integrated over atmospheric volume Eq. (7) yielded total atmospheric power $W = W_{III}$ (9), (3). The boundary condition $w_s = 0$ (15) turned

$W_{III}$ into $W_I$ (12). Then we used the continuity equations (6) and the equations of motion (19) with an explicitly specified interaction $\mathbf{F}_c$ between air and condensate particles to decompose $W$ (37) into three major terms, kinetic energy generation $W_K$, the gravitational power of precipitation $W_P$ and condensate loading $W_c$ (Fig. 2).

Pauluis et al. (2000) (hereafter PBH) identified two distinct terms in the atmospheric power budget, kinetic energy production and precipitation-related dissipation, and provided an expression for total power for a specific atmospheric model. An

important difference from our approach is that PBH did not derive the expression $W = W_I$ and thus could not clarify how their formulations relate to the equations of motion and continuity. The reason, as we discuss below, was an incorrect boundary condition $w_s \neq 0$ implied in the derivations of PBH (Fig. 2).

PBH assumed that droplets do not accelerate ($\mathbf{F}_a = 0$). Noting that condensate is falling at terminal velocity $\mathbf{w}_T \equiv \mathbf{w}_c - \mathbf{w}$ experiencing resistance force $\rho_c \mathbf{g}$, PBH defined the precipitation-related frictional dissipation as follows (we have added factor

$1/\mathcal{S}$ to enable comparison with our results), see Eq. (2) of PBH:

$$W_P^* \equiv -\frac{1}{\mathcal{S}} \int_{\mathcal{V}} \rho_c w_T g d\mathcal{V}. \tag{42}$$

Assuming that *at any level* $z = z_0$ in the atmosphere the upward flux of water wapor is balanced by the downward flux of condensate, see Eq. (3) of PBH,

$$\int_{z=z_0} \rho_c w_c d\mathcal{S} + \int_{z=z_0} \rho_v w d\mathcal{S} = 0, \tag{43}$$

PBH obtained, see their Eq. (4),

$$W_P^* = \frac{1}{\mathcal{S}} \int_{\mathcal{V}} (\rho_v + \rho_c) w g d\mathcal{V}. \tag{44}$$

To find out how this formulation relates to ours, we use Eq. (21) and the continuity equations for dry air $\nabla \cdot (\rho_d \mathbf{v}) = 0$ and water vapor $\nabla \cdot (\rho_v \mathbf{v}) = \dot{\rho}$, where $\rho_d$ and $\rho_v$ are densities of dry air and water vapor, $\rho = \rho_v + \rho_d$, to observe that $\int_{\mathcal{V}} \rho_v w g d\mathcal{V} = \int_{\mathcal{V}} \rho w g d\mathcal{V}$. Using this expression and Eq. (44) we find

$$W_P^* = W_P + W_c, \tag{45}$$

where $W_P$ and $W_c$ are defined in Eqs. (39) and (40). Thus, $W_P^*$ of PBH combines two terms with distinct meanings, with $W_c$ (condensate loading) depending on the interaction between the condensate and air and $W_P$ (the gravitational power of





**Figure 2.** Formulation of total atmospheric power and its budget in the present work (non-white rectangles) and in the work of Pauluis et al. (2000) (PBH) and Pauluis and Held (2002) (PH) (ellipses and white rectangles). The two red ellipses indicate incorrect statements. The trapezium shows a formulation PBH could have obtained for $W^*$ instead of their Eq. (10) if using our Eq. (34). See text for details.





precipitation) independent of it. Equations (42) and (44) of PBH are informative in clarifying that the sum of $W_P + W_c$ is always positive when $w_T < 0$, even though, as we discussed in Section 3.4, $W_c$ can be either positive or negative. On the other hand, Eqs. (42) and (44) do not reveal how $W_P^*$ (44) relates to the creation of potential energy $gz$ of the condensate particles by air parcels, cf. Eq. (39).

PBH further assumed that $W_P^*$ is *"proportional to the precipitation rate $P$ at the surface, which is given by the surface integral"* $-(1/\mathcal{S}) \int_{z=0} \rho_c w_T d\mathcal{S}$. In reality, however, $P = -(1/\mathcal{S}) \int_{z=0} \rho_c w_c d\mathcal{S}$, where $w_c = w + w_T$. The two integrals coincide, $-\int_{z=0} \rho_c w_c d\mathcal{S} = -\int_{z=0} \rho_c w_T d\mathcal{S}$, only if $w|_{z=0} \equiv w_s = 0$. But this is inconsistent with Eq. (43), since for $w_s = 0$ and $w_{cs} \neq 0$ Eq. (43) does not hold for $z_0 = 0$. Indeed, for $z_0 = 0$ Eq. (43) contradicts the boundary condition (15) $w_s = 0$. In particular, when local evaporation equals local precipitation, Eq. (43) gives $w_s = -\rho_{cs} w_{cs}/\rho_{vs} > 0$ (subscript $s$ denotes the corresponding surface values).

Nonetheless, despite being obtained by PBH using Eq. (43), equation (44) is correct. Its validity cannot be proved within PBH's approach that is based on Eq. (43). However, Eq. (44) can be obtained from Eq. (42) by analogy with Eq. (22) – using the continuity equations (17), (18) and Eq. (21). The surface integral in Eq. (22) is proportional to $zw_s$ and is thus zero at $z = 0$ irrespective of the magnitude of air velocity $w_s$ at the surface. So even if $w_s$ is specified incorrectly, it does not affect $W_P^*$.

But, as noted in Section 3.1, $w_s$ influences total atmospheric power. Since their basic equation (43) implies $w_s \neq 0$, PBH could not derive $W = W_I$ (1) from $W = W_{III}$ (3) (PBH should have been aware of the latter equation since it was listed by Pauluis and Held (2002) albeit without a derivation or reference). Without $W = W_I$ PBH could not, as we did, use the equations of motions to investigate the power budget by decomposing $W$ into $W_P^*$ and kinetic energy production (Fig. 2).

Instead, PBH had to postulate, see their Eq. (9), that *"total mechanical work by resolved eddies"* $W^*$ is equal to the sum of the *"frictional dissipation associated with convective and boundary-layer turbulence"* $W_F^*$ and the *"total dissipation rate due to precipitation"* $W_P^*$ (Fig. 2):

$$W^* \equiv W_F^* + W_P^*. \tag{46}$$

(In the notations of PBH $W^* = W_{\text{tot}}$, $W_F^* = W_D$, $W_P^* = W_p$, $w_T = -v_T$.)

Since no general specification for these turbulent processes exists, this formulation *per se*, unlike Eqs. (37)-(40), cannot guide an assessment of $W^*$ from observations. One has to specify $W_F^*$, which can only be done using the equations of motions (19) which PBH did not consider. Rather, the following formulation for $W_F^*$ was proposed by PBH with a reference to the model of Xu et al. (1992):

$$W_F^* = \frac{1}{\mathcal{S}} \int_{\mathcal{V}} \overline{\rho}_m g w \left[ \frac{\Theta'}{\overline{\Theta}} + \left( \frac{R_v}{R_d} - 1 \right) \frac{\rho_v}{\overline{\rho}_m} - \frac{\rho_c}{\overline{\rho}_m} \right] d\mathcal{V}, \tag{47}$$

This formula, which is Eq. (8) of PBH, is formally identical to the sum of Eqs. (8) and (9) of Xu et al. (1992). According to Xu et al. (1992), it describes the rate of buoyancy generation by convective eddies. Here $\rho_m \equiv \rho + \rho_c$ is the total density of gaseous air and condensate, $\overline{\rho}_m = \overline{\rho}_m(z)$ is the horizontally averaged total density, $\Theta' \equiv \overline{\Theta} - \Theta$ is the local departure of potential temperature $\Theta$ from its horizontally averaged value.



Summing Eq. (44) and Eq. (47) yields Eq. (10) of PBH for total atmospheric power[4]:

$$W^* = \frac{1}{\mathcal{S}} \int\limits_{\mathcal{V}} wg \left[ \bar{\rho}_m \frac{\Theta'}{\overline{\Theta}} + \rho_v \frac{R_v}{R_d} \right] d\mathcal{V}. \tag{48}$$

The formulation of the atmospheric power budget offered by PBH, Eqs. (46), (44) and (47), leaves the following question open: if, as proposed by PBH, the kinetic energy of air is generated by the buoyancy flux (i.e. by the lighter air parcels ascending and the heavier air parcels descending), what generates potential energy $gz$ of condensate particles, which further dissipates in

the form of $W_P^*$? How can we interpret the fact that the buoyancy flux *does not* generate the potential energy of condensate particles? Our own formulation of the atmospheric power budget and specifically Eq. (22) explains that the potential energy of condensate particles is generated because in the presence of condensation there is more gas rising (hence expanding) than descending (hence contracting). The net difference in the work of these air parcels, which is proportional to condensation rate and unrelated to buoyancy, is what creates the potential energy of condensate particles.

A major caveat with Eqs. (47) and (46) of PBH is that the model of Xu et al. (1992) employs a formulation of the continuity equations inconsistent with PBH's approach. The model of Xu et al. (1992) derives from the model of Krueger (1985, 1988), which in its turn is based on the model of Lipps and Hemler (1982). In the latter model the continuity equation (Eq. 3 of Lipps and Hemler, 1982) has a zero source/sink, $\dot{\rho} = 0$. If $\mathbf{V}$, defined as *air velocity* in the model of Lipps and Hemler (1982), is the velocity of gaseous air, then Eq. 3 of Lipps and Hemler (1982) corresponds to Eq. (17) with $\dot{\rho} = 0$. In this case, since the

gravitational power of precipitation $W_P$ (39) is proportional to $\dot{\rho}$, this model cannot evaluate $W_P^*$ or $W_P$.

On the other hand, if *the vector velocity of the air* $\mathbf{V}$ used in the model of Lipps and Hemler (1982) is the mean velocity $\mathbf{v}_m$ of gaseous air and condensate, then the continuity equation written for $\mathbf{v}_m$ with $\dot{\rho} = 0$ is correct. Indeed, from the continuity equations (17) and (18) we have

$$\nabla \cdot (\rho_m \mathbf{v}_m) = 0, \quad \mathbf{v}_m \equiv \frac{\rho \mathbf{v} + \rho_c \mathbf{v}_c}{\rho_m}, \quad \rho_m \equiv \rho + \rho_c. \tag{49}$$

This is Eq. (3) of Lipps and Hemler (1982) if their $\mathbf{V}$ is replaced by $\mathbf{v}_m$.

The same velocity $\mathbf{V}$ is used by Lipps and Hemler (1982) in their equations of motion that are based on an anelastic approximation. With $\mathbf{V} = \mathbf{v}_m$, the general form of the equations of motion would be

$$\rho_m (\mathbf{v}_m \cdot \nabla) \mathbf{v}_m = -\nabla p + \rho_m \mathbf{g} + \mathbf{F}. \tag{50}$$

If we also assume that for $z = 0$ we have $\mathbf{v}_m \cdot \mathbf{n} = 0$ (which is also incorrect, as it implies $w_s \neq 0$, see Section 3.1), then

multiplying Eq. (50) by $\mathbf{v}_m$, integrating the resulting equation over the atmospheric volume $\mathcal{V}$ and using the divergence theorem we find

$$-\frac{1}{\mathcal{S}} \int\limits_{\mathcal{V}} \mathbf{F} \cdot \mathbf{v}_m d\mathcal{V} = -\frac{1}{\mathcal{S}} \int\limits_{\mathcal{V}} \mathbf{v}_m \cdot \nabla p d\mathcal{V}. \tag{51}$$

---

[4]PBH provided their expression for $W_F^*$ (Eq. 8 of PBH) and $W^*$ (Eq. 10 of PBH) without either specifying the integration domain or writing out the differential $d\mathcal{V}$. Since the differential $d\mathcal{V}$ is not dimensionless, the latter omission changes the dimension of the expression under the sign of the integral. Such loose notations can cause errors. In particular, Eq. (48) for $W^*$ (Eq. 10 of PBH) would correspond to atmospheric power only for an integration domain enclosed by a surface with $\mathbf{v} \cdot \mathbf{n} = 0$ (see Sestion 3.1), while $W_F^*$ (47) has no such implications and can be defined for any local volume.



We can see that after this procedure the second term in the right-hand part of Eq. (50), $\rho_m \mathbf{g} \equiv (\rho + \rho_c)\mathbf{g}$, which gave rise to the condensate-related terms $W_P + W_c$ in Eq. (37), has disappeared. In its absence, the right-hand part of Eq. (51) looks very similar to $W_I$ (1) except $\mathbf{v}$ in Eq. (1) is now replaced by $\mathbf{v}_m$ in Eq. (51). Thus, if the incorrect boundary condition with $w_s \neq 0$ is used and no clear distinction is recognized between using $\mathbf{v}_m$ and $\mathbf{v}$ in the equations of motion, one could be misled by Eq. (51) and conclude that $W_I$ describes production (and dissipation) of the kinetic energy of air even in the presence of condensation. Thus, having in mind Eq. 3 of Lipps and Hemler (1982), PBH could conclude that $W_I$ (1) describes kinetic power alone and not the total atmospheric power $W$ (37). This is consistent with Pauluis (2015) referring to $W_I$ estimated by Laliberté et al. (2015) as *kinetic energy production*.

All these confusions result from the fact that PBH did not explicitly consider the continuity equations when deriving their atmospheric power budget. While PBH formulate $W_P^*$, $W_F^*$ and $W^*$ using velocity $w$ of gaseous air, they then used the model of Lipps and Hemler (1982) where *air velocity*, according to Eq. 3 of Lipps and Hemler (1982), is the mean velocity of gaseous air and condensate. Furthermore, Eq. 3 of Lipps and Hemler (1982) is mathematically inconsistent with their Eqs. 6, 7 and 8 for mixing ratios of water vapor, cloud water and rain water. In the latter equations $\dot{\rho}$ is not ignored. This inconsistency does not affect the results of Lipps and Hemler (1982), who are only concerned with kinetic energy production, but undermines calculations of total atmospheric power since here the main terms are proportional to $\dot{\rho}$, see Fig. 1. Unsurprisingly, the estimate $W_P^* = 3.6$ W m$^{-2}$ derived by PBH from the model of Lipps and Hemler (1982) turned out to be unrealistic. A later estimate of $W_P^*$ from the TRMM observations by Pauluis and Dias (2012) first yielded 1.8 W m$^{-2}$ and after a corrigendum 1.5 W m$^{-2}$ (Pauluis and Dias, 2013), which is by a factor of 2.4 smaller than the model-derived $W_P^*$ (see also Makarieva et al., 2013b).

Since PBH did not make it clear how their $W_F^*$ (47) relates to the equations of motion of moist air, to understand how $W^*$ of PBH relates to total atmospheric power $W$ (37) we need to specify such a relationship. We assume that $W_F^* = W_F$, where $W_F$ is defined in Eq. (20). PBH also assumed that the particles do not accelerate, i.e. $\mathbf{F}_a = 0$ in Eq. (29) and $\mathbf{F}_c = \rho_c \mathbf{g}$ in Eq. (20). Then we conclude from Eq. (20) that $W = W^*$ under the additional assumption $\dot{K} = 0$.

Finally, we note that while $W_P^*$ (44) of PBH obtained from Eq. (43) is mathematically equivalent to $W_P + W_c$ (39), (40) we obtained using Eq. (21),

$$W_P + W_c = \frac{1}{\mathcal{S}} \int_{\mathcal{V}} (\rho_v + \rho_c) w g d\mathcal{V} = \frac{1}{\mathcal{S}} \int_{\mathcal{V}} (\rho_v + \rho_d + \rho_c) w g d\mathcal{V} = \frac{1}{\mathcal{S}} \int_{\mathcal{V}} \rho_m w g d\mathcal{V}, \qquad (52)$$

our formulation provides a simpler expression for total atmospheric power $W^*$ in a model where kinetic energy is produced by the buoyancy flux, i.e. where

$$W_F^* = \frac{1}{\mathcal{S}} \int_{\mathcal{V}} w g (\overline{\rho}_m - \rho_m) d\mathcal{V}. \qquad (53)$$

(Note that Eq. (47) of PBH is an approximation of the exact buoyancy flux given by Eq. (53). Equation (47) of PBH neglects the impact of horizontal pressure perturbations to density perturbations and is valid on a small horizontal scale only where horizontal pressure differences can be neglected compared to temperature differences. Generally, the anelastic approximation underlying the model of Lipps and Hemler (1982) used by PBH is not valid on a global scale (see, e.g., Davies et al., 2003).)





Indeed, adding the last expression in Eq. (52) to $W_F^*$ (53) we immediately obtain from Eq. (46)

$$W^* = \frac{1}{\mathcal{S}} \int_{\mathcal{V}} \overline{\rho}_m g w d\mathcal{V} \qquad (54)$$

instead of Eq. (48). For those who seek to calculate $W^*$ from a numerical model Eq. (54) makes a difference not just in terms of physical clarity but also in terms of computer time. Indeed, total power of an atmosphere powered by the buoyancy flux turns out to be a simple function of just two variables ($w, \overline{\rho}_m$) rather than a more complex expression involving five variables

($w, \overline{\rho}_m, \overline{\Theta}, \Theta', \rho_v$) offered by Eq. (48).

In summary, PBH obtained a valid expression for precipitation-related dissipation $W_P^* = W_P + W_c$ (44). At the same time, PBH did not derive a general relationshp $W = W_I$ for total atmospheric power. Nor did PBH present a consistent derivation of the atmospheric power budget from an explicit consideration of the continuity equations and the equations of motion with correctly specified boundary conditions. As we examine in the next section, these limitations contributed to errors in analyses

that built on that of PBH.

## 4    Practical implications

In a recent effort to constrain the atmospheric power budget, Laliberté et al. (2015) used the thermodynamic identity

$$T\frac{ds}{dt} \equiv \frac{dh}{dt} - \alpha\frac{dp}{dt} + \mu\frac{dq_m}{dt}, \qquad (55)$$

where $s$ is entropy, $h$ is enthalpy, $\mu$ is chemical potential (all per unit mass of moist air), $\alpha = 1/\rho$ is specific air volume and $q_m$

is the mass fraction of total water.[5] Laliberté et al. (2015) neglected the atmosphere's liquid and solid water content and put $q_m = q_v$, where $q_v$ is the mass fraction of water vapor.

Integrating Eq. (55) over atmospheric mass $\mathcal{M}$ and taking the long-term mean of the resulting integral Laliberté et al. (2015) sought to estimate $W_I$ (1), since $d\mathcal{M} = \rho d\mathcal{V}$ and

$$-\frac{1}{\mathcal{S}} \int_{\mathcal{M}} \frac{dp}{dt} \alpha d\mathcal{M} = -\frac{1}{\mathcal{S}} \int_{\mathcal{V}} \left( \frac{\partial p}{\partial t} + \mathbf{v} \cdot \nabla p \right) d\mathcal{V} = W_I - \frac{1}{\mathcal{S}} \int_{\mathcal{V}} \frac{\partial p}{\partial t} d\mathcal{V}. \qquad (56)$$

Laliberté et al. (2015) assumed that after the integration of Eq. (55) the enthalpy term vanishes, $\int_{\mathcal{M}} (dh/dt)d\mathcal{M} = 0$. They justified this by noting that the atmosphere is approximately in a steady state. However, using Eq. (25) with $X = h$ we have

$$I_h \equiv \frac{1}{\mathcal{S}} \int_{\mathcal{M}} \frac{dh}{dt} d\mathcal{M} = -\frac{1}{\mathcal{S}} \int_{\mathcal{V}} h\dot{\rho} d\mathcal{V} \neq 0. \qquad (57)$$

We can see that $I_h$ is zero only if there are no sources or sinks of water vapor in the atmosphere, i.e. when $\dot{\rho} = 0$.

The physical meaning of this is as follows. Enthalpy change per unit time in all air parcels (material elements) combined is

indeed zero in a steady-state atmosphere. However, $dh/dt$ is *not* equal to enthalpy change per unit mass of a given air parcel.

---

[5]The unconventional sign at the chemical potential term follows from $\mu$ being defined in Eq. (55) relative to dry air: hence, when the relative dry air content diminishes this term is negative. For details see p. 8 in the Supplementary Materials of Laliberté et al. (2015).



(Likewise $pd\alpha/dt$ is not equal to work per unit time per unit mass of a given air parcel, see Eq. (11) above.) Indeed, for an air parcel of mass $\tilde{m}$ total enthalpy of the parcel is $\tilde{h} \equiv h\tilde{m}$; its change per unit mass is $(d\tilde{h}/dt)/\tilde{m} = dh/dt + (h/\tilde{m})d\tilde{m}/dt \neq dh/dt$. Therefore, the integral of $dh/dt$ over total atmospheric mass is not zero. As is clear from Eq. (57), it is the integral of $(d\tilde{h}/dt)/\tilde{m}$ that is zero.

Since the expression for $I_h$ (57) is straightforward, the question arises why Laliberté et al. (2015) put $I_h = 0$. A likely explanation is that air velocity $\mathbf{v}$ in the formulation of atmospheric power $W = W_I$ used by Laliberté et al. (2015) was replaced by the mean velocity $\mathbf{v}_m$ of gaseous air and condensate combined. If $\mathbf{v}$ is replaced by $\mathbf{v}_m$ in the definition of material derivative (5) then by analogy with Eq. (25) for any quantity $X$, including $X = h$, using Eq. (49) we have

$$\int_{\mathcal{V}} (\rho + \rho_c)\frac{dX}{dt_m}d\mathcal{V} = 0, \quad \frac{dX}{dt_m} \equiv (\mathbf{v}_m \cdot \nabla)X. \tag{58}$$

While Eq. (58) indicates why Laliberté et al. (2015) could have put $I_h = 0$ when integrating Eq. (55), it does not justify this choice. As we discussed in Section 2, work in the atmosphere is performed by expanding *air parcels*. The local work of air parcels per unit time per unit volume is given by $W_a = p\nabla \cdot \mathbf{v}$ (8), where $\mathbf{v}$ is the velocity of the *gaseous air* (see also Pauluis and Held, 2002, their Eq. 4). Namely this velocity describes how the parcel's volume changes as it moves. The same velocity $\mathbf{v}$ is retained in the formulation $W = W_I$ (37), which derives from $W_a$ (7). Using mean velocity $\mathbf{v}_m$ in an assessment of global

atmospheric power is an error. Put in a different way, if Laliberté et al. (2015) used $\mathbf{v}_m$, Eq. (49) and Eq. (58) instead of $\mathbf{v}$, Eq. (17) and Eq. (25), the magnitude they estimated from Eq. (55) is not that of atmospheric work output.[6]

The magnitude of $I_h$ (57) can be estimated assuming that evaporation and condensation are localized at, respectively, the surface $z = 0$ and the mean condensation height $z = \mathcal{H}_P$. This approximation allows $\dot{\rho}$ in (57) to be explicitly specified using the Dirac's delta function $\delta(z)$:

$$\dot{\rho} = E(x,y)\delta(z - l_f) - P(x,y)\delta(z - \mathcal{H}_P), \quad \int_0^{z(p_t)} \dot{\rho}\,dz = E(x,y) - P(x,y). \tag{59}$$

$E(x,y)$ and $P(x,y)$ are local evaporation and precipitation at the surface (kg m$^{-2}$ s$^{-1}$) with global averages $E = P$, $l_f \sim 10^{-7}$ m is a microscopic length scale of the order of one free path length of air molecules (Section 3.1). From (59) we have

$$I_h \approx -Eh_s + Ph(\mathcal{H}_P) \equiv -P\Delta h_c, \quad \Delta h_c \equiv h_s - h(\mathcal{H}_P), \quad h = c_pT + \mathcal{L}q_v. \tag{60}$$

Here $c_p = 10^3$ J kg$^{-1}$ K$^{-1}$ is heat capacity of air at constant pressure, $\mathcal{L} = 2.5 \times 10^6$ J kg$^{-1}$ is latent heat of vaporization.

We can see that $I_h$ is proportional not to the difference between evaporation and precipitation (which can be locally arbitrarily small), but to the intensity of the water cycle $E = P$ multiplied by the difference in air enthalpy between $z = 0$ and $z = \mathcal{H}_P$.

For $\mathcal{H}_P \approx 2.5$ km (Makarieva et al., 2013b) and $q_v(\mathcal{H}_P) \ll q_{vs}$ we have $-P\Delta h_c = -P(c_p\mathcal{H}_P\Gamma + \mathcal{L}q_{vs}) \approx -1$ W m$^{-2}$. Here $q_{vs} = 0.0083$ corresponds to global mean surface temperature $T_s = 288$ K and relative humidity 80%; mean tropospheric

---

[6]Our hypothesis that Laliberté et al. (2015) based their analysis on Eqs. (49) and (58) appears to be supported by an anonymous referee (see doi:10.5194/acp-2016-203-RC3, p. C3). The referee noted that Laliberté et al. (2015) assumed "the absence of mass source and sink in the continuity equation", cf. Eq. (49).




lapse rate is $\Gamma = 6.5$ K km$^{-1}$. Global mean precipitation $P$ (measured in a system of units where liquid water density $\rho_w = 10^3$ kg m$^{-3}$ is set to unity) is equal to $P \sim 1$ m yr$^{-1}$, which in SI units corresponds to $P = 3.2 \times 10^{-5}$ kg m$^{-2}$ s$^{-1}$. A more sophisticated estimate presented in Appendix B yields $I_h = -1.6$ W m$^{-2}$ with an accuracy of about 30%.

These estimates show that the enthalpy term in Eq. (55) cannot be neglected on either theoretical or quantitative grounds. By absolute magnitude $I_h$ (60) is greater than one third of the total atmospheric power $W \approx 4$ W m$^{-2}$ estimated by Laliberté et al. (2015) for the MERRA re-analysis (3.66 W m$^{-2}$) and the CESM model (4.01 W m$^{-2}$).

Laliberté et al. (2015) appear to have first calculated the mass integral of $T ds/dt$ from the right-hand side of Eq. (55), then calculated $\mu dq_m/dt$ from atmospheric data and then used the obtained values and again Eq. (55) to estimate the total atmospheric power as $-(1/\mathcal{S}) \int_{\mathcal{M}} \alpha (dp/dt) d\mathcal{M}$. In such a procedure, putting $\int_{\mathcal{M}} (dh/dt) d\mathcal{M} = 0$ would overestimate $W$ by about 1.6 W m$^{-2}$. Since the omitted term is proportional to the global precipitation, it is required not only for estimating $W$, but also for understanding any trends related to changing precipitation.

Note that even in the correct form, with the enthalpy term retained, Eq. (55) does not provide a theoretical constraint on $W$. This equation is an identity: it defines $ds/dt$ in terms of measurable atmospheric properties. As seen in Eq. (12), $W$ can be estimated from these data without involving entropy. We illustrate this in the next section.

# 5 Assessing the atmospheric power budget from re-analyses

## 5.1 $W$, $W_K$ and $W_P$ in MERRA

In meteorological databases including the MERRA dataset MAI3CPASM that we used $dp/dt$ is often represented as a separate variable named *pressure velocity* (omega). We estimated $W_K$ from Eq. (38) and $W$ as

$$W = \langle \Omega \rangle, \ \ \Omega \equiv -\frac{1}{\mathcal{S}} \int_{\mathcal{V}} \omega d\mathcal{V}, \ \ \omega \equiv \frac{dp}{dt} \equiv \frac{\partial p}{\partial t} + \mathbf{v} \cdot \nabla p. \tag{61}$$

Time averaging denoted as $\langle \rangle$ was made for 1979-2015. In the MAI3CPASM dataset the instantaneous values of pressure velocity, horizontal velocity $\mathbf{u}$ and geopotential height (from which the horizontal pressure gradient $\nabla p$ is estimated, Eq. (C9) 595 in Appendix C) are provided every three hours on the $1.25° \times 1.25°$ grid for 42 pressure levels from 1000 hPa to 0.1 hPa.

The results depend on how the surface values $X_s$ ($X = \omega$ or $\mathbf{u} \cdot \nabla p$) are estimated. One approach is to assume that the surface air velocity is zero, $\mathbf{v}_s = 0$; another is to find $X_s$ by extrapolation from the two pressure levels nearest to the surface (see Appendix C for details). We report results obtained assuming $\mathbf{v}_s = 0$ unless stated otherwise.

Assuming $\mathbf{v}_s = 0$ we find $W = 3.27$ W m$^{-2}$, $W_K = 2.46$ W m$^{-2}$ and their difference $W_P = W - W_K = 0.81$ W m$^{-2}$ 600 (Fig. 3). The corresponding values obtained by extrapolation are $W = 3.01$ W m$^{-2}$, $W_K = 2.62$ W m$^{-2}$ and $W_P = 0.39$ W m$^{-2}$. The alternative surface values have a relatively minor impact on $W_K$ and $W$ – changing them by 7% and −9%, respectively, see Section C3 in Appendix C. But since these changes are of opposite sign, $W_P = W - W_K$ is more significantly affected.

Both estimates of $W_P$, 0.81 and 0.39 W m$^{-2}$, are smaller than the independent estimate of global gravitational power of precipitation $W_P = 1.0$ W m$^{-2}$ (with uncertainty range from 0.9 to 1.3 W m$^{-2}$) obtained from consideration of precipitation



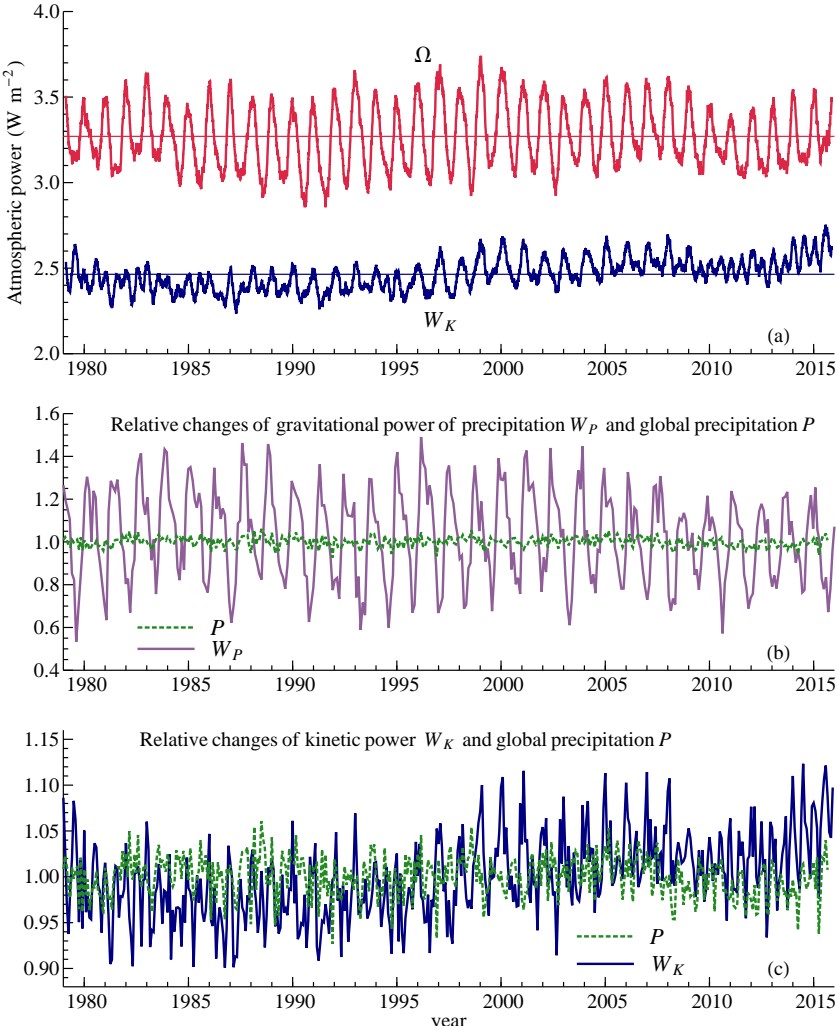

**Figure 3.** (a) Time series (90-day running mean of 3-hourly values) of the kinetic power (the blue lower curves) and the omega integral (61), (C6) corresponding to total atmospheric power (the red upper curves). The straight lines show the long-term mean values, see text. (b) and (c): Relative changes of the gravitational power of precipitation $W_P = \Omega - W_K$ (b) or kinetic power $W_K$ (c) compared to global precipitation $P$ (Adler et al., 2003, GPCP v. 2.2): monthly mean $W_P$, $W_K$ and $P$ divided by their long-term mean of $0.81$ W m$^{-2}$ for $W_P$, $2.46$ W m$^{-2}$ for $W_K$ in 1979-2015 and $2.67$ mm d$^{-1}$ for $P$ in 1979-2014.

rates and mean condensation height (Appendix B). This discrepancy can be explained by the dependence of MERRA-derived $W_P$ on data resolution. As illustrated by Eqs. (37)-(39), $W_P$ derives from the vertical air velocity and thus reflects rainfall associated with air motions at the considered scale. Meanwhile the theoretical estimate of $W_P$ is based on the total observed rainfall and thus assesses cumulative gravitational power of precipitation at all scales. If $W_P$ estimated from MERRA coincided with precipitation-based estimate of $W_P$, that would mean that no rainfall is associated with air motions at a scale finer than





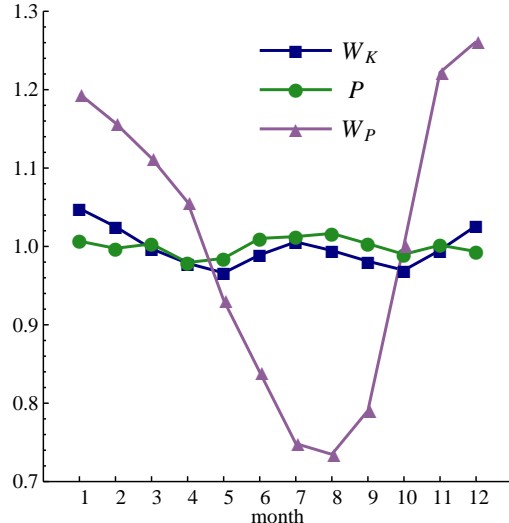

**Figure 4.** Long-term monthly means of $W_K$, $W_P$ and $P$ divided by their 1979-2014 annual mean values of, respectively, 2.46 W m$^{-2}$, 0.81 W m$^{-2}$ and 2.67 mm d$^{-1}$. The year 2015 was excluded because of incomplete precipitation data (see Fig. 3b,c).

100 km. Since the scale of convection can be of the order of a few kilometers or less, apparently some rain must remain unresolved by the larger-scale motions. Therefore, the fact that $W_P$ in MERRA is lower than its precipitation-based estimate can be explained by resolution rather than by inconsistencies in the database.

      Laliberté et al. (2015) using their Eq. (55) estimated total atmospheric power $W_L$ from the MERRA database as $W_L = 3.66$ W m$^{-2}$, which is higher than either of our two estimates. As discussed in Section 4, the difference between $W$ and

$W_L$, caused by the omission of the enthalpy term in Eq. (55), should be equal to $I_h = -1.6$ W m$^{-2}$, see Eq. (60). The actual difference is the same order of magnitude but is about 60% smaller: $W - W_L = -0.39$ W m$^{-2}$ or $-0.65$ W m$^{-2}$. Here data resolution is again a possible reason for the underestimate: since $I_h$ (60) is proportional to $P$, it should be underestimated when precipitation is not fully resolved. Another possible reason is the correction procedure applied by Laliberté et al. (2015) to the MERRA data; this is discussed in Section 5.2.

We see that seasonal changes of $W_P$ do not correlate with seasonal changes of precipitation (Fig. 3b). Moreover, on a monthly scale, the seasonal variability in $W_P$ is about one order of magnitude larger than variability of global precipitation $P$. In contrast, kinetic power $W_K$ appears correlated with $P$ (Fig. 3c and Fig. 4). On a monthly scale, the seasonal variability in $W_K$ is of the same order as in $P$, while in $W_P$ it is about one order of magnitude larger. Both $W_K$ and $P$ have two peaks, one in summer and another in winter (Fig. 4). The seasonal variability of $W_K$ and $P$ does not exceed five per cent. Meanwhile

$W_P$ in December is nearly twice its value in August. The minimum of $W_P$ in July and August corresponds to the maximum of global precipitation.

      While the global mean values of $W$ and $W_K$ in MERRA differ by a relatively small margin, the local values of $\langle \omega \rangle$ and $\mathbf{u} \cdot \nabla p$ differ significantly (Fig. 5). The kinetic power per unit mass, $-\mathbf{u} \cdot \nabla p / \rho$, has a relatively uniform spatial distribution. It




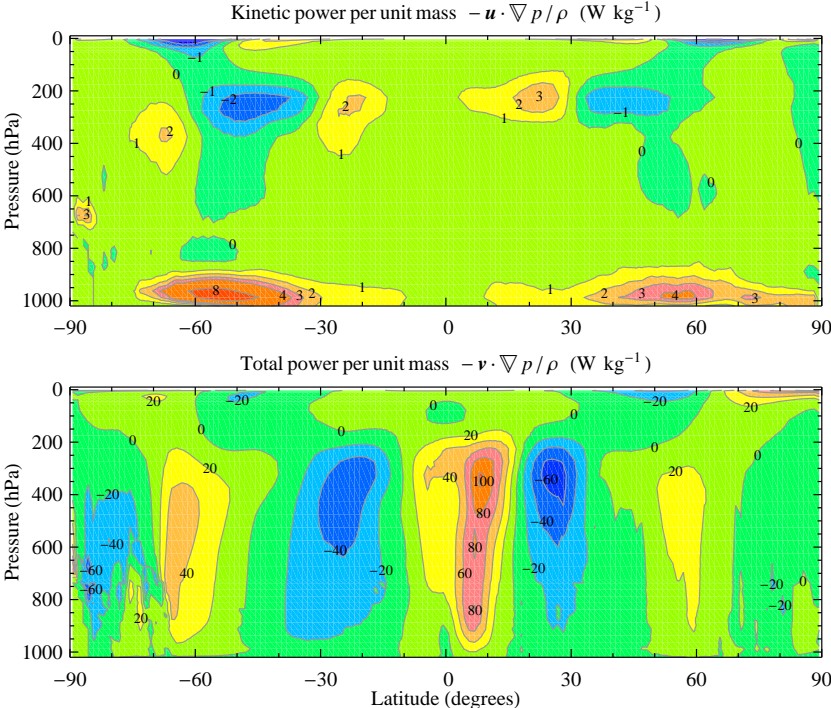

**Figure 5.** The zonally averaged kinetic $W_K$ and total $W$ atmospheric power versus latitude and pressure level (mean values for $1979-2015$).

is nearly ubiquitously positive in the lower atmosphere: we found that 59% of global kinetic power is generated below 800 hPa.

In the upper atmosphere negative kinetic power is found in the region of the atmospheric heat pumps (Ferrel cells) (Makarieva et al., 2017). Meanwhile $-\omega/\rho$ is positive (negative) in the regions of ascent (descent). Note that since work per unit time per unit volume of an air parcel, $W_a = p\nabla \cdot \mathbf{v}$ (8) is not equal either to $\mathbf{v} \cdot \nabla p$ or to $\mathbf{u} \cdot \nabla p$, the regions where the latter magnitudes are positive are not the regions where the air parcels perform positive work (cf. Tailleux, 2010; Makarieva et al., 2017).

While local values of $I_K \equiv -\int_0^{z(p_t)} \mathbf{u} \cdot \nabla p \, dz$ (W m$^{-2}$) are similar to their global mean value $W_K = 2.5$ W m$^{-2}$, local values

of $I_\omega \equiv -\int_0^{z(p_t)} \omega \, dz$ (W m$^{-2}$) can differ from their global mean value $W = 3.3$ W m$^{-2}$ by up to two orders of magnitude (Fig. 6). Indeed, the vertical pressure gradients $|\nabla_w p| \sim p_s/\mathcal{H} = 10^4$ Pa km$^{-1}$ are four orders of magnitude larger than typical horizontal pressure gradients $|\nabla_u p| \sim 1$ Pa km$^{-1}$. With $w/u_p \sim 10^{-2}$ we have for the ratio $|w\nabla_w p/(u_p\nabla_u p)| \sim 10^2$ (here $u_p$ is the cross-isobaric horizontal velocity component, see Section 3.4).

This means that the accuracy of the determination of $W$ and, hence, $W_P$ is different from that of $W_K$. The global values of

$W$ and $W_P$ represent the small differences between two larger terms associated with ascending and descending air. Since $W$ and $W_P$ are of the same order of magnitude as $W_K$, in order to retrieve $W$ and $W_P$ with the same accuracy as $W_K$, one needs to perform the observations of $w\nabla_w p$ with a two orders of magnitude better accuracy than $u_p\nabla_u p$. For example, if $u_p\nabla_u p$ is determined with an accuracy of 10%, then $w\nabla_w p$ must be determined with an accuracy of around 0.1%.





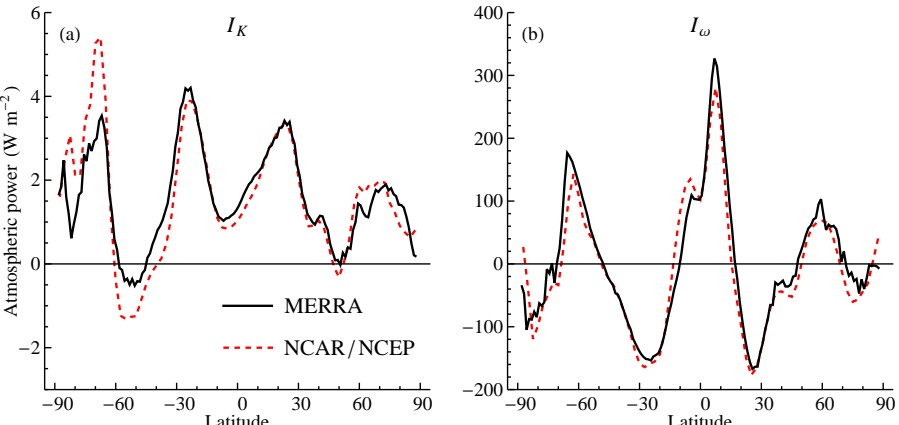

**Figure 6.** Long-term mean zonally averaged atmospheric power calculated from daily mean data for 1979-2015 in the MERRA versus NCAR/NCEP re-analysis as dependent on latitude (black solid curve: MERRA, red dashed curve: NCAR/NCEP). (a) $I_K \equiv -\int_0^{z(p_t)} \mathbf{u} \cdot \nabla p \, dz$, (b) $I_\omega \equiv -\int_0^{z(p_t)} \omega \, dz$; $p_t = 0.1$ hPa for MERRA and $p_t = 100$ hPa for NCAR/NCEP.

However, this degree of accuracy is unobtainable, since the major source of information about the vertical air flow is the

continuity equation and the observations of the horizontal air flow (see Appendix D for details). This uncertainty about vertical flows results in major uncertainties in estimating the associated power from available data as we show in the next section.

## 5.2    Atmospheric power budget in the MERRA versus NCAR/NCEP re-analyses

Kinetic power $W_K$ (38) is derived from horizontal wind velocities. As these velocities at the scale resolved in the re-analyses are larger than vertical velocities and thus have smaller relative errors, $W_K$ estimates should be more robust than $W$ and $W_P$

that require vertical velocities. This is confirmed by comparison of zonally averaged $I_K$ across the daily mean MERRA and NCAR/NCEP databases (Fig. 6a). Here not only the profiles of zonally averaged $I_K$ are similar in value at most latitudes, but the global means differ by only 10%: 1.56 W m$^{-2}$ for the NCAR/NCEP and 1.73 W m$^{-2}$ for the MERRA (see Appendix C for details).

With global atmospheric power $W$ the situation is different. The dependence of the zonally averaged vertically integrated

pressure velocity $I_\omega$ on latitude in the NCAR/NCEP versus the MERRA database is shown in Fig. 6b. The differences are again relatively small. However, as $I_\omega$ can locally exceed its global mean value by about two orders of magnitude, these small local differences between $I_\omega$ from the two databases lead to marked differences in global atmospheric power $W$. Our estimate of $W$ from the NCAR/NCEP data is in fact *negative*: $-7$ W m$^{-2}$ versus $2.46$ W m$^{-2}$ in MERRA (using daily mean values).

To our knowledge, atmospheric power has not been systematically assessed in re-analyses in the straightforward way out-

lined by Eq. (61) – i.e. as the integral of pressure velocity over atmospheric volume. Thus we cannot compare our NCAR/NCEP results with any published estimate. Rather, past estimates of atmospheric power considered total dissipation rate in the atmospheric energy cycle, i.e. work per unit time of the turbulent friction force $W_D$ (47) – see, e.g., Eq. (A3) of Boer and Lambert



(2008). Comparing atmospheric power across the re-analyses and global circulation models Boer and Lambert (2008, their Table 3) quoted a figure of 2.1 W m$^{-2}$ for the 6-hourly instantaneous NCAR/NCEP data. Our results for the daily mean

NCAR/NCEP data for $W_K$ is 1.75 W m$^{-2}$. This is consistent with the estimate of Boer and Lambert (2008) taking into account the dependence of $W_K$ on temporal resolution (see Fig. 8b below).

The discrepancies of $W$ and $W_P$ across the datasets depend on how vertical velocities are estimated. The problem is that local values $\dot{\rho}(x, y, z)$ of the mass source/sink in the continuity equation are not available from observations. Therefore, the vertical velocities are retrieved from horizontal velocities assuming $\dot{\rho} = 0$ in the continuity equation, see Eq. (6) and Eq. (D1)

in Appendix D. This approximation naturally results in violation of mass conservation and other known inconsistencies (Trenberth, 1991; Trenberth et al., 1995), of which a negative $W$ that we found in the NCAR/NCEP data may be one more example. Various correction procedures to ensure a more plausible wind field have been previously proposed including the so-called barotropic correction (Trenberth, 1991), see also Laliberté et al. (2015). This correction adds a $z$-invariant term $-\mathbf{v}^c(x, y)$ to the velocity vector $\mathbf{v}^*$ obtained from the continuity equation with $\dot{\rho} = 0$. Here $\mathbf{v}^c$ is determined by requiring that the resulting

velocity $\mathbf{v} \equiv \mathbf{v}^* - \mathbf{v}^c$ obeys the vertically integrated continuity equation, where now the mass sink/source is present in the form $\int_0^{z(p_t)} \dot{\rho} \, dz = E(x, y) - P(x, y)$.

In the MERRA re-analysis the retrieval of vertical velocity includes an adjustment step (Rienecker et al., 2011). To recover vertical velocities from the raw MERRA data, Laliberté et al. (2015) also performed a correction procedure; it was referred to as *standard* without providing details. Commenting on the goodness of this correction procedure Laliberté et al. (2015, p. 1 in

their Supplementary Materials) noted that *the recovered $\omega$ is very close to the vertical mass flux found in dataset MAT3NECHM*. However, Fig. 6 shows that while the vertical mass flux (represented by $I_\omega$) among the datasets can be *very close*, the residual minor differences can cause huge differences in the corresponding global values of atmospheric power $W$. Thus, the correction procedure of Laliberté et al. (2015) could significantly modify their resulting estimate $W_L$ of global atmospheric power. In particular, such a procedure could partially mask the omission of the enthalpy integral $I_h$, such that the difference between

$W_L = 3.66$ W m$^{-2}$ of Laliberté et al. (2015) and our $W = 3.27$ W m$^{-2}$, also derived from MERRA, turned out to be less than the theoretically estimated value of the omitted term $I_h = -1.6$ W m$^{-2}$ (see Section 4).

That the MERRA data yield more reasonable (e.g. positive rather than negative) long-term values for $W$ and $W_P$ can be a byproduct of the correction procedure, since it does incorporate some information about the local water cycle (and hence local moisture sources and sinks) into account. However, since none of the ways of estimating vertical motions consider the

physical processes behind the gravitational power of precipitation, the reliability of $W$ and $W_P$ derived from re-analyses remains uncertain. As discussed above, the seasonal cycle of $W_P = Pg\mathcal{H}_P$ (39) appears implausible (Fig. 8c). Likewise the multi-year trend of MERRA-derived $W_P$ (Fig. 7), whereby $W_P$ decreased in 1979-2015 by about 20%, cannot be reconciled with the trend of the MERRA-derived precipitation $P$, which rose by about the same magnitude as $W_P$ declined (see Kang and Ahn, 2015, their Fig. 10b). These inconsistencies are all likely to be artefacts due to inaccuracies in how vertical velocity

is represented in the database. As we discuss in Appendix D, the seasonal cycle of $W$ and $W_P$ can be additionally impacted by the $\partial p/\partial t$ term in the definition of $\Omega$, which is not negligible on a seasonal scale (see Fig. 13a in Appendix D).





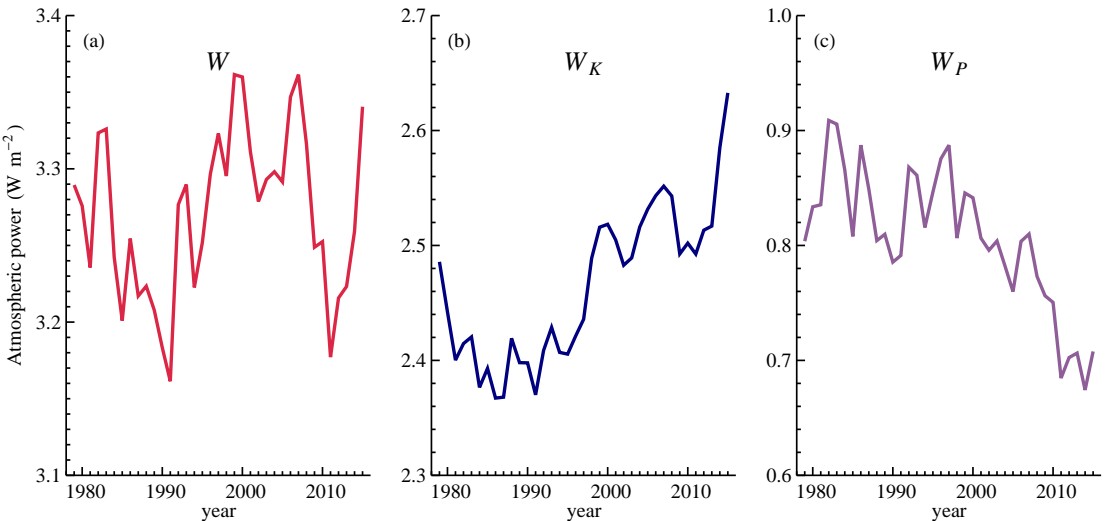

**Figure 7.** Trends in annual mean $W$, $W_P$ and $W_K$ derived from the 3-hourly instantaneous MERRA data.

Our results highlight a need for a systematic study of the atmospheric power budget across the re-analyses and also across global circulation models on the basis of Eqs. (37)-(39). The estimates of $W$ and $W_P$ from re-analyses should be compared to their independent observation-based and theoretical estimates to constrain the calculation of vertical velocities. This will improve the representation of atmospheric energetics.

### 5.3 Impact of temporal resolution

To explore the impact of temporal resolution on the atmospheric power budget we analyzed $W$, $W_P$ and $W_K$ calculated from daily and monthly mean MERRA and NCAR/NCEP data on $\omega$, $\mathbf{u}$ and $p$ (see Appendix C for details). A circulation pattern with velocity $u_p$ of the cross-isobaric flow and horizontal size $L$ has a characteristic time $\tau \sim L/u_p$. Thus, the daily averaged dataset (MERRA or NCAR/NCEP) with a spatial resolution $L \sim 100$ km will resolve atmospheric motions developing at this scale with characteristic horizontal cross-isobaric velocity $u_p \sim 1 \, \mathrm{m \, s^{-1}}$, since $100 \, \mathrm{km}/(1 \, \mathrm{m \, s^{-1}}) = 10^5 \, \mathrm{s} \approx 1$ d. It will also resolve circulation patterns like cyclones with a higher $u_p$ that develop over a horizontal scale of the order of a few hundred kilometers. On the other hand, monthly averaged datasets with the same spatial resolution will resolve the large-scale patterns of global circulation like Ferrel and Hadley cells with $\tau \sim 30$ d. The MERRA dataset with a spatial resolution of $L \sim 100$ km and instantaneous values of pressure and air velocity will resolve circulation patterns with characteristic time $\tau \sim L/u_p$ depending on $u_p$. For example, compared to the daily dataset, the instantaneous dataset will additionally resolve circulation patterns developing over $L \sim 100$ km with $u_p = 5 \, \mathrm{m \, s^{-1}}$ thus having $\tau \sim 6$ h.

Smaller-scale convective motions produce a typical rainfall $P_c \sim 10 \, \mathrm{mm \, h^{-1}}$ (e.g., Bauer and Schluessel, 1993). This rainfall is about two orders of magnitude higher than the global mean value of $P = 10^3 \, \mathrm{kg \, m^{-2} \, yr^{-1}} = 0.1 \, \mathrm{mm \, h^{-1}}$. Since global rainfall can be accounted for by vertical velocities of the order of $w \sim 10^{-2} \, \mathrm{m \, s^{-1}}$ (Fig. 1), the local convective motions should



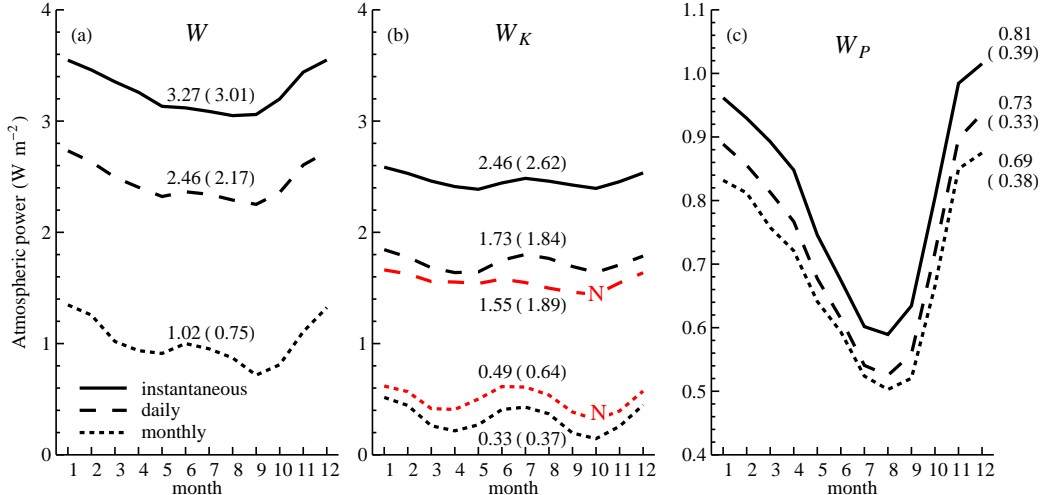

**Figure 8.** Long-term mean atmospheric power as dependent on temporal resolution: instantaneous (solid curves), daily (dashed curves) and monthly (dotted curves). (a) total power $W$ (37), (b) kinetic power $W_K$ (38), (c) gravitational power of precipitation $W_P = W - W_K$. Black curves: MERRA; red curves (marked with "N"): NCAR/NCEP. (Negative values for $W$ and $W_P$ obtained from NCAR/NCEP are not shown.) Curves are all obtained using $\mathbf{v}_s = 0$, see Eqs. (C3) and (C12). Figures at curves denote annual means in W m$^{-2}$ (values obtained using extrapolation to the surface are shown in braces; see Eqs. (C4) and (C13) and Section C3 in Appendix C for details).

involve a two orders of magnitude higher vertical velocity $w_c \sim 1$ m s$^{-1}$. A typical time scale of the convective motions is therefore $\tau_c \sim \mathcal{H}/w_c \sim 3$ h, where $\mathcal{H} = 10$ km is the tropospheric scale height.

We find that the estimated values of $W$, $W_K$ and $W_P$ increase in the MERRA re-analysis as we go from the monthly averaged to daily averaged to instantaneous datasets (Fig. 8). The same pattern is found for monthly and daily $W_K$ estimated

from NCAR/NCEP. Assuming a power law for the scaling of $W_K$

$$\frac{W_K(\tau_1)}{W_K(\tau_2)} = \left(\frac{\tau_1}{\tau_2}\right)^k, \quad k = \frac{\log[W_K(\tau_1)/W_K(\tau_2)]}{\log[\tau_1/\tau_2]}, \tag{62}$$

where $\tau$ is temporal resolution in hours, we can find $k$ from the observed monthly ($\tau = 30 \times 24$ h) and daily ($\tau = 24$ h) $W_K$ estimates and extrapolate the obtained relationship (62) to the convective scale $\tau = \tau_c$ (Fig. 9). Using the $k$ values obtained from a linear regression of MERRA and NCAR/NCEP monthly and daily values shown in Fig. 8b, we find that at the convective

scale the kinetic power should be about $W_K(\tau_c) = 3.7 - 4.5$ W m$^{-2}$ (Fig. 9). This is the kinetic power we would estimate from instantaneous values of air pressure and velocity resolved at a horizontal scale of the order of $L = H_w u_p/w_c \sim 1$ km. As we discuss in the next section, this is consistent with the theoretical estimate for condensation-induced air circulation $W_K^{\text{cond}} = 3.8$ W m$^{-2}$ (Fig. 9).




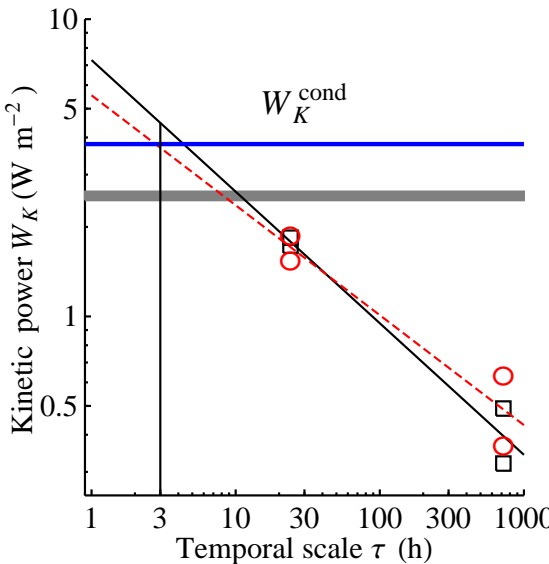

**Figure 9.** The dependence of kinetic power $W_K(\tau)$ on temporal resolution $\tau$ in MERRA (black line and squares) and NCAR/NCEP (red dotted line and circles). Symbols indicate monthly and daily values from Fig. 8b (four values for MERRA and four values for NCAR/NCEP). Lines are linear regressions (62) between the daily and monthly values. The vertical line as it intersects with the regression lines shows the estimated range of $W_K(\tau_c)$, $\tau_c = 3$ h. The grey area shows the range of instantaneous MERRA $W_K$ values from Fig. 8b. The blue horizontal line is the theoretical value $W_K^{\text{cond}}$, see Eq. (64).

## 6 Towards constraining the atmospheric power

We have derived an expression for global atmospheric power budget and assessed it from the MERRA and NCAR/NCEP re-analyses. Next we consider how our results are relevant to the problem of finding constraints on global atmospheric power.

### 6.1 The upper limit

According to the laws of thermodynamics, power output of a system cannot exceed the power output $W_C$ of the Carnot cycle. To quantify this limit on atmospheric power, three variables are required: input temperature $T_{in}$, output temperature $T_{out}$ and

735 heat flux $F$:

$$W_C = F\frac{\Delta T_C}{T_{in}}, \quad \Delta T_C \equiv T_{in} - T_{out}, \quad F = F_{\mathcal{L}} + F_S. \tag{63}$$

Here $F$ (W m$^{-2}$) is equal to the sum of latent $F_{\mathcal{L}}$ and sensible $F_S$ heat fluxes. The heat flux available to the Earth's atmospheric engine is limited by the incoming flux of solar radiation. The minimum output temperature $T_{out} = T_E$ is set by the Earth's albedo and orbital position: it is the temperature at which the atmosphere emits thermal radiation to space. If the actual output

temperature of the atmospheric engine is higher, $T_{out} \geq T_E$, the part of the atmosphere that produces work will release heat




not directly to space but to the upper atmospheric layers. The upper atmosphere will transmit heat to space without generating work. The input temperature is bounded from above by temperature $T_s$ of the Earth's surface, $T_{in} \leq T_s$, and thus depends on the magnitude of the greenhouse effect $\Delta T \equiv T_s - T_E$. However, this magnitude is *a priori* unknown. With the Earth's extensive oceans, there is a positive feedback between surface temperature and atmospheric moisture, since this moisture is itself a major greenhouse substance. The greenhouse effect on an Earth-like planet could range within broad limits: even among the planets of the solar system the maximum Carnot efficiency $\Delta T/T_s$ varies at least six-fold (Schubert and Mitchell, 2013).

If we cannot predict $\Delta T$ from theory, there is only one robust theoretical limit on $W$ that we can infer from thermodynamics: $W$ cannot be larger than approximately $F(T_S - T_E)/T_S$, where $T_S$ is the Sun's temperature. This is the upper limit that is given by consideration of entropy production on the Earth. The global efficiency of solar energy conversion into useful work amounts to about 90% (Wu and Liu (2010); see also Pelkowski (2012) for a rigorous theoretical discussion). This is about two orders of magnitude larger than the observed efficiency of atmospheric circulation. The thermodynamic theoretical upper limit alone is therefore of limited use for constraining the atmospheric power. We need additional constraints.

One arises from consideration of the dynamic properties of atmospheric water vapor. The pressure of saturated water vapor is controlled by temperature (unlike temperature and molar density as occurs for any non-condensable gas). In the presence of a gravitational field, this property has important consequences: while dry air can rise adiabatically in a state infinitely close to hydrostatic equilibrium, the saturated water vapor cannot.

The resulting dynamics can be illustrated on the example of a simple system: a horizontally homogeneous atmosphere composed of pure water vapor, where there is only vertical motion, $\mathbf{v} = \mathbf{w}$. The water vapor condenses as it rises and water returns to the Earth in its solid or liquid form. In this case kinetic energy is produced per unit volume at a rate of $-w(\partial p_v/\partial z + \rho_v g) = -w(\partial p_v/\partial z + p_v/\mathcal{H}_v)$. Here $\rho_v$ is mass density of water vapor and $\mathcal{H}_v \equiv RT/(M_v g) \approx 13$ km is the hydrostatic scale height for water vapor (see also Makarieva et al., 2013c, 2014, and references therein). If the pressure distribution of water vapor were hydrostatic, then total power $-\mathbf{w} \cdot \nabla p$ (W m$^{-3}$) would be spent to raise the potential energy of the ascending gas, leaving nothing to kinetic power. When saturated water vapor rises and cools, its partial pressure diminishes governed by decreasing temperature, the sum in braces is not zero and the hydrostatic equilibrium is not possible.

In the real atmosphere, in the presence of a sufficient amount of non-condensable gases a hydrostatic equilibrium is possible. If it is realized, the kinetic power $W_K^{\text{cond}}$ that derives from condensation of water vapor (which retains a non-hydrostatic distribution) is generated in the horizontal plane:

$$W_K^{\text{cond}} = -\frac{1}{\mathcal{S}} \int_{\mathcal{V}} \mathbf{u} \cdot \nabla p \, d\mathcal{V} = -\frac{1}{\mathcal{S}} \int_{\mathcal{V}} w \left( \frac{\partial p_v}{\partial z} + \frac{p_v}{\mathcal{H}} \right) d\mathcal{V} = -\frac{1}{\mathcal{S}} \int_{\mathcal{V}} wNRT \frac{\partial \gamma}{\partial z} d\mathcal{V} \approx \Pi RT_c = Pg\mathcal{H}_v, \tag{64}$$

$$wN \frac{\partial \gamma}{\partial z} \approx \dot{N} \equiv \frac{\dot{\rho}}{M_v} \quad (z > 0), \tag{65}$$

$$T_c \equiv -\frac{1}{P\mathcal{S}} \int_{z>0} T(z)\dot{\rho} \, d\mathcal{V} \approx 270 \text{ K}. \tag{66}$$

Here $P$ (23) and $\Pi \equiv P/M_v$ are global precipitation in units of kg m$^{-2}$ s$^{-1}$ and mol m$^{-2}$ s$^{-1}$, respectively; $\gamma \equiv p_v/p = N_v/N$, $\mathcal{H}_v \equiv RT_c/(M_v g)$, $\mathcal{H} \equiv RT_c/(Mg)$; $T_c$ is mean temperature at which condensation occurs. Its global value $T_c =$


270 K is estimated in Appendix B. Details of theoretical estimate (64) were elaborated elsewhere (see Makarieva et al., 2013c, 2015a, and references therein). Here we discuss not the result *per se*, but its implications for understanding the atmosphere as a heat engine.

From Eqs. (37)-(39) and (64) for total power $W^{\mathrm{cond}}$ of the condensation-driven circulation we obtain

$$W^{\mathrm{cond}} = W_K^{\mathrm{cond}} + W_P = \left(1 + \frac{\mathcal{H}_P}{\mathcal{H}_v}\right)\Pi R T_c = \left(1 + \frac{\mathcal{H}_P}{\mathcal{H}_v}\right)\frac{R T_c}{\mathcal{L}} F_{\mathcal{L}}. \tag{67}$$

Here $\mathcal{L} = 45 \times 10^3$ J mol$^{-1}$ is the latent heat of vaporization, $\Pi = F_{\mathcal{L}}/\mathcal{L}$. A remarkable property of Eq. (67) is that total power is proportional to the absolute temperature $T_c$ and, unlike the Carnot equation (63), is not related to any temperature difference. For cases when $\Delta T_C \ll T_s \approx T_c$, the two equations combined constrain $\Delta T_C$. Putting $W^{\mathrm{cond}} = W_C$ in Eqs. (67) and (63) we find

$$\Delta T_C = T_c \left(\frac{R T_s}{\mathcal{L}}\right)\frac{1 + \mathcal{H}_P/\mathcal{H}_v}{1 + F_S/F_{\mathcal{L}}} \approx 15 \text{ K}. \tag{68}$$

Here we used $F_{\mathcal{L}} = 85$ W m$^{-2}$, $F_S = 19$ W m$^{-2}$ (Ohmura and Raschke, 2005), $\mathcal{H}_P = 3.4$ km, see Appendix B, $\mathcal{H}_v = R T_c/(M_v g) = 12.7$ km, and $T_s = 288$ K as the global mean surface temperature.

This theoretical estimate of $\Delta T_C$ obtained under the assumption (64) that the circulation on Earth is condensation-driven coincides within 20% with an independent estimate of $\Delta T_c \equiv T_s - T_c = 18$ K between the surface temperature and the mean condensation temperature $T_c$, Eq. (B5). This consistency suggests that the condensation-driven circulation on Earth is equivalent to Carnot cycle operating between the surface temperature and the mean temperature $T_c = 270$ K where condensation occurs. This agrees with the observation that a major part of kinetic power is generated in the lower atmosphere, Fig. 5. The gravitational power of precipitation follows the vertical profile of the water vapor mixing ratio and is also maximum in the lower atmosphere, see Fig. 2 of Pauluis and Dias (2012) and Fig. 2 of Makarieva et al. (2013b).

The global kinetic power estimated from Eq. (64) using $T_c = 270$ K and $P = 0.96$ m yr$^{-1}$ is 3.8 W m$^{-2}$. This is consistent with the estimate we obtained in the previous section extrapolating $W_K$ estimated at different temporal scales $\tau$ to $\tau_c = 3$ h at which most convective motions should be resolved (Fig. 9).

## 6.2 The lower limit

Why does the atmosphere generate any appreciable power at all, i.e. what determines the lower limit of $W$? Atmospheric power $W$ can be viewed as a measure of the dynamic disequilibrium of the Earth's atmosphere. In equilibrium, for example, under conditions of hydrostatic and geostrophic or cyclostrophic balance, no power is generated: $W = 0$. There is no vertical air motion and practically no precipitation. There are no surface fluxes of sensible and latent heat. In global circulation models a non-zero rate of kinetic energy generation is achieved by introducing an *ad hoc* intensity of turbulent diffusion, which is chosen by fitting the model to observations. Turbulent diffusion determines the rate at which kinetic energy is dissipated (and, in the steady state, generated) (see also discussion in Makarieva et al., 2015b). It is this, and related parameterizations of dissipative processes, that postulate a certain non-zero value of $W$ in the terrestrial atmosphere and control its behavior. For example, putting turbulent diffusion in the atmospheric interior close to zero, Held and Hou (1980) described an otherwise realistic





general circulation in a dry atmosphere that was about an order of magnitude less intense than observed on Earth. We find no obvious grounds to expect that the atmospheric circulation on Earth could not be significantly weaker than it is today.

To what degree does atmospheric power depend on the Earth possessing a moist atmosphere? A moist atmosphere differs from a dry atmosphere by manifesting distinct processes that can generate air motion. One, the release of latent heat in the ascending air, has received much attention in studies of the atmospheric heat engine (e.g., Goody, 2003; Pauluis, 2011; Kleidon et al., 2014; Kieu, 2015). Whether latent heat release generates any positive atmospheric power is wholly dependent on the sufficiently rapid cooling of the descending air (Goody, 2003). Without such cooling, atmospheric power production from latent heat is impossible.

Condensation-induced dynamics introduces a distinct mechanism: any upward motion of a saturated air parcel results in condensation and precipitation which diminishes local surface pressure via a hydrostatic adjustment. This leads to air convergence towards the resulting low pressure. Irrespective of whether this initial air motion gets extinguished or sustains itself via persistent condensation of laterally imported water vapor, a certain amount of kinetic energy is generated. This condensation-related mechanism permits *self-induced* air motion. Condensation occurs and the condensation-related potential energy is released as the air rises in the gravitational field of Earth. This implies a positive feedback between the motion of moist air and the release of potential energy that sustains it. In a dry atmosphere such positive feedback is absent.

One can expect that this positive feedback will drive the atmosphere to a state when it will consume all available power such that condensation rate is maximized. Such an atmosphere will be circulating with its vertical velocity constrained by the absorbed solar power and the condition of maximum precipitation and minimum net radiative and sensible heat fluxes. On Earth, precipitation accounts for a major part of the solar power absorbed so this situation is realistic. In a dry atmosphere, with such mechanisms for self-induced air motions absent, atmospheric power can remain much lower.

Whether atmospheric power would be negligible on a dry Earth is a theoretical question. The parameterization of turbulence in current models is generally unrelated to the hydrological cycle (i.e. one and the same turbulent diffusion coefficient can be used in both dry and moist models). Therefore, comparing $W$ across current models with varying intensity of the hydrological cycle cannot clarify the role of water vapor. What is needed are theoretical insights that could be tested against observations. Direct tests are unfeasable – one cannot dry the Earth's atmosphere to see what happens. But one can compare kinetic power between circulation patterns that do not require condensation and those that do (e.g., anticyclones versus cyclones, Curry, 1987; Makarieva et al., 2015a) and see whether similar processes can clarify the magnitude of the global atmospheric power. One can also investigate circulation power on planets with or without intense phase transitions.

Furthermore, one could re-formulate dissipative processes in the existing global circulation models such that they conform to condensation-driven dynamics and see how they perform. Currently the parameterization of dissipative processes is governed by the requirement that the observed pressure gradients must yield the observed wind velocities. Within broad limits any model, dry or moist, can be parameterized to yield any desirable rate of wind power generation/dissipation. However, if indeed wind power is linked to condensation, then models that neglect this relationship – though they may be calibrated to replicate observed wind velocities – cannot predict circulation intensity, precipitation patterns and other related phenomena under changing climatic conditions (Bony et al., 2015).





## 7   Summary of main results

1. We defined global atmospheric power $W$ as the combined work per unit time of all atmospheric air parcels, Eq. (9), and examined four distinct expressions found in the literature, $W_I$ (1), $W_{II}$ (2), $W_{III}$ (3) and $W_{IV}$ (4), to determine which of them corresponds to $W$ in a moist atmosphere. We found that in the presence of phase transitions (as well as in their absence) $W = W_{III}$. Meanwhile, in the presence of phase transitions $W \neq W_{IV}$, such that $W_{IV}$ cannot be used to assess $W$ in a moist atmosphere (Section 2).

2. We showed that with a boundary condition $w_s = 0$, where $w_s$ is the vertical velocity of gaseous air at the surface, $W = W_{III} = W_I$:

$$W = W_{III} \equiv \frac{1}{\mathcal{S}} \int_{\mathcal{V}} p \nabla \cdot \mathbf{v} d\mathcal{V} = W_I \equiv -\frac{1}{\mathcal{S}} \int_{\mathcal{V}} \mathbf{v} \cdot \nabla p d\mathcal{V}. \qquad (69)$$

These equations are valid for a hydrostatic as well as a non-hydrostatic atmosphere and do not assume stationarity (Section 3.1). Importantly, $\mathbf{v}$ is the velocity of gaseous air alone and not the mean velocity of gas and condensate.

3. We showed that assuming $w_s \neq 0$ results in errors, whereby the estimated atmospheric power may exceed the incoming solar radiation. A mathematically and physically consistent approach requires putting $w_s = 0$ and formulating evaporation at $z = 0$ as a point source of water vapor (Dirac's delta function) and not as a vertical flux of water vapor $\rho_{vs} w_s$ with $w_s \neq 0$ (Section 3.1).

4. Using $W = W_I$, the continuity equations (17) and (18) and the equations of motions (32) for gaseous air with an explicitly specified interaction (31) between gaseous air and condensate particles, we formulated the steady-state global atmospheric power budget. This budget is a sum of three terms: kinetic energy production by horizontal pressure gradients $W_K = W_{II}$ (38), the gravitational power of precipitation $W_P$ (39) and condensate loading $W_c$ (40). This three term formulation is valid with an accuracy of $(w/u)^2$, where $w$ and $u$ are the vertical and horizontal air velocities at the resolved scale (Section 3.4). At a horizontal scale of about 100 km the condensate loading term makes only a minor contribution to $W$ of the order of one per cent (Fig. 1).

5. We compared our results to the formulation of the atmospheric power budget provided by Pauluis et al. (2000) (Section 3.5 and Fig. 2). We showed that Pauluis et al. (2000) obtained a valid expression for precipitation-related dissipation $W_P^* = W_P + W_c$ (44). They also offered an expression for total atmospheric power $W^*$ (48) in a model based on an anelastic approximation. We showed that to be valid this expression requires two clarifications: $\dot{K} = 0$ and that $w$ in Eq. (48) is the vertical velocity of gaseous air (not including condensate), which obeys the continuity equation (17).

   Since they assumed $w_s \neq 0$, Pauluis et al. (2000) could neither derive a generally valid relationship $W = W_I$ nor establish how their formulations relate to the equations of motion and continuity. Furthermore, Pauluis et al. (2000) appear to have misinterpreted mean velocity $\mathbf{v}_m$ of gaseous air and condensate for velocity $\mathbf{v}$ of gaseous air in the model of Lipps and Hemler (1982) which Pauluis et al. (2000) used to numerically evaluate their formulations.





6. We showed that the same two factors, an incorrect boundary condition $w_s \neq 0$ and confusion between $\mathbf{v}$ and $\mathbf{v}_m$ in the expression for atmospheric power, may explain the omission of a major term from the atmospheric power budget in the analysis of Laliberté et al. (2015) (Section 4). This omission of the enthalpy integral (57) is crucial for their approach, where the atmospheric power is constrained from the first law of thermodynamics based on the works of Pauluis et al. (2000), Pauluis and Held (2002) and Pauluis (2011). The theoretical representation of atmospheric power as a sum of only two terms, one related to entropy and another to chemical potential, Eq. (55), as well as the quantitative estimates of atmospheric power and its trends resulting from this representation, appear invalid.

7. Our formulation for the atmospheric power budget, Eq. (37), reveals that the gravitational power of precipitation $W_P$ can be estimated as $W_P \approx W - W_K$ from the known atmospheric pressure gradient and air velocity – without knowing atmospheric moisture content, local condensation rate, evaporation or precipitation. This formulation also highlights that while kinetic power $W_K$ depends on horizontal velocities, $W_P$ and $W_c$ and, hence, total power $W = W_P + W_c + W_K$ (37), depend on vertical velocity; thus, observation-based estimates of $W$, $W_c$ and $W_P$ should be less accurate than estimates of $W_K$ (Section 5.1).

8. We used daily and monthly mean MERRA and NCAR/NCEP data and 3-hourly instantaneous MERRA data for 1979-2015 to estimate the atmospheric power budget using the obtained formulations (Section 5.1). We found that while kinetic power $W_K$ is relatively robust among the datasets, the estimates of $W$ and $W_P$ differ: they are positive in the MERRA re-analysis and negative in the NCAR/NCEP re-analysis. We discussed how these differences reflect inherent uncertainty in vertical velocities derived from the continuity equations in re-analyses (Section 5.2). Unlike NCAR/NCEP, the correction procedure used to retrieve vertical velocities from the continuity equations in MERRA incorporates some information about the water cycle, which may explain the more realistic $W_P$ values.

   Even in the MERRA re-analysis the gravitational power of precipitation $W_P$, which should positively correlate with global precipitation $P$, see Eq. (39), does not do so on either seasonal (Fig. 4) or multi-year scale. While global precipitation in MERRA has increased from 1979 to 2015 (Kang and Ahn, 2015), $W_P$ has declined by a similar margin (Fig. 7). These apparent inconsistencies highlight a need for a systematic study of the atmospheric power budget in re-analyses.

9. We discussed how the representation of atmospheric energetics in the re-analyses can be improved with use of independent, precipitation-based estimates of $W_P$: these will help constrain vertical velocities via Eqs. (37)-(40). We obtained such estimates using the observed value of global precipitation, TRMM-derived estimates for tropical $W_P$ obtained by Pauluis and Dias (2012) and theoretical estimates of precipitation path-length based on the approach of Makarieva et al. (2013b) (Appendix B). The global gravitational power of precipitation was estimated at $W_P = 1\ \mathrm{W\ m^{-2}}$, which is 20% higher than the original estimate by Makarieva et al. (2013b).

10. Our formulation for the atmospheric power budget, Eqs. (37)-(40), highlights that the magnitude of atmospheric power $W$ and its components is scale-specific. We illustrated this scale dependence for kinetic power $W_K$: with the temporal resolution of the dataset increasing from one month to one day, $W_K$ in MERRA and NCAR/NCEP rise approximately



fivefold (Section 5.3 and Fig. 8). At the finest resolution (instantaneous MERRA data) total atmospheric power equals
$$W \approx W_K + W_P = 3.3 \text{ W m}^{-2}.$$

11. Extrapolation of the observed dependencies of $W_K$ on time scale in the MERRA and NCAR/NCEP re-analysis to the convective time scale reveals that at this scale $W_K$ can be close to the theoretical prediction of condensation-induced dynamics: $W_K^{\text{cond}} = \Pi R T_c = 3.8$ W m$^{-2}$ (64), where $\Pi = P/M_v$ is precipitation, $T_c$ is the mean temperature where condensation occurs and $R$ is the universal gas constant (Section 6). Further analyses are required to estimate $W_K$ at the convective scale more reliably. At the finest resolution the MERRA-derived $W_K = 2.5$ W m$^{-2}$ is 50% less than the theoretical estimate $W_K^{\text{cond}}$.

12. In agreement with the theoretical prediction $W_K = W_K^{\text{cond}} = \Pi R T_c$, we found that seasonal variability of the global kinetic power $W_K$ is close in its magnitude and behavior to the variability of the global precipitation $P$: both variables change from month to month by a few per cent and reach their minimal values in spring and autumn (Fig. 4). We also show that most kinetic power (59%) is generated in the lower atmosphere in the layer up to 800 hPa.

13. We demonstrated that an atmosphere where $W^{\text{cond}} = W_K^{\text{cond}} + W_P$ corresponds to a Carnot cycle with a temperature difference $\Delta T_C \approx 15$ K. We showed that this temperature difference is close to the independent estimate of the mean temperature difference $\Delta T_c \equiv T_s - T_c \approx 18$ K between the ground surface and the mean height $\mathcal{H}_P$ where condensation occurs (Section 6.1).

14. We showed that condensation-induced dynamics can explain the magnitude of observed wind power (Section 6). This suggests that determinants of atmospheric water can have a major influence. Deforestation disrupts terrestrial evaporation and resulting condensation and diminishes the soil moisture store. One possible effect is changed partitioning between small-scale atmospheric power generated on the scale of convective eddies and larger-scale atmospheric power generated at continental scale. This would influence circulation patterns and resulting rainfall. Such mechanisms may contribute to the changes in rainfall already noted in various regions, e.g., Brazil and the Mediterranean (e.g., Marengo and Espinoza, 2016; Dobrovolski and Rattis, 2015; Cook et al., 2016); they may also explain the phenomenon of self-perpetuating droughts in the continental interior investigated in recent modelling studies (e.g., Koster et al., 2016). We urge increased attention to the dynamic effects of condensation.

## Appendix A: Deriving $W$ (9) from $W_a$ (7) for ideal gas

The equation of state for ideal gas is

$$pV = RT, \text{ or } p = NRT. \tag{A1}$$

Here $T$ is temperature, $N \equiv V^{-1}$ is air molar density (mol m$^{-3}$), $V$ is the atmospheric volume occupied by one mole of air, $p$ is air pressure and $R = 8.3$ J mol$^{-1}$ K$^{-1}$ is the universal gas constant.





Using (A1) and taking into account the following relationships,

$$
\quad \tilde{V} = \tilde{N}V, \quad \frac{1}{N}\frac{dN}{dt} = -\frac{1}{V}\frac{dV}{dt}, \quad p\frac{\tilde{N}}{\tilde{V}}\frac{dV}{dt} = p\frac{1}{V}\frac{dV}{dt} = -pV\frac{dN}{dt}, \tag{A2}
$$

where $\tilde{N}$ is the number of moles of gas within volume $\tilde{V}$, we can write $W_a$ (7) as

$$
W_a = \frac{p}{\tilde{V}}\frac{d\tilde{V}}{dt} = \frac{p}{\tilde{V}}\left(\tilde{N}\frac{dV}{dt} + V\frac{d\tilde{N}}{dt}\right) = RT\left(-\frac{dN}{dt} + \frac{1}{\tilde{V}}\frac{d\tilde{N}}{dt}\right). \tag{A3}
$$

The number of molecules (moles) $\tilde{N}$ in each air parcel can only change via an inflow (outflow) of molecules through the parcel's boundary. This change results from either diffusion of molecules between the adjacent parcels or from phase transitions or from both. Since in the case of diffusion any molecule leaving one parcel, $d\tilde{N}_1/dt < 0$, arrives to some other parcel, $d\tilde{N}_2/dt = -d\tilde{N}_1/dt > 0$, all the diffusion terms cancel in the global sum of the last term in Eq. (A3) over all parcels. What remains corresponds to phase transitions:

$$
\sum_{i=1}^{n}\frac{d\tilde{N}_i}{dt} = \int_{\mathcal{V}}\frac{1}{\tilde{V}}\frac{d\tilde{N}}{dt}d\mathcal{V} = \int_{\mathcal{V}}\dot{N}d\mathcal{V}, \tag{A4}
$$

where $\dot{N}$ is the molar rate of phase transitions per unit volume (mol m$^{-3}$ s$^{-1}$). Its integral over volume $\mathcal{V}$ is equal to the total rate of phase transitions in all the $n$ air parcels. By virtue of the conservation relationship (A4) $\dot{N}$ includes the inflow (outflow) into all the air parcels from all liquid or solid surfaces (droplet surface in the atmospheric interior or the Earth's surface).

Using Eqs. (A3) and (A4) we can write total power $W$ of the $n$ air parcels composing the atmosphere as

$$
W \equiv \frac{1}{\mathcal{S}}\int_{\mathcal{V}}W_a d\mathcal{V} = \frac{1}{\mathcal{S}}\int_{\mathcal{V}}RT\left(\dot{N} - \frac{dN}{dt}\right)d\mathcal{V}. \tag{A5}
$$

Here $dN/dt \equiv \partial N/\partial t + \mathbf{v}\cdot\nabla N$ is the material derivative of $N$ with $\mathbf{v}$ being the gas velocity. The term in braces in Eq. (A5) is based on the equation of state (A1).

Using the continuity equation $\dot{N} = \partial N/\partial t + \nabla\cdot(N\mathbf{v})$ we have

$$
\dot{N} - \frac{dN}{dt} \equiv \dot{N} - \frac{\partial N}{\partial t} - \mathbf{v}\cdot\nabla N = N\nabla\cdot\mathbf{v}. \tag{A6}
$$

Multiplying Eq. (A6) by $RT$ and noting Eq. (A1), we find that Eq. (A5) turns into Eq. (9).

The physical meaning of Eq. (A5) becomes clear from consideration of an atmosphere that is motionless on a large scale, such that $\mathbf{v} = 0$ and $\nabla\cdot\mathbf{v} = 0$. Then condensation that occurs instantaneously on a smaller scale is described by the source term $\dot{N} < 0$ that represents the large-scale mean. The compensatory expansion of the adjacent air is described by the large-scale mean $\partial N/\partial t < 0$ showing that the molar concentration of air diminishes. As is clear from Eqs. (A5) and (A6), since $\dot{N} - \partial N/\partial t = 0$, no resulting work is performed on the considered scale: $W = 0$.

We also note that using Eq. (A1) we can write $W_a$ (7) as

$$
\quad W_a = -\frac{dp}{dt} + RN\frac{dT}{dt} + \frac{RT}{\tilde{V}}\frac{d\tilde{N}}{dt} = -\frac{dp}{dt} + RN\frac{dT}{dt} + RT\dot{N}. \tag{A7}
$$



By analogy with Eq. (25), for any $X$ in the view of the continuity equation (A6), definition of material derivative (5) and the boundary conditions (13) and (14) we have $\int_{\mathcal{V}} N(dX/dt)d\mathcal{V} = \int_{\mathcal{V}} \left( \partial(XN)/\partial t - X\dot{N} \right) d\mathcal{V}$. Thus, from Eq. (A7) with $X = T$ using Eq. (A1) we find that $W \equiv (1/\mathcal{S}) \int_{\mathcal{V}} W_a d\mathcal{V} = W_I$ (1).

### Appendix B: Estimating the enthalpy integral (60), $W_P$ (39) and $T_c$ (66)

We follow the approach of Makarieva et al. (2013b). We assume that moist air having temperature $T_s$ and relative humidity $80\%$ at the surface first rises dry adiabatically up to height $z_1$ where water vapor becomes saturated. Then it rises moist adiabatically to $z_2$, where condensation ceases. At $z_2$ the air preserves share $\zeta$ of its initial water vapor content, $\zeta \equiv \gamma(z_2)/\gamma_s = \gamma(z_2)/\gamma(z_1)$. Here $\gamma \equiv p_v/p$, where $p_v$ is water vapor partial pressure and $p$ is air pressure. Moist adiabatic distributions of $\gamma(z)$, $T(z)$ and $p(z)$ with $p_s = 1000$ hPa were calculated according to Eqs. (A3)-(A5) of Makarieva et al. (2013b).

For $T_s$ ranging from 260 to 310 K and for $\zeta$ ranging from 0.001 (complete condensation) to $3/4$, we estimated mean condensation height $\mathcal{H}_P$, mean condensation temperature $T_c$ and mean enthalpy per mole $h_c$, Fig. 10 and Table 1:

$$\mathcal{H}_P(T_s,\zeta) = \frac{1}{\gamma(z_2) - \gamma(z_1)} \int\limits_{z_1}^{z_2} z\frac{\partial\gamma}{\partial z}dz, \tag{B1}$$

$$T_c(T_s,\zeta) = \frac{1}{\gamma(z_2) - \gamma(z_1)} \int\limits_{z_1}^{z_2} T(z)\frac{\partial\gamma}{\partial z}dz, \qquad \Delta T_c \equiv T_s - T_c, \tag{B2}$$

$$h_c(T_s,\zeta) = \frac{1}{\gamma(z_2) - \gamma(z_1)} \int\limits_{z_1}^{z_2} h(z)\frac{\partial\gamma}{\partial z}dz, \qquad \Delta h_c \equiv h_s - h_c, \qquad h(z) = c_pT(z) + \mathcal{L}\gamma(z), \qquad c_p = (7/2)R. \tag{B3}$$

To find the corresponding global mean values we consider the tropics (the area between 30°S and 30°N) and the extratropics separately. The two regions have equal areas. The mean annual temperatures for 2009-2014 at 1000 hPa are 296.5 K and 277 K for the tropics and the extratropics, respectively. Most tropical rainfall is associated with temperatures above 299 K (Johnson and Xie, 2010; Sabin et al., 2013), so we take $T_{s\ tr} = 300$ K as a representative value for tropical rainfall. In the extratropics there is also a tendency for higher rainfall at higher temperature, Fig. 11, we take $T_{s\ ex} = 280$ K. According to the Global Precipitation Climatology Project (GPCP) version 2.2 dataset, tropical and extratropical precipitation in 2009-2014 was, respectively, $P_{tr} = 3.03$ mm d$^{-1}$ and $P_{ex} = 2.24$ mm d$^{-1}$. We estimated mean global values of $\mathcal{H}_P$, $\Delta T_c$ and $\Delta h_c$ from Eqs. (B1)-(B3) as

$$X(\zeta) = \frac{X(T_{s\ tr},\zeta)P_{tr} + X(T_{s\ ex},\zeta)P_{ex}}{P_{tr} + P_{ex}}. \tag{B4}$$

The results are shown in Table 1.

Assuming that tropical $W_{P\ tr} = 1.5$ W m$^{-2}$ according to the TRMM measurements analyzed by Pauluis and Dias (2012), we conclude from Table 1 that under our assumptions the results with $\zeta > 0.4$ (more than 40% of water vapor does not condense) corresponding to the global mean $W_P < 0.75$ W m$^{-2}$ are not realistic. This is smaller than half the tropical average and is thus impossible. For the same reason our previous estimate $W_P = 0.8$ W m$^{-2}$, which corresponds to a negligible contribution from





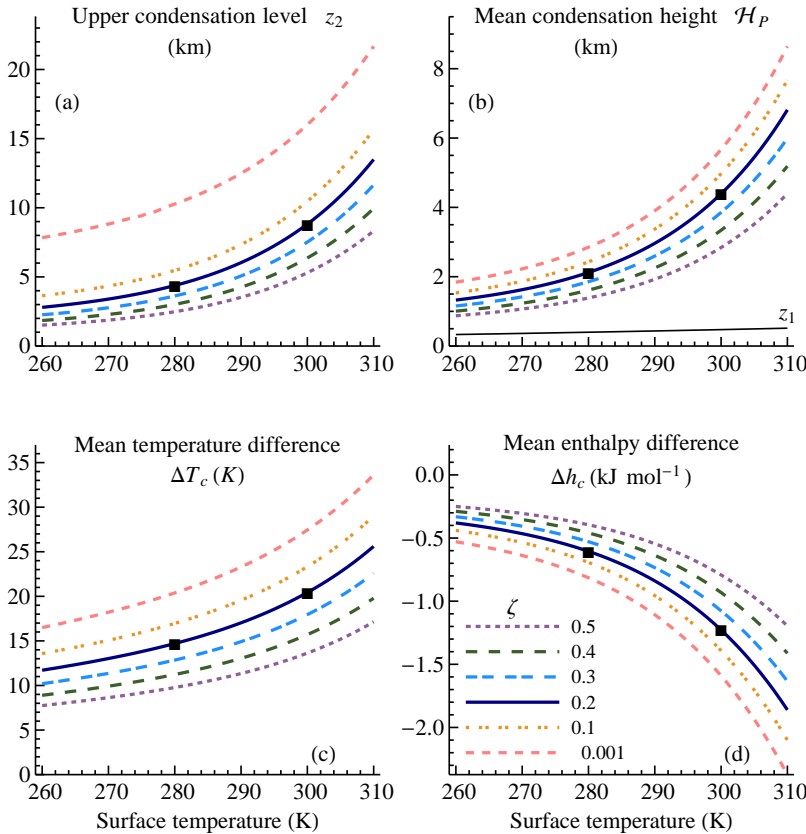

**Figure 10.** The upper condensation level $z_2$ (a), lower condensation level $z_1$ and $\mathcal{H}_P$ (B1) (b), $\Delta T_c$ (B2) (c) and $\Delta h_c$ (B3) (d) as dependent on surface temperature $T_s$ and incompleteness of condensation $\zeta$. Solid squares show values used for the global mean estimates (B5).

the extratropical rainfall to total gravitational power, appears an underestimate.[7] The tropical estimate $W_{P\,tr}$ coincides with the TRMM-derived estimate of Pauluis and Dias (2012) for $\zeta = 0.2$. In this case $W_P = 1$ W m$^{-2}$. We will thus use the case $\zeta = 0.2$ as a representative value for the global mean, Table 1:

$$\zeta = 0.2, \quad \Delta T_c = 18 \text{ K}, \quad I_h = -P\Delta h_c = -1.6 \text{ W m}^{-2}, \quad W_P = 1 \text{ W m}^{-2}. \tag{B5}$$

Note that in the interval $0 \leq \zeta \leq 0.3$ all the values in Table 1 change about 1.3-fold, which suggests that the uncertainty of the global values should be under 30%.

---

[7]The estimate of $W_P = 0.8$ W m$^{-2}$ was obtained by Makarieva et al. (2013b) from Eq. (39) assuming that $\mathcal{H}_P = 2.5$ km is a representative value for the global average. This height corresponds to the following case: at the surface the ascending air has a global mean surface temperature 288 K and relative humidity 80%; above the point of saturation it rises with mean tropospheric lapse rate 6.5 K km$^{-1}$; about one quarter of the water vapor does not condense and remains in the ascending air, i.e. $\zeta = 1/4$.



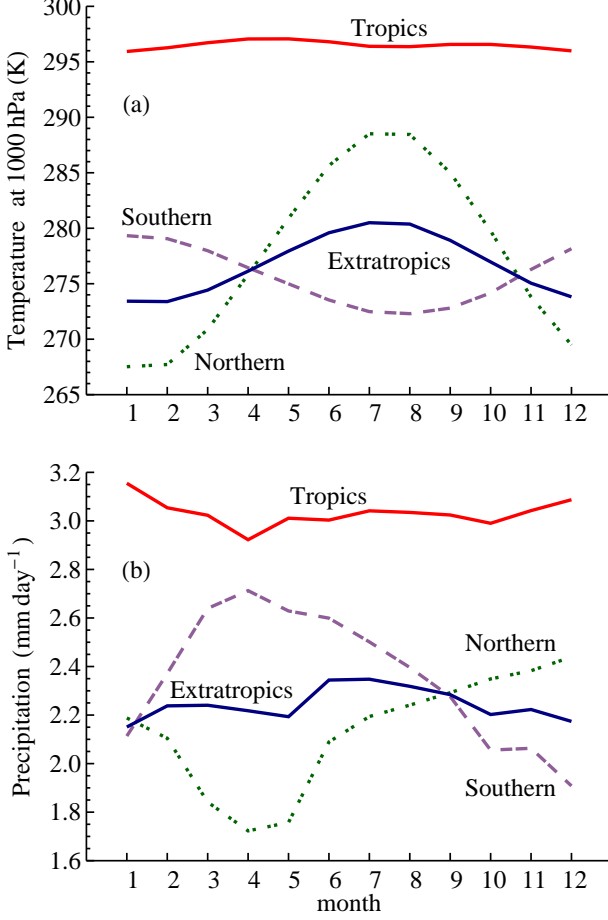

**Figure 11.** Mean monthly temperature at 1000 hPa (a) and GPCP v. 2.2 precipitation (b) in 2009-2014 in the tropics ($30°$S - $30°$N) and the extratropics ($90°$S - $30°$S and $30°$N - $90°$N).

In particular, the bottomline for $W_P$ is provided by the TRMM-derived estimate of Pauluis and Dias (2012), which is 1.5 W m$^{-2}$ for the area between $30°$ N and $30°$ S. So, global $W_P$ cannot be lower than 0.75 W m$^{-2}$. If it is 0.75 W m$^{-2}$, this means that there is no precipitation at all in the extratropics. However, since extratropical precipitation is significant (2.3 mm d$^{-1}$ versus 3.1 mm d$^{-1}$ in the tropics (Fig. 11), it will contribute to the global value of $W_P$. Even we assume that all extratropical rainfall precipitates from $\mathcal{H}_P = 1$ km (which is clearly an underestimate), global $W_P$ will constitute 1000    0.87 W m$^{-2}$. Therefore, the uncertainty of the lower limit of our estimate $W_P = 1$ W m$^{-2}$ is about 10%. If all rainfall in the extratropics precipitates from the same mean height as in the tropics (which is clearly an overestimate), then we would have $W_P = 1.3$ W m$^{-2}$.





**Table 1.** Global mean estimates of the mean condensation height $\mathcal{H}_P$, mean condensation temperature difference $\Delta T_c$ and enthalpy difference $\Delta h_c$ (B4) and global gravitational power of precipitation $W_P$ dependent on $\zeta$ (incompleteness of condensation). Subscripts "$tr$" and "$ex$" refer to corresponding values in the tropics and extratropics. Boldfaced line shows values for $\zeta = 0.2$ which are consistent with TRMM-derived estimate for tropical $W_{P\ tr} = 1.5$ W m$^{-2}$ obtained by Pauluis and Dias (2012). Note that $P = (P_{tr} + P_{ex})/2$ and $W_P = (W_{P\ tr} + W_{P\ ex})/2$.

| $\zeta$ | $\mathcal{H}_{P\ tr}$ km | $\mathcal{H}_{P\ ex}$ km | $\mathcal{H}_P$ km | $\Delta T_{c\ tr}$ K | $\Delta T_{c\ ex}$ K | $\Delta T_c$ K | $-P_{tr}\Delta h_{c\ tr}$ W m$^{-2}$ | $-P_{ex}\Delta h_{c\ ex}$ W m$^{-2}$ | $-P\Delta h_c$ W m$^{-2}$ | $W_{P\ tr}$ W m$^{-2}$ | $W_{P\ ex}$ W m$^{-2}$ | $W_P$ W m$^{-2}$ |
|---|---|---|---|---|---|---|---|---|---|---|---|---|
| 0.00 | 5.7 | 2.9 | 4.5 | 27.4 | 20.4 | 24.4 | −3.10 | −1.17 | −2.14 | 1.95 | 0.72 | 1.34 |
| 0.10 | 5.0 | 2.4 | 3.9 | 23.3 | 17.0 | 20.6 | −2.71 | −1.00 | −1.85 | 1.71 | 0.62 | 1.16 |
| **0.20** | **4.4** | **2.1** | **3.4** | **20.4** | **14.7** | **18.0** | **−2.39** | **−0.87** | **−1.63** | **1.51** | **0.54** | **1.02** |
| 0.25 | 4.1 | 2.0 | 3.2 | 19.2 | 13.7 | 16.9 | −2.24 | −0.82 | −1.53 | 1.42 | 0.50 | 0.96 |
| 0.30 | 3.9 | 1.9 | 3.0 | 17.9 | 12.8 | 15.8 | −2.10 | −0.76 | −1.43 | 1.32 | 0.47 | 0.90 |
| 0.40 | 3.3 | 1.6 | 2.6 | 15.7 | 11.2 | 13.8 | −1.82 | −0.66 | −1.24 | 1.15 | 0.41 | 0.78 |
| 0.50 | 2.8 | 1.4 | 2.2 | 13.6 | 9.8 | 12.0 | −1.54 | −0.57 | −1.06 | 0.98 | 0.35 | 0.66 |
| 0.60 | 2.4 | 1.2 | 1.9 | 11.7 | 8.4 | 10.3 | −1.28 | −0.48 | −0.88 | 0.81 | 0.30 | 0.55 |
| 0.70 | 1.9 | 1.0 | 1.5 | 9.8 | 7.2 | 8.7 | −1.02 | −0.40 | −0.71 | 0.64 | 0.25 | 0.45 |
| 0.75 | 1.6 | 0.9 | 1.3 | 8.9 | 6.6 | 7.9 | −0.89 | −0.36 | −0.62 | 0.56 | 0.22 | 0.39 |

## Appendix C: Details of calculating $W$ and $W_K$

The MERRA dataset MAI3CPASM version 5.2.0 was downloaded for the years 1979-2015 (it contains one file for each day) from *http://mirador.gsfc.nasa.gov*. We chose this dataset because it contained the pressure velocity $\omega$ necessary for calculating total atmospheric power. The data are provided for eight times of the day ($t = 1, ..., 8$): 00, 03, 06, 09, 12, 15, 18 and 21 hours. The latitude/longitude grid has a resolution of $1.25°$. Latitude coordinate of the grid cell center spans from $-90 + 1.25/2$ to $90 - 1.25/2$ degrees Northern latitude ($i = 1, ..., 144$). Longitude coordinate of the grid cell center spans from $-180 + 1.25/2$ to $180 - 1.25/2$ degrees Eastern longitude ($j = 1, ..., 288$). The vertical dimension is represented by 42 fixed pressure levels ($k = 1, ..., 42$), from $p_1 = 1000$ hPa to $p_{42} = 0.1$ hPa (1000, 975, 950, 925, 900, 875, 850, 825, 800, 775, 750, 725, 700, 650, 600, 550, 500, 450, 400, 350, 300, 250, 200, 150, 100, 70, 50, 40, 30, 20, 10, 7, 5, 4, 3, 2, 1, 0.7, 0.5, 0.4, 0.3, 0.1 hPa). For each day in the studied years we used the following variables $X_k(t, i, j)$: geopotential height $H$, meridional and zonal velocity $v$ and $u$, pressure velocity omega $\omega$, temperature $T$ and the mass fraction of water vapor $q_v \equiv \rho_v/\rho$. We also used surface pressure $p_s(t, i, j)$. To calculate $\partial X/\partial t$ for time $t$ in a given grid cell we used the next (i.e. 3 hours after the considered time point) and the previous (3 hours before) $X$ values and divided their difference by $\Delta t = 6$ hr.





For each day, the daily averaged values of all variables were obtained using the "perform mean on daily file" option while downloading MAI3CPASM data from *http://disc.sci.gsfc.nasa.gov/daac-bin/FTPSubset.pl?LOOKUPID_List=MAI3CPASM*. Monthly averaged values are from the MERRA dataset MAIMCPASM version 5.2.0. These data have the same spatial resolution as MAI3CPASM.

NCAR/NCEP daily mean and monthly mean variables were downloaded from *http://www.esrl.noaa.gov/psd/data/gridded/data.ncep.reanalysis.pressure.html* and *http://www.esrl.noaa.gov/psd/data/gridded/data.ncep.reanalysis.surface.html*. NCAR/NCEP data have a resolution of $2.5°$ and 12 pressure levels for $\omega$ (1000, 925, 850, 700, 600, 500, 400, 300, 250, 200, 150, 100 hPa). The atmospheric power was calculated for these levels assuming that $q_v = 0$ for $p < 300$ hPa (because the upper pressure level for $q_v$ is 300 hPa). How

the integration of Eqs. (37) and (38) was performed is illustrated below on the example of MAI3CPASM dataset. A similar procedure was used for all the data.

### C1    Calculation of $W$ in MERRA MAI3CPASM dataset

The procedure is best illustrated using an example. For example, we are interested in the time point 15.00 ($t = 6$) on 1 July 2010 for a grid cell with numbers $i = 80$ (latitude) and $j = 100$ (longitude). This grid cell is centered at $9.375°$ Northern latitude and

$-55.625°$ Eastern longitude and has an area of $S(i) = 1.906 \times 10^{10}$ m$^2$. The atmospheric column is composed of elementary volumes $\Delta \mathcal{V}_k$ that are enclosed by the neighboring pressure levels $k$ and $k+1$:

$$\Delta \mathcal{V}_k(t,i,j) = S(i)[H_{k+1}(t,i,j) - H_k(t,i,j)], \tag{C1}$$

where $H_k(t,i,j)$ is the geopotential height of the $k$-th pressure level. For example, for $k = 1$ (pressure level $p_1 = 1000$ hPa) we have in the time and place of interest $H_1 = 125$ m, $H_2 = 348$ m and $\Delta \mathcal{V}_1 = 4.25 \times 10^{12}$ m$^3$. The omega value corresponding

to each elementary volume $\Delta \mathcal{V}_k$ (C1) was calculated as $\omega = (\omega_k + \omega_{k+1})/2$, i.e. as the average of the omega values at the neighboring pressure levels defining the elementary volume.

We also need to calculate the contribution of the near surface layer that is enclosed between the pressure levels $p_s$ (surface pressure) and $p_{k_{min}}$, where $k_{min}(t,i,j)$ is the number of the level with maximum pressure for which the data exists for a given grid cell and time point. For example, in mountainous areas there are no atmospheric layers with $k = 1$ and $p_1 = 1000$ hPa: in

such areas $k_{min} > 1$.

To find the vertical thickness $\Delta z$ of the surface layer we used the hydrostatic equation $\partial p / \partial z = -p/\mathcal{H}$, where $\mathcal{H} = RT/Mg$ is the local exponential scale height for air pressure. We estimated the elementary volume $\Delta \mathcal{V}_s(t,i,j)$ in the surface layer as

$$\Delta \mathcal{V}_s = S(i)\Delta z = S(i)\frac{p_s - p_{k_{min}}}{p_{k_{min}}}\mathcal{H}_{k_{min}}, \quad \mathcal{H}_k \equiv \frac{RT_k}{M_k g}, \quad M_k \equiv \frac{M_d}{1 + (M_d/M_v - 1)q_{vk}}. \tag{C2}$$

Here $M_d = 0.0289$ kg mol$^{-1}$ and $M_v = 0.018$ kg mol$^{-1}$ are molar masses of dry air and water vapor, respectively. For our cell

($t = 6$, $i = 80$, $j = 100$) we have $k_{min} = 1$, $p_{k_{min}} = p_1 = 10^5$ Pa, $p_s = 101409$ Pa, $q_{v1} = 0.0179$, $M_1 = 0.0286$ kg mol $^{-1}$, $T_1 = 299.2$ K and $\Delta \mathcal{V}_s = 2.38 \times 10^{12}$ m$^3$.



We now need to find the omega value at the surface $\omega_s$. (While there are surface data in the MERRA database, they are provided with a different spatial resolution than in MAI3CPASM.) This can be done in two ways, which should give identical results in the limit of infinitely small elementary volumes, $\Delta\mathcal{V}_k \to 0$, but different results for finite $\Delta\mathcal{V}_k$. The first way is to assume that at the surface wind velocity is zero, such that $\nabla \cdot \mathbf{v} = 0$ and omega is by definition equal to surface pressure tendency, see Eq. (61):

$$\omega_s = \frac{\partial p_s}{\partial t}. \tag{C3}$$

The second way is to extrapolate the omega dependence on pressure linearly to the surface assuming that the derivative of omega over pressure does not change from the surface to the $(k_{min} + 1)$-th layer:

$$\frac{\omega_s - \omega_{k_{min}}}{p_s - p_{k_{min}}} = \frac{\omega_{k_{min}+1} - \omega_{k_{min}}}{p_{k_{min}+1} - p_{k_{min}}}. \tag{C4}$$

The obtained results differ by about 10%, see Section C3 below.

Finally, the integral of omega over the atmospheric column in each grid cell is given by

$$\Delta\mathcal{V}_s(t,i,j)\frac{\omega_s(t,i,j) + \omega_{k_{min}}(t,i,j)}{2} + \sum_{k=k_{min}}^{41} \Delta\mathcal{V}_k(t,i,j)\frac{\omega_k(t,i,j) + \omega_{k+1}(t,i,j)}{2}. \tag{C5}$$

The global integral of omega for a given time point $t$ was found as the sum of Eq. (C5) over all grid cells

$$\Omega(t) \equiv -\frac{1}{\mathcal{S}}\int_\mathcal{V} \omega(t,z,y,x)d\mathcal{V} \equiv -\frac{1}{\mathcal{S}}\sum_{i=1}^{144}\sum_{j=1}^{288}\left(\Delta\mathcal{V}_s\frac{\omega_s + \omega_{k_{min}}}{2} + \sum_{k=k_{min}}^{41}\Delta\mathcal{V}_k\frac{\omega_k + \omega_{k+1}}{2}\right). \tag{C6}$$

with with $\omega_s$ estimated from either Eq. (C3) or Eq. (C4), Table 2.

To calculate time derivative $\partial p/\partial t$ at geopotential height $H_k$ corresponding to the $k$-th pressure level, we used the hydrostatic equation in the form

$$\left(\frac{\partial p}{\partial t}\right)_{H_k} = -\frac{\partial H_k}{\partial t}\frac{\partial p}{\partial z} = \frac{\partial H_k}{\partial t}\frac{p}{\mathcal{H}_k}. \tag{C7}$$

For the global integral we have similar to Eq. (C6):

$$\Psi \equiv \frac{1}{\mathcal{S}}\int_\mathcal{V}\frac{\partial p}{\partial t}d\mathcal{V} = \frac{1}{\mathcal{S}}\sum_{i=1}^{144}\sum_{j=1}^{288}\left\{\frac{\Delta\mathcal{V}_s}{2}\left[\frac{\partial p_s}{\partial t} + \left(\frac{\partial p}{\partial t}\right)_{H_{k_{min}}}\right] + \sum_{k=k_{min}}^{41}\frac{\Delta\mathcal{V}_k}{2}\left[\left(\frac{\partial p}{\partial t}\right)_{H_k} + \left(\frac{\partial p}{\partial t}\right)_{H_{k+1}}\right]\right\}. \tag{C8}$$

## C2 Calculation of $W_K$ in MERRA MAI3CPASM dataset

We calculated zonal and meridional pressure gradients at pressure level $k$ as follows:

$$\left(\frac{\partial p}{\partial x}\right)_k = \left(\frac{\partial p}{\partial z}\right)_k\frac{\partial H_k}{\partial x} = \frac{p_k}{\mathcal{H}_k}\frac{\partial H_k}{\partial x}, \quad \left(\frac{\partial p}{\partial y}\right)_k = \frac{p_k}{\mathcal{H}_k}\frac{\partial H_k}{\partial y}, \tag{C9}$$

$$\frac{\partial H_k(t,i,j)}{\partial x} = \frac{H_k(t,i,j+1) - H_k(t,i,j-1)}{2 \times 1.25 \times L(i)}, \quad \frac{\partial H_k(t,i,j)}{\partial y} = \frac{H_k(t,i+1,j) - H_k(t,i-1,j)}{2 \times 1.25 \times L_p}, \tag{C10}$$





where $L(i)$ is the length of 1 degree arc along the parallel at the corresponding latitude, $L_p = 111.127$ m is the length of one degree arch along the meridian.

Kinetic energy generation $K_k$ per unit volume (W m$^{-3}$) at pressure level $k$ is calculated from (C9) and (C10)

$$K_k(t,i,j) \equiv -u_k(t,i,j)\left(\frac{\partial p}{\partial x}\right)_k - v_k(t,i,j)\left(\frac{\partial p}{\partial y}\right)_k. \tag{C11}$$

The value of $K_s$ at the surface is found in two ways, one by analogy with Eq. (C3) assuming that at the surface $\mathbf{v} = 0$ and

$$K_s = 0 \tag{C12}$$

and second by analogy with Eq. (C4):

$$\frac{K_s - K_{k_{min}}}{p_s - p_{k_{min}}} = \frac{K_{k_{min}+1} - K_{k_{min}}}{p_{k_{min}+1} - p_{k_{min}}}. \tag{C13}$$

For the global integral $W_K$ for a given time point $t$ we have

$$W_K(t) \equiv -\frac{1}{\mathcal{S}}\int_{\mathcal{V}} \mathbf{u}\cdot\nabla p\,d\mathcal{V} \equiv \frac{1}{\mathcal{S}}\sum_{i=2}^{143}\sum_{j=1}^{288}\left(\Delta\mathcal{V}_s\frac{K_s(t,i,j)+K_{k_{min}}(t,i,j)}{2} + \sum_{k=k_{min}}^{41}\Delta\mathcal{V}_k\frac{K_k(t,i,j)+K_{k+1}(t,i,j)}{2}\right) \tag{C14}$$

with $K_s$ estimated from either Eq. (C12) or Eq. (C13), Table 2.

### C3 Two ways of estimating surface values of $\omega$ and $\mathbf{u}\cdot\nabla p$

Attention to the boundary layer is justified by the fact that here the horizontal velocity experiences non-uniform vertical changes. The surface layer averages about 13 hPa higher pressure than $p_1 = 1000$ hPa, the pressure of the first layer in the MERRA and NCAR/NCEP database. This difference corresponds to an atmospheric layer about $H_1 \sim 100$ m thick. We can assume that at the surface $\mathbf{v} = 0$ (C3) and $\mathbf{u}\cdot\nabla p = 0$ (C12). However, within the boundary layer air velocity reaches a few meters per second at a height $H_v$ of just a few meters (e.g., Beare et al., 2006). Thus, linear extrapolation from $\mathbf{v} = 0$ at the surface to its known value at pressure level $p_1$, i.e. from $z = 0$ to $z = H_1 \gg H_v$, does not accurately reflect the velocity profile of the surface layer between $p_s$ and $p_1$. On the other hand, Eqs. (C4) and (C13) assume that within the surface layer the integrated quantity varies in the vertical in the same manner as it does between the two pressure levels nearest to the surface – e.g., between $p_1 = 1000$ hPa and $p_2 = 975$ hPa (MERRA) and $p_2 = 925$ hPa (NCAR/NCEP). With increasing vertical resolution, the two estimates should coincide. But using the available data they produce somewhat different results (Table 2).

Specifically, $W_K$ calculated by extrapolation, Eq. (C13), turns out to be higher than $W_K$ calculated assuming $\mathbf{v} = 0$, Eq. (C12). This is related to the vertical profile of $-\mathbf{u}\cdot\nabla p$ shown in Fig. 12. Kinetic energy generation grows with increasing pressure in the lower atmosphere. Extrapolation of this dependence to the surface yields a positive surface value for kinetic energy generation for $z = 0$.

In contrast, $W$ is smaller when extrapolated, Eq. (C4), than when assuming zero velocity at the surface, Eq. (C3). This can be explained by a different distribution of pressure velocity over pressure levels, Fig. 12. Here the lowest layer between 975





**Table 2.** Annual mean atmospheric power budget (W m$^{-2}$) in 2005-2015 estimated either assuming $\mathbf{v}_s = 0$ (subscript 1), see Eqs. (C3), (C12), or by extrapolation (subscript 2), see Eqs. (C4), (C13). Variables without subscripts are means of the two estimates.

| Variable | 2005 | 2006 | 2007 | 2008 | 2009 | 2010 | 2011 | 2012 | 2013 | 2014 | 2015 |
|---|---|---|---|---|---|---|---|---|---|---|---|
| $W_{K1}$ | 2.53 | 2.54 | 2.55 | 2.54 | 2.49 | 2.50 | 2.49 | 2.51 | 2.52 | 2.59 | 2.63 |
| $W_{K2}$ | 2.69 | 2.71 | 2.72 | 2.70 | 2.65 | 2.67 | 2.65 | 2.67 | 2.68 | 2.75 | 2.80 |
| $W_K$ | 2.61 | 2.62 | 2.63 | 2.62 | 2.57 | 2.59 | 2.57 | 2.59 | 2.60 | 2.67 | 2.72 |
| $W_1$ | 3.29 | 3.35 | 3.36 | 3.31 | 3.25 | 3.25 | 3.18 | 3.22 | 3.22 | 3.26 | 3.34 |
| $W_2$ | 3.06 | 3.09 | 3.09 | 3.08 | 3.00 | 2.96 | 2.92 | 2.94 | 2.93 | 2.98 | 3.08 |
| $W$ | 3.18 | 3.22 | 3.23 | 3.20 | 3.12 | 3.10 | 3.05 | 3.08 | 3.07 | 3.12 | 3.21 |
| $W_{P1}$ | 0.76 | 0.80 | 0.81 | 0.77 | 0.76 | 0.75 | 0.68 | 0.70 | 0.71 | 0.67 | 0.71 |
| $W_{P2}$ | 0.37 | 0.39 | 0.37 | 0.37 | 0.35 | 0.29 | 0.26 | 0.27 | 0.25 | 0.23 | 0.28 |
| $W_P$ | 0.56 | 0.59 | 0.59 | 0.57 | 0.55 | 0.52 | 0.47 | 0.49 | 0.48 | 0.45 | 0.49 |

hPa and the surface makes a large negative contribution to $W$. This is because the air predominantly descends in the regions of higher surface pressure. For example, with the same $\omega$ at 975 hPa, the layer between $p_s$ and 975 hPa is thicker where the air descends and $p_s = 1020$ hPa than where the air ascends and $p_s = 1000$ hPa. Since $W$ is proportional to $-\omega$, the net contribution of the lowest layer to global $W$ is negative. The net contribution of the higher pressure layers is positive, because there the ascent is associated with warmer and, hence, thicker pressure layers, an effect that appears more pronounced in the upper atmosphere.

The difference between the two estimates for $W$ and for $W_K$ is about 10% and of different sign. Extrapolation increases $W_K$ but diminishes $W$ (Table 2). The difference between the alternative $W_P$ estimates is greater: $W_P$ obtained by extrapolation is considerably smaller than $W_P$ obtained assuming zero velocity at the surface. This suggests that our conclusion about $W_P$ being underestimated in MERRA is robust.

**Appendix D:  Volume integral of pressure tendency**

As noted in Section 3, any magnitudes related to vertical velocity, including total atmospheric power $W$ (37), are associated with significant uncertainty. The vertical velocity is usually small compared to horizontal velocity. Rather than being observed directly, the vertical velocity is estimated from the generally larger horizontal velocities using the continuity equation. Minor uncertainties in the horizontal components permit major uncertainties in the vertical components.





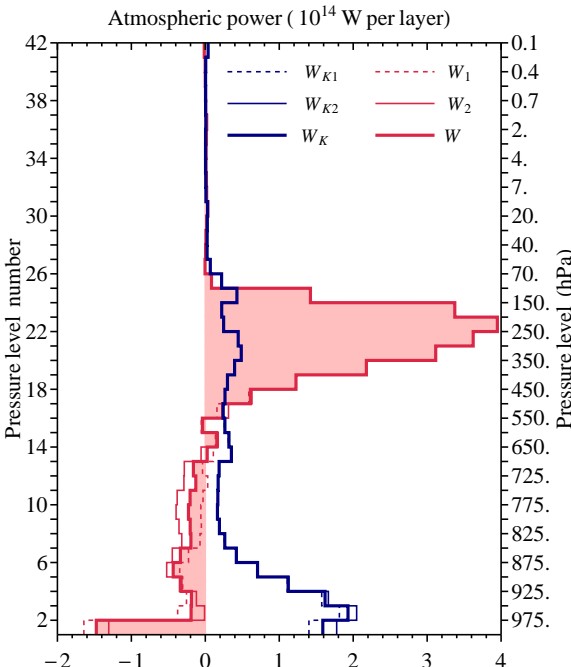

**Figure 12.** Atmospheric power within the 41 pressure layers enclosed by the 42 pressure levels in the MERRA dataset MAI3CPASM in 1979-2015. Each bar of the histogram contains the contribution from the corresponding pressure layer $(p_i, p_{i+1})$, where $i$ is pressure level number, plus the contribution from layer $(p_s, p_i)$ if $p_i \leq p_s$ in the considered cell is the pressure level nearest to the surface. For example, the lowest bar of the histograms corresponds to the layer with pressure less than 975 hPa (i.e. the layer from $p_1 = 1000$ hPa to $p_2 = 975$ hPa plus the layer from $p_s$ to $p_1$). Sum of the histogram values over all layers gives the global values of $W$ and $W_K$. Subscripts 1 and 2 refer to the two ways of estimating $W$ and $W_K$, see Table 2 for details.

Pressure velocity (61), which depends on vertical velocity, is calculated using the additional assumption of hydrostatic equilibrium from the continuity equation in the following form:

$$\nabla_p \cdot \mathbf{u} + \frac{\partial \omega}{\partial p} = 0, \quad \omega(p) = -\int_{p_s}^{p} (\nabla_{p'} \cdot \mathbf{u}) dp' + \omega(p_s). \quad (D1)$$

Here subscript $p$ at the nabla operator indicates that it is evaluated at constant pressure. For details see, for example, Kasahara (1974, his Eq. 6.4).

Pressure velocity calculated from Eq. (D1) is distinct from the material derivative of pressure $dp/dt$ (5). Consider a dry axisymmetric uniformly heated hydrostatic non-rotating atmosphere, which experiences slow periodic cooling and warming. In such an atmosphere the surface pressure tendency $\partial p_s / \partial t$ is zero (because the amount of gas does not change) and horizontal





velocity **u** is also zero (because of the spherical symmetry), so $\omega(p_s) = 0$. Therefore, according to Eq. (D1), omega must be zero at all heights and at all times.

However, it is clear that for any height $z > 0$ the instantaneous pressure tendency (and hence the material derivative of pressure) is not zero: it must reflect the temperature variation. In the simplest case when $p(z) = p_s \exp(-z/\mathcal{H})$, where $\mathcal{H} = RT/(Mg)$ is independent of $z$ (an isothermal atmosphere), we have $\partial p/\partial t = p(z/\mathcal{H}^2)\partial \mathcal{H}/\partial t$. In such an atmosphere the volume integral of $\partial p/\partial t = dp/dt$ is positive when the atmosphere is warming, and negative when it is cooling:

$$\Psi = \int\limits_0^\infty \frac{\partial p}{\partial t} dz = p_s \frac{\partial \mathcal{H}}{\partial t} = p_s \mathcal{H} \frac{1}{T} \frac{\partial T}{\partial t}. \tag{D2}$$

We calculated the global integral of pressure tendency $\Psi$ (C8) for the year 2010. It is shown in Fig. 13a together with $\Omega$ (61), (C6) and $W_K$ (38). We can see that $\Psi$ does indeed reflect the change of global temperature, Fig. 13b. By absolute magnitude, $\Psi$ constitutes a considerable part (about one quarter) of total long-term atmospheric power $W$ estimated from pressure velocity. This magnitude is derived from Eq. (D2) for a slowly warming/cooling atmosphere. Global mean temperature $T$ changes by $3°$ or by about 1% in half a year, $(1/T)(\partial T/\partial t) \sim 6 \times 10^{-10}$ s$^{-1}$, Fig. 13b. Global mean surface pressure changes insignificantly (by about 0.04% ) over the same period, Fig. 13c. So during the warming phase (the first half of the year) with $p_s = 10^5$ Pa and $\mathcal{H} = 10^4$ m we obtain $\Psi = 0.6$ W m$^{-2}$. This agrees well with Fig. 13a.

If we formally added the integral of pressure tendency to the integral of omega (61), the result would be absurd: at certain times of the year total power would have been smaller than kinetic power, Fig. 13a, dashed curve. This illustrates that omega includes only those contributions from the pressure tendency that are associated with macroscopic air motions and thus non-zero gradients of horizontal velocity, Eq. (D1).

Since the long-term average of $\Psi$ is zero (for the year 2010 we have $\Psi = 0.017$ W m$^{-2} \ll W$), it does not appear to affect the long-term mean estimate of $W$. However, this term can be important for quantifying conversion rates of the available potential energy to kinetic energy. Different approaches to estimating pressure tendency, either via Eq. (D1) or via temperature tendency Eq. (C8) should give different results for these rates (see, e.g., Kim and Kim, 2013).

Generally, Fig. 13 shows that instantaneous values of $\Omega$ do not reflect the instantaneous values of global atmospheric power $W$. Consequently, the difference $\Omega - W_K$ is not equal to the instantaneous value of the gravitational power of precipitation $W_P$. This may explain why on a seasonal scale $W_P$ is not correlated with precipitation $P$ as shown in Figs. 3b and 4.

*Author contributions.* All the authors designed the research, performed the research, discussed the results and wrote the paper. A.M.M., V.G.G. and A.V.N. performed the MERRA and NCAR/NCEP calculations.

*Acknowledgements.* We thank Silvio N. Figueroa, Paulo Nobre, Rémi Tailleux and three anonymous referees for useful comments. MERRA data used in this study/project were provided by the Global Modeling and Assimilation Office (GMAO) at NASA Goddard Space Flight Center through the NASA GES DISC online archive. NCAR/NCEP data were provided by Physical Sciences Division, Earth System Research Laboratory, NOAA, Boulder, Colorado, from their Web site at http://www.esrl.noaa.gov/psd/. This work is partially supported by the





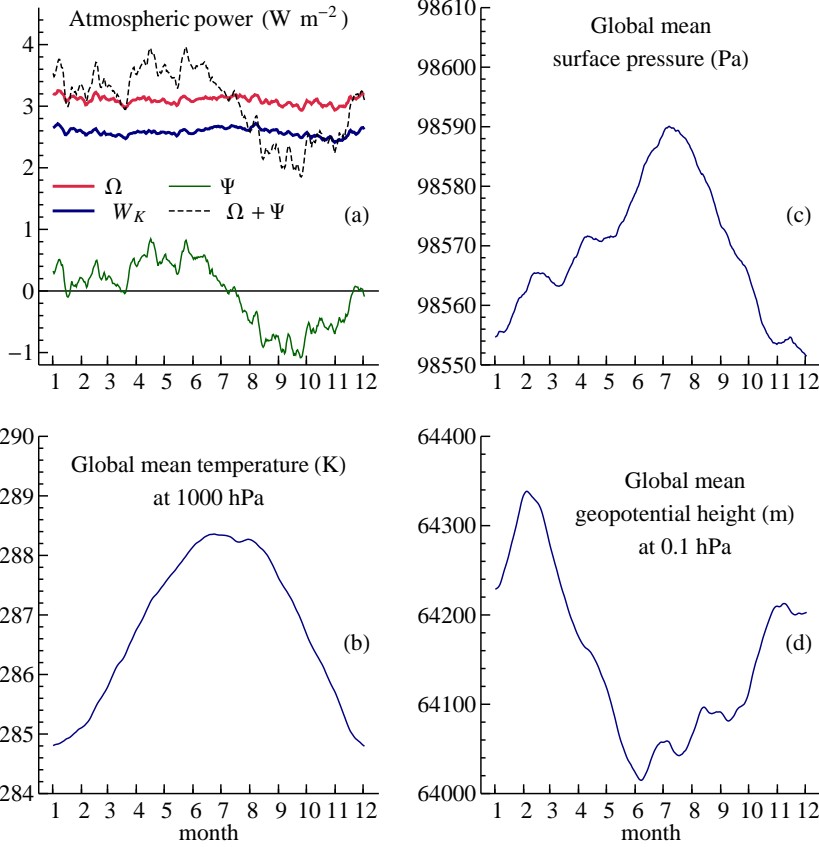

**Figure 13.** Time series (30-day running mean of daily values for the year 2010) of (a) the global integral of the pressure tendency $\Psi$ (C8), the omega integral $\Omega$ (61) and kinetic power $W_K$ (38), cf. Fig. 3; global mean surface temperature (b), global mean surface pressure (c) and global mean geopotential height at $p_t = 0.1$ hPa (d). This pressure level moves with vertical velocity $w_t$ of about 300 m in half a year, $w_t \sim 2 \times 10^{-5}$ m s$^{-1}$, which corresponds to $I_t \sim p_t w_t \sim 10^{-4}$ W m$^{-2} \ll W$ in Eqs. (13) and (14). Ticks on the horizontal axes correspond to the 15th day of each month.

University of California Agricultural Experiment Station, Russian Scientific Foundation under Grant no. 14-22-00281, Australian Research
Council project DP160102107 and the CNPq/CT-Hidro - GeoClima project Grant 404158/2013-7.



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
