# Peer review of "Quantifying the global atmospheric power budget"

_Atmospheric Chemistry and Physics, 2017_

## Referee Comment (RC1) · R. Tailleux (Referee) · 30 Mar 2017

**Summary and recommendation** I have reviewed previous incarnations of this paper a number of times now, and although I never liked it, I have at turns oscillated between overtly critical to more supportive, to the point of recommending its publication last time around. Feeling exhausted from reading yet another incarnation, and being disappointed that the authors have still not managed to improve clarity and succinctness, having also identified new major flaws, I have reverted to my original assessment that the paper is neither a useful contribution to the topic of atmospheric energetics, nor of sufficient scientific standard to be publishable in ACP or any other atmospheric journal.

The following provide some of my major objections to the paper. The paper is also much too long, often too speculative, contains too many technical errors, it lacks a

consistent description of boundary conditions, and does not even appear to capture such an effect as the enthalpy flux to the atmosphere due to freshwater entering the atmosphere having a higher temperature than that of the precipitation, which one would expect to be part of the problem. Moreover, it also relies too heavily on the continuity equations (5), which I believe is not the way moisture is handled in most climate and numerical weather prediction models, but is never mentioned.

**Major issues**

1. The authors motivate their paper by reviewing a number of definitions for atmospheric power that are equivalent for a dry atmosphere but not necessarily so in presence of condensation/precipitation, to ask the question of how best to define it in this case. This approach to the problem, however, is not optimal and furthermore misleading, because to a large extent, what one should call the atmospheric power depends on the model assumptions and equations used to describe the atmosphere. For instance, if one were to use the hydrostatic compressible Navier-Stokes equations

$$\rho \frac{Du}{Dt} + fk \times u + \nabla_h p = \rho F \tag{1}$$

then the steady-state kinetic energy budget would be:

$$\underbrace{-\int_V u \cdot \nabla p \, \mathrm{d}V}_{P} = \int_V \rho \varepsilon_K \mathrm{d}V \tag{2}$$

and one would naturally use the pressure gradient work of the horizontal velocity against the horizontal pressure gradient as the net atmospheric power. This is not really satisfactory either, however, because as pointed out in Tailleux (GRL,
2010), the pressure gradient work is a source of kinetic energy only when $-u \cdot \nabla_h p$ is positive, but a sink of kinetic energy and hence a dissipative term when it is negative. The correct approach is to split $P = P^+ - P^-$ as the difference between production and dissipation, in which case the balance becomes:

$$P^+ = P^- + \int_V \rho \varepsilon_K \, \mathrm{d}V. \tag{3}$$

In my opinion, such a decomposition is essential to understand atmospheric energetics, because as it is now increasingly realised, e.g., Tailleux (2013,Physica Scripta), dissipation of kinetic energy is not just achieved by viscous dissipation, as is often erroneously assumed, but by thermal dissipation as well (the so-called APE dissipation), which represents a second form of Joule Heating. There are two main ways to seek a decomposition of $P$ into a production and dissipation term from first principles, based respectively on using the APE or entropy budgets, which in both cases yields:

$$P = \gamma Q_{in} - D_{irr} \tag{4}$$

where the thermodynamic efficiency $\gamma_{APE}$ obtained from the APE budget is usually distinct from $\gamma_{entropy}$ obtained from the entropy budget. In the ocean, the distinction matters because $\gamma Q_{in}$ and $D_{irr}$ obtained via the entropy budget are about two orders of magnitude larger than the values obtained from the consideration of the APE budget, and only the APE budget appears to provide reasonable estimates. In the atmosphere, the differences between the APE and entropy budgets are less important. Pauluis (JAS, 2007) has made the point that based on his estimates of the sources of APE in a moist atmosphere, viscous dissipation alone could not balance the source of APE, which suggest that $P^-$ might be larger than the viscous dissipation term when moisture effects are explicitly retained. In the ocean for instance, the difference $G(APE) - D(APE)$ is about two orders of magnitude than each term taken separately. As far as I understand
the issue, only the use of the entropy or APE budget can shed light on the term $P^+$, which is the only meaningful way to define atmospheric power, since that the only term that is associated with the production of kinetic energy. Of course, using non-hydrostatic compressible equations and retaining the effects of a drag with the condensate will yield different kinetic energy budget, and hence a different form for what should be called the atmospheric power, but in all cases, only the positive part will matter. In order to make progress, the authors would need to explain how the term the thermodynamic efficiency $\gamma$ and $Q_{in}$ in the expression $\gamma Q_{in}$ are affected by condensation/evaporation. I do not understand how their approach is able to achieve that goal.

2. **Section 3.1 Boundary conditions** In this section, the authors discuss the boundary conditions for the air velocity at the surface. Physically, the problem is very simple: one needs a way to describe how freshwater enters and leaves the atmosphere via evaporation and precipitation. In practice, this is often done by enforcing a no-mass flux condition for the air velocity, freshwater entering the atmosphere via specification of the diffusive flux of freshwater. In ocean models, this has been the common practice for many years, evaporation and precipitation being then treated as a virtual salt flux. Over the past 20 years, however, ocean modelers have decided that it would be more physical to allow for the freshwater to enter and leave the ocean, as this is needed for instance to deal with sea level change due to land ice melting. In that case, one needs to modify the boundary condition for the diffusive part of the salt flux, so that no salt leaves or enter the ocean. A priori, exactly the same ideas could be used for the atmosphere, but the authors seem to argue that using a non-zero boundary condition for the vertical air velocity at the surface would lead to absurd results. This point is made in paragraph lines 187-194, and is based on pointing out that the term $I_s = p_s w_s$ would be dominate the energy budget. I admit that I was extremely puzzled by this statement last time around, which seemed to be wrong, but let

it pass because I could not immediately figure out the origin of the error. Upon reflection, the origin of the problem is quite simple, and arises from Makarieva et al.'s focus on the compressible work term $PDv/Dt$ in isolation, whereas the correct approach is to consider the internal energy budget in its totality. Indeed, if the full internal energy equation, the authors would have realised that the term $p_s w_s$ naturally combines with $\rho e_w w_s$, with $e_w$ the internal energy for freshwater, leading to a term $\rho(e_w + p_s/\rho)w_s = \rho h_w w_s$ which can be recognised as a flux of the enthalpy of freshwater, whereby the non-zero $w_s$ converts the low enthalpy of the freshwater into the high enthalpy of the water vapour. Physically, this term is related to the latent heat flux in the ocean, and with an additional enthalpy flux resulting from the temperature of the evaporated freshwater being on average larger than the temperature of the precipitating freshwater. The whole section is therefore completely erroneous and misleading. Without a proper understanding of how boundary conditions for freshwater work, I don't see how they can claim to provide new insights into the role of the condensation/evaporation in the atmospheric energy cycle.

3. **Section 4. Practical implications.** In this section, the authors criticize the neglect of the integral $\int_V dh/dt \, d\mathcal{M}$ in Laliberte et al. (2015). If the equations considered were formulated in terms of the full barycentric velocity, I would agree that this term is non-zero, and physically related to the exchange of freshwater between the land/ocean and the atmosphere, whereby the atmosphere gains freshwater at a higher temperature than it returns it to the land/ocean in the form of precipitation. The existence of this term has long been known by oceanographers, and was recently mentioned in Tailleux (2015), but it is generally considered to be sub-dominant in the ocean heat budget and often neglected. In practice, this term is not easy to incorporate in climate models, as it requires predicting the temperature at which rain falls into the ocean, but I am aware of some climate models which do it. A priori, this term is a enthalpy sink for the ocean,

and hence a heat source for the atmosphere. Whether Laliberte et al. (2015) were right or wrong in neglecting this term depends on the equations of motion underlying the MERRA re-analysis. Makarieva et al.'s objection is based on the assumption that the water vapour and condensate in the MERRA re-analysis satisfy the continuity equations (5)

$$\nabla \cdot (\rho v) = \dot{\rho}, \qquad \nabla \cdot (\rho_c b_c) = -\dot{\rho} \tag{5}$$

These equations, however, apply to the non-averaged equations of motion, but not necessarily to the large-scale motions considered in most climate and numerical weather prediction models. Indeed, most models seem to assume that the velocity carried in the model satisfy a continuity equation of the form

$$\frac{\partial \rho}{\partial t} + \nabla \cdot (\rho v) = 0 \tag{6}$$

with a separate evolution equation for

$$\frac{\partial r_v}{\partial t} + v \cdot \nabla r_v = -\nabla \cdot F_v + \dot{r}_v \tag{7}$$

for the mixing ratio or specific humidity $r_v$. If one furthermore assumes a no-mass flux condition at the surface, and represent evaporation via a diffusive flux of $r_v$ (the term $F_v$) above, then it seems correct to me to regard the integral of $Dh/Dt$ as zero, as assumed by Laliberte et al. (2015).

4. **Section 6** This section is quite speculative and philosophical in nature. In this section, the authors try to make the link with their biotic pump/condensation-driven theories, which are widely regarded as controversial, and which I have been pondering about for a very long time. Recently re-reading Makarevia et al. (2009), Precipitation on land versus distance from the ocean: evidence for a forest pump of atmospheric moisture, Ecological complexity 6, 302-307, I believe
I finally understood where Makarieva et al. got it wrong. In this paper, Makarieva et al. correctly points out that for a hydrostatic atmosphere, neither the dry air nor the water vapor are individually in hydrostatic balance. However, they incorrectly argue that $-[\partial p_v/\partial z + \rho_v g]$ is the force driving water vapor upward, where $p_v$ and $\rho_v$ are the pressure and density of water vapor. Indeed, the equation obeyed by water vapor is

$$\frac{D_v \rho}{Dt} + 2\Omega \times v_v + \frac{1}{\rho_v}\nabla p_v = -gk + \frac{1}{\rho_v}F_{dv} + \cdots \qquad (8)$$

where $F_{dv}$ is the intermolecular forces between dry air and water vapor responsible for keeping the difference between the dry air and water vapor individual velocities very small. To ensure momentum conservation, $F_{dv}$ appears with an opposite sign in the momentum equation for dry air. In general, the smallness of the velocity difference between dry air and water vapor means that $F_{dv}$ must balance at leading order $-[\partial p_v/\partial z + \rho_v g]$ and hence that the force actually acting on the water vapour is considerably smaller than assumed by Makarieva et al., invalidating their idea.

---

## Author Comment (AC1) · 31 Mar 2017

We thank our referee for the time spent on our work and for the numerous comments and criticisms which we sincerely value. While as we clarify below we do not agree with the criticisms, we nevertheless believe that this exchange is useful as it once again exposes and illustrates the many confusions and misconceptions surrounding the notion of atmospheric power.

**1. Atmospheric power**

1a. The referee characterises our approach to finding a proper formulation for atmospheric power as *"not optimal and furthermore misleading, because to a large extent, what one should call the atmospheric power depends on the model assumptions and equations used to describe the atmosphere."*

This is incorrect: there is no such freedom. To get a complete set of equations in atmospheric modelling one must use the first law of thermodynamics, which, as is well-known, relates heat input, mechanical work and change of internal energy of a given body. Since work in the atmosphere is performed by air parcels, to be consistent with, and constrained by, the laws of thermodynamics, global atmospheric power must be defined as work per unit time of all air parcels constituting the atmosphere. There is no other option.

We showed that independent of the presence of phase transitions –and indeed, contrary to the referee's claim, independent of the formulation of the equations of motion– total work per unit time of air parcels is given by the volume integral of $p\nabla \cdot \mathbf{v}$ (Eq. (3) on page 2), where $p$ is pressure and $\mathbf{v}$ is the velocity of gaseous air. The referee has never commented on this result (item 1 on page 34 in the list of results in the concluding section).

1b. The referee's statement that in a hydrostatic atmosphere the "net atmospheric power" is given by the integral of $-\mathbf{u}\nabla p$ (Eq. 2 in the review, $\mathbf{u}$ is horizontal velocity) is likewise incorrect. In the presence of phase transitions "net atmospheric power", i.e. cumulative work performed per unit time by expanding (positive work) and contracting (negative work) air parcels, is given by the sum of the integral of $-\mathbf{u}\nabla p$ and the gravitational power of precipitation (see Figure 1 in our paper). That is to say, if the *work per unit time* term in the first law of thermodynamics is integrated over the entire atmosphere, the result is not equal to the integral of $-\mathbf{u}\nabla p$.

1c. Even if the referee specified the "net" atmospheric power correctly, the statement that *"only the positive part [of this power] will matter"* does not have a scientific justification. The "net" atmospheric power certainly matters, which is why it has been the focus of so many researchers, both before and after Lorenz, Laliberté et al. (2015) included. It matters because it is this power that is constrained by the laws of thermodynamics. Therefore, it is important to have this power properly formulated for a moist atmosphere, which is done in our paper and has not been done before.

[Figure]

1d. Finally, we note that Tailleux (2010) considers decomposition of $pd(1/\alpha)$, $\alpha \equiv 1/\rho$ into a postive and a negative component, while in the present review the referee proposes to decompose the integral of $-u\nabla p$. These are however different magnitudes and neither is "net" atmospheric power. The first is atmospheric power minus a term proportional to the rate of phase transitions (see Eq. (11) on page 5) and the second one is kinetic power (kinetic energy generated per unit time, see Eq. (38) on page 13). Indeed, work and kinetic energy generation are not equivalent: there are processes where work is positive while kinetic energy generation is negative and vice versa (for details, see Makarieva et al. (2017) Tellus http://www.tandfonline.com/doi/full/10.1080/16000870.2016.1272752 ).

**The boundary condition**

2a. The referee disagrees with the boundary condition $w_s = 0$ for the vertical velocity at the surface, which we argue is necessary for the formulation of atmospheric power to be physically consistent. The reason given is *"Upon reflection, the origin of the problem is quite simple, and arises from Makarieva et al.'s focus on the compressible work term $PDV/Dt$ in isolation, whereas the correct approach is to consider the internal energy budget in its totality."*

We see a logical flaw here. In Section 3.1 we showed that if we assume a non-zero vertical velocity in the formulation of Eq. 3 of Pauluis et al. (2000) (hereafter PBH), then their own expression for atmospheric power, which is indeed the compressible work term, yields a figure exceeding solar power (see Figure 2 on page 16, right column). Since atmospheric power defined as "compressible work" is constrained by the first law of thermodynamics (and thus cannot exceed solar power), this means that assuming $w_s \neq 0$ as PBH did is not valid. This physical inconsistency has not been previously exposed. Thus the referee's objection appears to be pointless. Irrespective of whether we consider *"the internal energy budget in its totality"* or not, the obtained conclusion about atmospheric power remains unchanged.

2b. Furthermore, PBH state (and we agree) that the atmospheric power is

$$W = \int_{\mathcal{V}} p\nabla \cdot \mathbf{v} d\mathcal{V}. \tag{1}$$

At the same time, Laliberté et al. (2015) endorsed by Pauluis (2015) state (and we agree) that the atmospheric power is

$$W = -\int_{\mathcal{V}} \mathbf{v} \cdot \nabla p d\mathcal{V}. \tag{2}$$

As discussed in Section 3.1 of our paper, these two expressions agree with each other if and only if

$$\overline{p_s w_s} = 0. \tag{3}$$

If the referee disagrees with this boundary condition, he must disagree with either PBH or with Laliberté et al. (2015) or both regarding their formulation of atmospheric power. The referee, however, avoids commenting on the formulation of $W$, which is a bit unexpected, since the entire paper under review revolves around this issue.

2c. How the surface enthalpy flux should be consistently taken into account without assuming $w_s \neq 0$ is discussed in Section 4, see in particular Eqs. (59) and (60) on p. 21.

**3. Practical implications**

3a. The referee attempts to defend Laliberté et al. (2015) in that the enthalpy term was neglected by them correctly. To do so he states that *"most models seem to assume that the velocity carried in the model satisfy a continuity equation of the form"* of Eq. 6 in the review, which has a zero source/sink term.

Indeed, there are models including that underlying MERRA, which assume a zero sink/source in the continuity equation. However, as is well-known, such models do

not conserve mass (of the water vapor and dry air alike). Therefore, such models' output is not valid for the purposes of Laliberté et al. (2015) who intend to study how atmospheric power is affected by the water cycle (i.e. by mass non-conservation).

To get around this problem, Laliberté et al. (2015) adopt a velocity correction procedure which does take into account the sources and sinks in the continuity equation. This correction and its implications are discussed in our paper on page 27 but appear to have escaped the referee's attention. Moreover, we show using NCAR/NCEP data that atmospheric power calculated without such a correction (i.e. without taking the source/sink term into account) is unrealistic (negative). This illustrates that all quantitative conclusions of Laliberté et al. (2015) strongly depend on the correct account of the sink/source term in the continuity equation.

Our understanding of the situation is that Laliberté et al. (2015) following PBH just did not realize how crucially their conclusions depend on the correct form of the continuity equations. Indeed, it is unlikely that Laliberté et al. (2015) intended their approach to be only applicable to a model with a deficient representation of the water cycle (zero source/sink), as some colleagues are now trying to argue defending the omission. Rather, using the words of Pauluis (2015), they aimed to present an *elegant* physical approach to constrain the atmospheric power, an approach that would be generally valid such that using it one could conclude something about the real atmosphere. However, owing to an insufficient attention to the constraints imposed by mass conservation, the enthalpy term was incorrectly neglected in their formulations.

3b. The referee proposes that Laliberté et al. (2015) use Eq. 7 on p. c6 in the review for the mixing ratio. This is incorrect. The equation they use is given on p. 2 of their Supplementary Material; contrary to the referee's claim, this equation, $\partial q/\partial t + \mathbf{v} \cdot \nabla q = \dot{q}$, does not contain the diffusion term.

**4. Section 6** The referee states that he has found an error in the biotic pump physics by noting, following Makarieva et al. (2009) that dry air has a vertical distribution deviating

from the equilibrium in an opposite direction than the water vapor.

The referee states in his Eq. 8 on p. c7 that the non-equilibrium vertical pressure gradient of water vapor acts on the water vapor molecules alone. Then, considering that the velocity difference between water vapor molecules and dry air molecules is very small (in fact practically zero), he concludes that *"the force actually acting on the water vapour is considerably smaller than assumed by Makarieva et al., invalidating their idea."* This conclusion, as well as Eq. 8, is based on the incorrect premise that a pressure gradient chooses which molecules to act upon.

In reality, if there is a pressure gradient it will act on all air molecules irrespective of the gas that brought it in existence. It is a misconception to assume that a pressure gradient of water vapor drives water vapor, while a pressure gradient of dry air drives dry air. For example, in a hurricane if the air moves along an isothermal surface with a constant relative humidity, the pressure gradient is solely created by dry air molecules (since water vapor pressure is constant). Nevertheless, there is no doubt that the water vapor is also accelerated by the dry air pressure gradient. Thus nothing about atmospheric dynamics, including the biotic pump or indeed any other circulation driver, can be deduced from the consideration of the zero velocity difference between the different gases. So we feel that the referee's objection could have been better justified, which we would welcome very much. In any case, we are delighted that the referee has apparently found the biotic pump thought-provoking as it has kept the referee thinking *"for a very long time"*.

The biotic pump physics is about the non-equilibrium pressure gradient that forms as water vapor condenses. It is a vertical pressure gradient; however, because of a nearly instantaneous hydrostatic adjustment this gradient does not drive rapid vertical air motions. Instead, because of air re-distribution (which causes dry air to deviate from hydrostatic equilibrium), there appears a horizontal pressure gradient driving air motions with the same power as the non-equilibrium vertical gradient would. This theoretical proposition is, as we argue in a number of papers including the present one, confirmed

by observations.

**References**

F. Laliberté, J. Zika, L. Mudryk, P. J. Kushner, J. Kjellsson, and K. Döös. Constrained work output of the moist atmospheric heat engine in a warming climate. *Science*, 347:540–543, 2015. doi: 10.1126/science.1257103.

A. M. Makarieva, V. G. Gorshkov, and B.-L. Li. Precipitation on land versus distance from the ocean: Evidence for a forest pump of atmospheric moisture. *Ecol. Complexity*, 6:302–307, 2009.

A. M. Makarieva, V. G. Gorshkov, A. V. Nefiodov, D. Sheil, A. D. Nobre, P. L. Shearman, and B.-L. Li. Kinetic energy generation in heat engines and heat pumps: The relationship between surface pressure, temperature and circulation cell size. *Tellus A*, 69:1272752, 2017. doi: 10.1080/16000870.2016.1272752. URL http://www.tandfonline.com/doi/full/10.1080/16000870.2016.1272752.

O. Pauluis, V. Balaji, and I. M. Held. Frictional dissipation in a precipitating atmosphere. *J. Atmos. Sci.*, 57:989–994, 2000. doi: 10.1175/1520-0469(2000)057<0989:FDIAPA>2.0.CO;2.

Olivier M. Pauluis. The global engine that could. *Science*, 347:475–476, 2015. doi: 10.1126/science.aaa3681.

R. Tailleux. Entropy versus APE production: On the buoyancy power input in the oceans energy cycle. *Geophys. Res. Lett.*, 37:L22603, 2010. doi: 10.1029/2010GL044962.

---

## Editor Comment (EC1) · T. Garrett (Editor) · 6 Apr 2017

Hoping that this helps reviewers interpret the arguments of Makarieva et al., and evaluate their validity, I attach a response file that was not published online for a prior submission of the article: http://www.atmos-chem-phys-discuss.net/acp-2016-203/ The new response file may help in particular address some of the issues raised by Dr Remi Tailleux.

Tim Garrett

Please also note the supplement to this comment:
http://www.atmos-chem-phys-discuss.net/acp-2017-17/acp-2017-17-EC1-supplement.pdf

---

## Referee Comment (RC2) · Anonymous Referee #2 · 25 Apr 2017

I am not a specialist of the energetics of the atmosphere. As a physicist and climatologist, I first thought this paper could be, for me, a good opportunity to better understand this topic. Indeed, if the dynamics of the atmosphere appears rather well studied and understood, this is not quite true concerning its energetics. On this point, I must say I am quite disappointed. Indeed, the manuscript is long, not always consistent, and relies more on mathematical equations than on good physics. To summarize, it is more confusing than enlightening. I have looked through the rather numerous and lengthy discussions that this manuscript and its previously submitted version (in 2016) have generated. This did not help me to clarify what the main point of the paper actually is. I therefore do not recommend its publication.

1- Quite interestingly, Reviewer#2 of the 2016 submission said (point 5, acp-2016-203-RC2): "The way I see it, there are approximately three manuscripts in this study". At

this time, the manuscript was 29 pages long. Now, in this new submission, the scientific content is to a large extent mostly the same, but the manuscript has been significantly expanded to 52 pages. The manuscript ends with a summary of the main results are: a bullet list with 14 points. I therefore strongly agree with the above review's quote, and suggest to split the content into several separate papers (at least three, possibly more. . .). In its current state, I don't understand what is the main point of the paper. Is it about the definition of "atmospheric power" ? the role of condensation in this power ? the evaluation of atmospheric power using the MERRA re-analyses ? or in fine the speculation that moisture accounts for most of the atmospheric power as suggested in the last part ? I must admit I am rather sympathetic with this final speculation, but as it stands, I cannot defend such a confusing manuscript.

2 - In the introduction (part 1) the authors claim that the definition of atmospheric power varies quite a lot in the literature, depending on authors, and relate this difficulty to phase transitions of a moist atmosphere. They insist in defining power through an integral over the atmosphere of local quantities like velocity and pressure (equation 9). The discussion is quite strange. For instance, line 108: Âń Considering that pressure, too, varies across the parcel as velocity does, [. . .] total atmospheric power W would invariably be zero, which does not make sense Âż If I understand well, the authors suggest that the spatial scale of pressure should be different than the scale of velocity (line 104), which looks a bit strange to me. On the other side, the fact that the true integral of the work of the atmosphere should be zero does perfectly make sense : the global volume and pressure of the atmosphere being almost constant, there is little work transferred from the atmosphere to other components (except for ocean waves or hydraulic gravitational energy over land) in the usual thermodynamic sense. Thermodynamics deals with integral quantities of macroscopic objects (pistons and engines), and transfers of energy between them (heat and work). Therefore the rather clumsy, not convincing discussion, on page 4 of the manuscript, where the authors introduce sums of virtual boxes (line 88) without explaining what they represent in the real world. The key point is that mechanical energy (or power) generated in some parts of the atmosphere can be

dissipated in others. An (almost) null integral for the "net power" is therefore something quite natural and expected from simple thermodynamics. In physics, work is defined by force times length, with force being an interaction between TWO objects. Similarly, in thermodynamics, power implicitly involves TWO objects: an engine and an end-user that "dissipates the mechanical energy" and it is easy to define power through the forces acting between these TWO objects. The situation is obviously less clear when considering only ONE object. Here "atmospheric power" concerns the energetics of winds, or the generation of mechanical (kinetic and potential) energy WITHIN the atmosphere. The approach followed in this manuscript is to consider one term of the equations, the work of compression/expansion forces (W in eq. (9)), as the "power" system and, implicitly, viscous friction as "dissipation" or "end-user". This is a legitimate choice. But as noted by R. Tailleux in his review (acp-2017-17-RC1), this might not be the most appropriate one, since compression/expansion forces also include negative terms that dissipate energy: consequently "work of compression/expansion forces" is NOT equivalent to "generation of mechanical energy from heat". In any case, this is a choice. I strongly disagree with the author's point of view that "power" is a quantity defined a priori from the equations of the fluid. Power can only be defined as the mechanical energetic output of a "power system". Obviously, concerning the atmosphere, the "power system" is only an abstract part of the atmosphere, that needs to be specified first. I believe this misunderstanding largely explains why the introduction and the definition of "power" is so confusing in this manuscript.

3 – The first half of the paper (parts 1, 2 and 3) is 20 pages long, with 54 numbered equations (. . . plus many more in the text). The final paragraph reads: Âń In summary, PBH obtained a valid expression for precipitation-related dissipation WP* = WP + WC (44). Âż I certainly understand that the authors disagree with Pauluis et al (2000) (PBH) on the way to derive this expression. But if the subject of the paper is to discuss this point, then this should be the end of the manuscript. If the subject of the paper is to estimate atmospheric power from reanalysis, then this should be the start.

4 – As a motivation for examining atmospheric power, Makarieva et al. note a discrepancy between observations and models (lines 28-34). They mention a globally increasing intensity of observed winds, by citing de Boisséson et al. (2014) who are documenting such an increase in the equatorial Pacific only, and Huang & McElroy (2015) who are in fact using reanalysis, not wind data. In contrast, they do not mention the well documented phenomena of "global stilling": the statistically most robust trend for (surface) winds is a decreasing one (at least over continents where we have direct data) and possibly an increasing one at higher altitudes. This looks like a blatant misrepresentation of current knowledge on climatological wind trends.

---

## Author Comment (AC2) · 26 Apr 2017

We thank the referee for the time spent on our work and useful comments. In our response below we want to focus on the physical misconception of interpreting negative work as dissipation (suggested by Dr. Tailleux and seemingly approved by Referee 2 in his comment No. 2).

The more subjective presentation arguments are never as clear cut as physical arguments; so they are more difficult to address. We respect the referee's concern about our insufficient succinctness and clarity. To be quite honest when submitting the current version we felt very good precisely because we thought that we had made our presentation as clear and detailed as possible. Lack of a positive feedback to any of our efforts came quite puzzling to us though; such that we do not have any guidance

as to in which direction to move to achieve more appeal to our potential readers in the future. But we will continue our efforts.

**1. Length of the paper** We agree that the paper is long and contains many equations. However, we thought that if all of them are correct, and the results valuable, the length and the number of equations are not by themselves a shortcoming. It is good sometimes rather than dealing with the so-called "salami slicing" to have all the material in one place. Such a text does not necessarily make an easy reading but it can be used as a convenient reference document where you can find all the essential information related to a particular topic of interest.

The referee says: *"In its current state, I don't understand what is the main point of the paper. Is it about the definition of "atmospheric power"? the role of condensation in this power? the evaluation of atmospheric power using the MERRA re-analyses ? or in fine the speculation that moisture accounts for most of the atmospheric power as suggested in the last part ? I must admit I am rather sympathetic with this final speculation, but as it stands, I cannot defend such a confusing manuscript."*

In fact, the referee quite clearly names the main parts of our paper: it defines atmospheric power; discusses how condensation impacts the formulation of this power; evaluates this power using the obtained formulation and MERRA **and NCAR/NCEP re-analyses** (not mentioned by the referee) and then presents a discussion about moisture accounting for most of the atmospheric power. We thought that dividing these materials into several papers would just lead to unnecessary repetitions in each of them.

**2.1 The definition of atmospheric power**

The referee notes (our emphasis) that *"The approach followed in this manuscript is to consider one term of the equations, the work of compression/expansion forces ($W$ in eq. (9)), as the "power" system and, implicitly, viscous friction as "dissipation" or "end-user". This is a **legitimate choice**. But as noted by R. Tailleux in his review*

*(acp-2017-17-RC1), this* **might not be the most appropriate one** *since compression/expansion forces also include negative terms that dissipate energy: consequently "work of compression/expansion forces" is NOT equivalent to "generation of mechanical energy from heat". In any case, this is a choice.*

We need to turn to the basics of thermodynamics. Consider a Carnot cycle – a textbook example of how mechanical energy is generated from heat. Carnot cycle consists of two adiabates and two isotherms, the warmer one and the colder one. At the warmer isotherm the air expands (positive work). Then it expands at the first adiabat (positive work). Then it compresses at the colder isotherm (negative work). Finally, it compresses at the second adiabat to return to the initial point (negative work).

This cycle, as is well-known, consists of reversible processes only; no dissipation occurs at any stage. Still, the mechanical energy produced by this cycle is equal to the net work performed by the working body (gas), i.e. to the sum of the negative and positive amounts of work performed at the different stages of the cycle. It is exactly the case when *"work of compression/expansion forces" IS equivalent to "generation of mechanical energy from heat".*

The same situation occurs in the atmosphere: the ascending air motions are associated with expansion and positive work, the descending air motions are associated with compression and negative work. Horizontal air motions can be of either type.

Therefore, if we aim to have a definition of atmospheric power (mechanical energy generated per unit time) that conforms to the constraints of the laws of thermodynamics, we have **no other choice** but to define it as the net work performed by both compressing and expanding air parcels in the entire atmosphere.

The suggestion of Dr. Tailleux to define atmospheric power just as the positive part of work performed in a thermodynamic cycle appears to be based on a misinterpretation of the process of compression – negative work which he understands as "thermal dissipation". In reality the descending and compressing air parcels do not dissipate energy. Indeed, compressing air parcels increase their pressure, which plays the role of potential energy in the Bernoulli equation and can be reversibly transformed to kinetic energy. Compression of the descending air parcels in a hydrostatic atmosphere, where $p = \rho g h$, represents a reversible conversion of the diminishing gravitational energy into the increasing ideal gas pressure. Interpreting this as a dissipative process is incorrect.

Consider a simple analogy: a non-dissipating pendulum, which reversibly transforms kinetic energy to potential energy and back. In the picture of Dr. Tailleux, when the kinetic energy diminishes being transformed to potential energy we would be dealing with dissipation, since the work of the gravity force (work = path times force) is negative.

We can formally define a power for such a system: it is the rate at which potential energy is transformed to kinetic energy (or vice versa), that is, the store of energy divided by the halfperiod of the pendulum. The power of such a non-dissipative system will not be constrained by the laws of thermodynamics in the sense that it can be arbitrary – depending on the store of energy in the system and its internal dynamics.

Meanwhile if the pendulum is a dissipative one and loses a bit of its energy with each cycle of energy transformation – such that we need to add some energy (either potential or kinetic) to it to keep it steady, then the power of this system will be equal to the rate at which dissipation occurs. This power will be constrained by the laws of thermodynamics. It is for this definition of power that we argue.

**2.2 Let us not fully discount all previous literature on the subject**

The referee says *"I strongly disagree with the author's point of view that "power" is a quantity defined a priori from the equations of the fluid. Power can only be defined as the mechanical energetic output of a "power system". Obviously, concerning the atmosphere, the "power system" is only an abstract part of the atmosphere, that needs to be specified first. I believe this misunderstanding largely explains why the introduction and the definition of "power" is so confusing in this manuscript.*

[Figure]

This appears to be a peculiar situation: there is a huge literature on atmospheric power both before and after Lorenz (1967) who identified quantifying the global atmospheric power as a major challenge for atmospheric sciences. In all of this literature, including Pauluis et al. and Laliberte et al. whom we discuss in detail, atmospheric power is consistently defined and interpreted as the net work of compression/expansion with viscous dissipation as the "end-user" (in the words of Referee 2). In our work we show how this understanding of atmospheric power is formalized in the presence of phase transitions; how the resulting formulations depend on the form of the continuity equations, boundary conditions, equations of motions etc.

Meanwhile Dr. Tailleux and Referee 2 appear to oppose the whole of this literature by noting, with a reference to two papers by Tailleux (2010) and Tailleux (2015), that all this previous understanding is either "not the most appropriate" in the words of Referee 2 (who however admits that the conventional view is "legitimate") or even "misleading" in the words of Dr. Tailleux. Even if the idea of Dr. Tailleux about compression=dissipation would not have been a misconception, denying our paper the right for existence on this basis would have been unjustified, in our view.

**3. Pauluis et al. and manuscript structure**

The referee mentions again the number of equations; also in their first comment the referee notes that while at its first submission *"the manuscript was 29 pages long. Now, in this new submission, the scientific content is to a large extent mostly the same, but the manuscript has been significantly expanded to 52 pages.*

We will continue to make efforts to be as clear as possible but as different reviewers and readers have difficulty with different issues and items, this has inflated the text. In fact, the content is not exactly the same as much new material was added as follows:

(1) Referee 1 of our first submission suggested that we should look at other datasets beyond MERRA and other spatial/temporal scales – we undertook an effort to include NCAR/NCEP (and also extended the analysis time from fifteen to thirty five years).

This NCAR/NCEP, because of no velocity correction, presented quite different results than MERRA – this warranted an additional discussion which also had implications for our criticisms of Laliberte et al. Unfortunately, Referee 1 never showed up again; other referees showed no interest in MERRA versus NCAR/NCEP. But the paper became longer.

(2) Then, in our first submission, we did not criticize the study of Pauluis et al., which we respect and consider valuable even if not consistent when it comes to the power budget. We just referenced their work. However, precisely because we were laconic on this matter, Referee 3, who admitted being in contact with Dr. Pauluis when evaluating our work, claimed that we had appropriated their results. (Those very results that Dr. Tailleux and Referee 2 now find unsatisfying.) Thus, in the revised version we briefly explained which issues we had with the approach of Pauluis et al. This, however, appeared insufficient, since Referee 3 misinterpreted our criticisms when evaluating our revised manuscript. Providing recommendations for a possible re-submission, our handling editor advised that we should account for the previous work as clearly as possible. We thus extended our analysis of Pauluis et al. and provided a one page figure showing how our approaches are similar and where they differ. The paper became longer again.

(3) Finally, a major revision was undertaken following the recommendation of Dr. Tailleux who advised as follows (see page 14 here http://www.atmos-chem-phys-discuss.net/acp-2017-17/acp-2017-17-EC1-supplement.pdf )

*I would expect, however, that the definition of atmospheric power that one should use should be based on the full analysis of energetics, in particular, of the kinetic energy equation. However, the authors never make explicit what they assume the momentum equations to be, and it is unclear what their assumed global energy budget looks like. Physically, one would expect atmospheric power to satisfy a balance of the form Atmospheric Power = DISSIPATION Can they form a closed energy budget? What does the DISSIPATION term include? Does it include viscous dissipation only, or viscous*

*dissipation plus that due to the precipitation drag?*

The Editor likewise suggested that we should pay attention to this comment of Dr. Tailleux and represent the various parts of the atmospheric power budget as a box diagram. Accordingly, we performed all the requested analyses and added the needed diagram (Fig. 1). This has required a considerable extension of the presentation, because there are many subtleties and controversies in the momentum equations for the moist atmosphere – we tried to accurately sort all of them out. The paper became yet longer again, but we felt it was worth it. In our opinion, the paper has greatly improved since the first submission.

Unfortunately, Dr. Tailleux in his new evaluation of our work appeared uninterested in how we followed his previous recommendations; instead, he switched to the idea that the atmospheric power should be defined along the ways outlined in his 2010 paper (where negative work is incorrectly interpreted as a dissipative process).

**4. References to wind speed tendencies**

The referee points out that we do not list references about "wind stilling" in the lower atmosphere on land. We would be willing to include such a discussion. Our logic was to demonstrate that there are discrepancies between model predictions and observations: these discrepancies are among those factors that justify an increased attention to the atmospheric power topic. Accordingly, we listed references to such discrepancies, i.e. to evidence in favor of atmospheric intensification. DeBoisseson et al. is not a single piece of evidence for such discrepancies, a number of references they cite support this intensification too. True that the study of Huang and McElroy 2015 is based on a re-analysis but it is all we have to evaluate the global atmospheric power. The fact that the re-analysis data do not conform to the GCM predictions is an interesting fact in itself which should not, in our view, be dismissed without a consideration.

We thank our referees again for their efforts and this discussion.

---

## Editor Comment (EC2) · T. Garrett (Editor) · 22 Jun 2017

The article makes the following statements

"Here gas (water vapor) is created by evaporation and destroyed by condensation with a local rate  $\dot{\rho} \neq 0$  (kg m–3 s–1). As we will show, in this case each of the four candidate expressions  $W_I$ ,  $W_{II}$ ,  $W_{III}$  and  $W_{IV}$  are distinct. In a moist atmosphere the velocity notation in Eqs. (1)-(4) becomes ambiguous: is it the velocity of gaseous air alone or the mean velocity of gaseous air and condensate particles?... Consider an atmospheric parcel in a still atmosphere composed of pure water vapor. Let it condense into a droplet. Now the parcel's reduction in volume dV/dt

parcel expanding. equation of state...As a result, the expression for global atmospheric power does not explicitly depend on condensation rate. However, it is during such condensation-induced rapid expansion of the neighboring air parcels that the macroscopic pressure gradients can form to drive atmospheric circulation and determine the magnitude of atmospheric power W (9). The conventional view is that the circulation arises when some air parcels receive more heat than others and thus begin to expand. The cause of condensation-driven circulation is different. Here air parcels expand after condensation has reduced the concentration of gas in the adjacent space."

I am struggling to understand these statements based on a back of the envelope calculation of the rate of doing work. Let's take the expression due to density changes

$$\frac{dw}{dt} = RT \frac{d\ln\alpha}{dt}$$

For a very rough estimate of the displacement of air due to condensation, water vapor occupies, say 1/1000th of the volume on average in the atmosphere. When water vapor condenses, it turns into cloud, occupying perhaps 1/10th of the atmosphere. Condensate has an effectively negligible volume in comparison to water vapor, so condensation to form clouds concerns 1/10,000th of the atmosphere, that is  $d \ln \alpha \sim 1 \times 10^{-4}$ . Now, we consider that clouds condense over timescales roughly equivalent to the buoyancy period or about 100 s, therefore  $d \ln \alpha/dt \sim 1 \times 10^{-6}$ . Multiplying by  $RT \sim 10,000$  J/kg, we obtain a rate of doing work of 0.01 W/kg.

Now let's compare this to the latent heat release associated with condensation, also a molecular scale phenomenon since it is associated with phase changes. Using similar numbers, we might estimate a globally averaged condensate density of order 0.1 g/kg. Multiplying by the latent heat of condensation and dividing as before by the buoyancy period, one obtains.

$$\frac{dw}{dt} = \frac{dm}{dt}L$$

which works out to 2.5 W/kg.
So it would seem that of these two microscopic elements of work, the one discussed in the paper is negligible. Latent heat release is considered the primary mechanism for cloud production, since by reducing density, it enables cloud parcels to be positively buoyant with respect to their surroundings. LES models of cloud development appear to reproduce cloud phenomena very well without accounting for any reduction in atmospheric volume due to condensation. What is missing?

---

## Author Comment (AC3) · 11 Jul 2017

We thank Dr. Garrett for his interesting comment (hereafter EC). The view that latent heat release is more important than the reduced number of gas molecules has been advocated several times during our research on condensation-induced dynamics. In EC this view is offered as a query within a clear physical context. This makes it easier to demonstrate why it is incorrect: in brief, it misinterprets heat as work. Below we also explain how the atmospheric responses of the two effects have distinct time scales and why this matters.

[Figure]

**1 Heat vs work**

To estimate *the rate of doing work* associated with condensation, Dr. Garrett presents two expressions for what he refers to as the *two microscopic components of work*:

$$\frac{dw_1}{dt} = R^*T\frac{d\ln\alpha}{dt}, \quad \frac{dw_2}{dt} = -L^*\frac{d\ln m}{dt}. \tag{1}$$

Here $\alpha \equiv \tilde{V}/m$ (m$^3$ kg$^{-1}$) is the specific volume, $\tilde{V}$ (m$^3$) and $m$ (kg) are the volume and mass of the atmospheric gases, $L = 45$ kJ mol$^{-1}$ is the latent heat of vaporization, $L^* = L/M_v$, $R = 8.3$ J mol$^{-1}$ K$^{-1}$ is the universal gas constant, $R^* = R/M_v$, $M_v = 18$ g mol$^{-1}$ is molar mass of water vapor, $T$ is temperature.

Since the total atmospheric volume $\tilde{V}$ does not change, we have $dm/m = -d\alpha/\alpha$ and obtain from (1)

$$\left(\frac{dw_2}{dt}\right) \Big/ \left(\frac{dw_1}{dt}\right) = \frac{L}{RT} \equiv \xi \approx 18. \tag{2}$$

This ratio is about an order of magnitide smaller than estimated in EC (which assumed $dm/m = -10d\alpha/\alpha$), but still it is much greater than unity. From this EC concludes that "it would seem that of these two microscopic elements of work, the one discussed in the paper is negligible", while latent heat release, which is given by $dw_2/dt$ in (1), dominates.

The problem with this interpretation is that heat and work are two distinct concepts, with the efficiency $\varepsilon$ of heat conversion to work central to thermodynamics. To assess how much work $w_Q$ results from latent heat release, we write

$$\frac{dw_Q}{dt} = -\varepsilon L^*\frac{d\ln m}{dt}. \tag{3}$$

In contrast, $dw_1/dt$ is not the rate of heat input; in agreement with *the definition of work*, it describes the expansion of air parcels with pressure $p = RT/V$, where $V = M\alpha$ is

molar volume and $M$ is air molar mass. In fact, $dw_1/dt$ is equivalent to work performed per unit time by the air expanding into empty space freed by the condensation of water vapor.

Thus to build a general argument about latent heat being more important than the reduced number of gas molecules, one would need to demonstrate from first principles that $dw_Q/dt > dw_1/dt$, or, which is the same, to show that $\varepsilon > RT/L \approx 0.05$.

Such a proof does not exist. As illustrated by Fig. 2 of Goody (2003), under typical atmospheric conditions $\varepsilon$ for heat-driven convection is of the order of $10^{-2}$ and can vary around zero; negative values are also possible. The reason is that any air circulation is not confined just to the rising portion (where latent heat contributes to the positive buoyancy of the ascending air). It also necessarily includes the descending portion, where the sinking air warms (and thus potentially becomes positively buoyant too). Therefore, if we consider the effect of latent heat alone (i.e. ignore the radiative heat exchange processes), the same heating that makes the rising air parcels positively buoyant, will be a drain on the circulation power when these warm air parcels will have to descend. The net effect of latent heat in a steady-state will be precisely zero ($\varepsilon = 0$, see Gorshkov et al., 2012).

In other words, since pushing warm positively buoyant air parcels down is costly, the power of any circulation based on temperature effects is ultimately limited not by the rate of latent heat release but by the rate at which heat can be disposed of by the descending component of the circulation (Goody, 2003). In particular, when the radiative cooling rates are low, latent heat release serves not as a driver but as a break prohibiting sustained circulation ($\varepsilon < 0$).

Previous versions of the argument about the greater significance of latent heat release, including the commentary of Dr. D. Rosenfeld[1] in 2008 and the more recent
* * *
[1]http://www.atmos-chem-phys-discuss.net/8/S12426/2009/acpd-8-S12426-2009.pdf, page S12436

commentary of Dr. A. Kleidon[2], were grounded in a modified relationship (2) of the type $L/(c_v T) \approx 7 \gg 1$, where $c_v = (5/2)R$. This implies warming the atmosphere with latent heat by $\Delta T = \Delta Q / c_v$ and calculating the associated buoyancy increase as $\Delta T / T$ to compare it with $\Delta \alpha / \alpha$. EC is distinct in addressing the ultimate question – circulation power (rate of doing work), rather than the buoyancy of the rising air which may or may not be associated with net positive work of the circulation as a whole.

To summarize, whatever the meaning of $dw_1/dt$ and irrespective of its relevance to large-scale atmospheric dynamics, which, we emphasize, should be justified separately as we do in our article, $dw_2/dt$ is not the rate of doing work. Thus neither $L/(RT) \gg 1$ nor, as a variant, $L/(c_v T) \gg 1$, prove *the greater importance of latent heat*.

**2 Time scales**

The distinct time scales of the different physical processes associated with phase transitions are crucial for describing moist dynamics (Makarieva et al., 2017). While the time scales of latent heat release and the reduction of the number of gas molecules are the same, the time scales of the atmospheric responses are not.

When water vapor condenses in rising air, the reduction in the number of gas molecules perturbs the vertical pressure distribution. Indeed, owing to the ideal gas law at constant temperature we have $\Delta \alpha / \alpha \sim -\Delta p / p$. Any pressure disturbance leads to an adjustment. Once the vapor is removed from the middle troposphere where condensation has occurred, the lower air must expand to fill the void.

This hydrostatic adjustment has two effects: (1) the pressure deficit shifts downward
* * *
[2]Pers. com. to Anastassia Makarieva 05 April 2016 and http://www.hydrol-earth-syst-sci-discuss.net/12/C4945/2015/hessd-12-C4945-2015.pdf, page C4946

producing a horizontal pressure gradient at the surface between the condensation area and the ambient environment and (2) air density is partially restored where condensation has occurred (since the air has risen during the adjustment and partially replaced the condensed water vapor). In the result of the hydrostatic adjustment (even if it is incomplete), the actual density change in the upper atmosphere becomes much less than $\Delta\alpha/\alpha$. In this sense it would be a misnomer to refer to the effect of the reduced number of gas molecules as a "density effect".

In contrast, the density change associated with temperature change, $\Delta T/T$, has a much longer relaxation time scale (governed by heat conduction which depends on radiative exchanges and turbulent air motions) than the hydrostatic adjustment.

Because of these distinct scales, the "slow" temperature effects are, via parameters for radiative exchange and turbulence, automatically included into the standard set of differential equations governing air motion. (In this aspect a moist atmosphere is formally identical to a dry atmosphere with a heat source equal to latent heat release.) The "instantaneous" dynamic effects of condensation must be incorporated manually, for example, via a corresponding constraint on the resulting circulation power.

(A similar situation, as we recently demonstrated, exists with the interaction of the air with condensate particles. If the time scale of this interaction is much smaller than the time scale of air motion, a specific term must be introduced into the equations of motion representing the drag resulting from this interaction (Makarieva et al., 2017). This term is absent in the standard set of equations and cannot be deduced from them. It is incorporated *ad hoc* based on a separate consideration of the relevant physical processes.)

We thank Dr. Garrett once again for interesting comments and would welcome any further opportunities to clarify these relationships. We address the query about cloud physics and models in a separate comment.

**References**

R. Goody. On the mechanical efficiency of deep, tropical convection. *J. Atmos. Sci.*, 60:2827–2832, 2003. doi: 10.1175/1520-0469(2003)060<2827:OTMEOD>2.0.CO;2.

V. G. Gorshkov, A. M. Makarieva, and A. V. Nefiodov. Condensation of water vapor in the gravitational field. *J. Exp. Theor. Phys.*, 115:723–728, 2012.

A. M. Makarieva, V. G. Gorshkov, A. V. Nefiodov, D. Sheil, A. D. Nobre, P. Bunyard, P. Nobre, and B.-L. Li. The equations of motion for moist atmospheric air. *Journal of Geophysical Research: Atmospheres*, 2017. doi: 10.1002/2017JD026773. 2017JD026773.

K. V. Ooyama. A dynamic and thermodynamic foundation for modeling the moist atmosphere with parameterized microphysics. *J. Atmos. Sci.*, 58:2073–2102, 2001. doi: 10.1175/1520-0469(2001)058<2073:ADATFF>2.0.CO;2.
* * *

---

## Author Comment (AC4) · 30 Jul 2017

Dr. Garrett commented: "Latent heat release is considered the primary mechanism for cloud production, since by reducing density, it enables cloud parcels to be positively buoyant with respect to their surroundings. LES models of cloud development appear to reproduce cloud phenomena very well without accounting for any reduction in atmospheric volume due to condensation. What is missing?"

In brief, it depends on cloud type and spatial scale. Our view is that condensation as a circulation driver is least important for shallow convective clouds, comparable to buoyancy for deep convective clouds and dominates at horizontal scales exceeding the atmospheric scale height.

In parallel, Large-Eddy Simulations (LES) are more unambiguous and thoroughly tested by observations for shallow non-precipitating clouds (where net condensation is zero) than they are for deep convection. For precipitating clouds LES are less robust and require additional tuning. Finally, LES models, where the large-scale environmental properties are set up externally, cannot in principle quantify the large-scale dynamic effects of condensation, i.e. the upscaling of condensation effects remains unaccounted for.

While latent heat release does make cloud parcels positively buoyant, this does not only enhance the upward motion but also suppresses the subsiding motion. The latter is equally important for cloud formation. For example, if evaporative cooling in downdrafts (which compensates for the latent heat release in updrafts) is switched off, deep convection in LES models may not form at all. The entrainment of environmental air is crucial for clouds. The role of condensation dynamics in this process is implicit in the corresponding parameterizations.

Below we discuss these statements in greater detail.

**1 Relevant scales for condensation dynamics**

The basic spatial scale of condensation dynamics is the scale height $h_\gamma \equiv -\gamma/(\partial\gamma/\partial z)$ of the relative partial pressure $\gamma \equiv p_v/p$ of saturated water vapor. It is of the order of the atmospheric height $h_\gamma \sim h \sim 10$ km. This is why water vapor condensation can generate atmospheric motions[1].

If there were no pressure adjustment, condensation in the ascending air would create a pressure perturbation of the order of $\Delta p \sim p_v h_c/h_\gamma$, where $h_c$ is the vertical dimension
* * *
[1] In contrast, evaporation cannot: here the relative partial pressure of water vapor varies over a microscopic scale of the order of one free path length above the evaporating surface (see Section 3.1 in our article). At this scale dominated by molecular viscosity macroscopic motions cannot arise by definition.
of the condensation region (Fig. 1a). If $h_c \ll h$, then the unperturbed pressure values $p_t$ and $p_s$ at the top and bottom of the condensation region are approximately equal, $p_t \approx p_s$. In this case the downward and upward pressure adjustment processes initiated by the pressure perturbation counteract each other and their cumulative impact on the ascending motion is negligible.

Likewise, if the horizontal scale $l_c$ of the condensation area is much smaller than the vertical one, $l_c \ll h_c$, the pressure adjustment processes will occur more rapidly in the horizontal plane than in the vertical plane. Since prior to adjustment the pressure perturbation would be maximum at the top of the condensation area, the horizontal air convergence at the cloud top could suppress the vertical motion and hence condensation itself.

Meanwhile if condensation occurs over a larger distance $l_c \gg h_\gamma$, the effect of the horizontal pressure adjustment is affecting only the edge of the condensation area and is thus minor relative to the upward vertical adjustment. Thus, condensation dynamics can act as a circulation driver at the scale $l_c \gg h_\gamma$ provided that $h_c \sim h_\gamma$ (Fig. 1b). This is the case of deep convection occupying a larger region with $l_c \gtrsim 100$ km.

**2  Shallow convective clouds**

In shallow convective clouds neither of the above two conditions is fulfilled. Instead, we have $h_\gamma \gg h_c \sim 2$ km and $h_c > l_c \sim 0.5$ km. The role of condensation as a possible driver of these clouds is minimal. Besides, in non-precipitating clouds condensation (in the cloud core) and evaporation (in the subsiding shell) compensate each other at a horizontal distance $l_c \ll h_\gamma$. In this case condensation dynamics does not have a direct impact on the larger scale circulation either. However, its within-clouds effects are implicitly included into the LES parameterizations of turbulence. We explain this below.

LES models have been primarily tested for shallow convective clouds (e.g., Rodts et al., 2003; Neggers et al., 2003; Jonker et al., 2008; Heus et al., 2009; Katzwinkel et al., 2014). A remarkable finding confirmed by observations was a subsiding shell surrounding each cloud. This finding altered the previous view of cumulus convection which presumed that subsidence occurred over a large area rather than was concentrated near the cloud (Jonker et al., 2008).

It was hypothesized that the subsiding shell can either be driven by negative buoyancy (i.e. when the subsiding air is colder than the ambient air) or by the mechanical forcing (i.e. by a relative pressure surplus at the cloud top that would be pushing the air down) (Rodts et al., 2003; Jonas, 1990).

In the latter case the subsiding shell could be positively buoyant – like the warm air descending in the hurricane eye. From the thermodynamic viewpoint (i.e. considering the circulation energy budget), such a motion is possible. If the potential energy associated with buoyancy (the conventional CAPE) is transformed into kinetic energy of the rising air and if this kinetic energy is then transformed into the potential energy of the pressure surplus at the cloud top, this pressure surplus could make the cloudy air descend even if it is relatively warm.

In reality, however, LES models and observations showed that the subsiding cell is driven by negative buoyancy caused by evaporative cooling of ambient air. There were no positively buoyant downdrafts. This means that the kinetic energy of the ascending motion dissipates to heat rather than transforms to the potential energy of the pressure perturbation. This property is set by the parameterization of the dissipative processes (i.e., turbulence) in the LES model. Thus *it is turbulence that determines to what degree the positive buoyancy of the ascending air is able to drive the circulation as a whole* (and to what degree this circulation depends on additional factors like air entrainment and evaporative cooling).

Evaporation can cool only those air parcels that have not been previously warmed by
latent heat release (if a droplet evaporates in the same air parcel where it condensed the net effect will be zero). Thus, given the dominant role of negative buoyancy, the entrainment of external air into the cloud is crucial for cloud formation (de Rooy et al., 2013). Condensation initiates pressure adjustment processes and thus impacts the entrainment process both in the vertical and horizontal plane (Fig. 1a). This impact should be implicitly taken into account into those parameters of turbulence that are fitted to observations.

In summary, large-scale air flow interacting with the planetary boundary would produce shallow convection in both dry and moist atmosphere. In this sense condensation does not directly drive shallow convection. However, the dynamic effects of condensation modify the properties of the shallow clouds by impacting entrainment and turbulence.

**3  Deep convection**

For deep convection we have $h_\gamma \sim h_c$ and $l_c \lesssim h_c$ for individual clouds. It is therefore the minimal horizontal scale where condensation can act as the driver of circulation. LES studies of deep convection demonstrate that by choosing a proper turbulence scheme, spatial resolution and cloud microphysics it is possible to quantitatively describe some cloud properties (Khairoutdinov and Randall, 2006; Morrison et al., 2015; Heath, 2015; Potvin et al., 2017; Fiori et al., 2017). However, what determines the correct choice of key parameters remains uncertain.

Thus, current LES models of deep convection do not prove that condensation dynamics is unimportant compared to buoyancy. With maximum air velocity of the order of $\sqrt{2p_v/\rho} \sim 60$ m s$^{-1}$ available from condensation, to drive convective clouds with their typical velocities of 1-2 m s$^{-1}$ requires only a minor portion of total potential energy associated with the partial pressure of water vapor. Compared to observations, deep convection simulated without accounting for condensation dynamics might, for
example, display an excessive positive (negative) temperature anomaly of the updrafts (downdrafts). Furthermore, condensation dynamics, with its non-equilibrium vertical pressure gradients, can play a significant role in overcoming convective inhibition. This cannot be elucidated by current LES models.

Most importantly, in deep convective clouds LES outputs are crucially dependent on the parameters of the large-scale circulation that are externally imposed onto the LES model. (As a simple example, a large-scale subsiding motion will suppress deep convection.) Without an explicit account of condensation dynamics it is not possible to correctly quantify the feedback of deep convection on the large-scale motion and hence to obtain a self-consistent picture of atmospheric circulation.

**4   What is missing on a larger scale?**

Potential energy available for conversion to kinetic energy is associated with spatial heterogeneity. In the conventional approach outlined by Lorenz, available potential energy arises when some parts of the atmosphere are warmed (or cooled) more than the others. If the atmosphere is uniformly warmed or cooled, potential energy is not available.

The situation is similar for condensation. Consider a horizontally isothermal atmosphere in hydrostatic equilibrium. We remove some gas uniformly across the entire atmosphere in such a manner that the hydrostatic equilibrium is not perturbed (i.e. we remove a constant air fraction at each altitude.) In such a case pressure declines everywhere, but no motion results.

Now consider removing gas by condensation and precipitation from a large but limited region of horizontal size $l_c \gg h$ (e.g. the equatorial region), once again without perturbing the hydrostatic equilibrium. Now, as the region's surface pressure declines, air will flow towards it from the surrounding atmosphere. The greater the surface pressure

perturbation $\Delta p_s$, the greater the cross-isobaric horizontal velocity $u$ of the air flow.

Historically though meteorological sciences attributed observed $\Delta p_s$ to differential heating assuming its dominant role in driving atmospheric circulation. Since it was not possible to quantify $u$ and $\Delta p_s$ from theory knowing $\Delta p$ in the upper atmosphere (Fig. 2a,b), the values of $u$ and $\Delta p_s$ were fitted to observations by adopting the necessary parameterizations of turbulence (see Introduction in Makarieva et al. (2017) for a more detailed discussion).

Having thus built a plausible model of dry atmospheric dynamics, people then added a "mass sink" via the moist continuity equations, which specify how precipitation influences the surface pressure tendency in a hydrostatic atmosphere. Since little changed in the resulting circulation patterns, it was concluded that the "mass sink" (and, hence, condensation changing the amount of gas) is inconsequential.

However, this overlooked the main effect of condensation: the formation of a local non-equilibrium vertical pressure gradient in air parcels rising in the gravitational field. As we discussed in our previous comment[2], it may have been overlooked because this gradient elicits a pressure adjustment and vanishes on time scales significantly shorter than the characteristic time scales of the large-scale air circulation it generates. Indeed, condensation is not equivalent to a hydrostatic mass sink. Any new droplet instantaneously produces an upward pressure gradient: since oversaturation of water vapor in a dry adiabatically ascending air parcel increases with height, condensation removes more gas from the upper part of any volume affected by droplet formation than it does from the lower. If $h_c \sim h_\gamma$ the process of hydrostatic adjustment affects the entire atmospheric column (Fig. 1b), enhances the ascending motion and transforms potential energy contained in the condensation-induced pressure perturbation into the kinetic energy of macroscopic air motions.

Assuming that it is condensation that drives the circulation explains, first, why the char-

[2] https://doi.org/10.5194/acp-2017-17-AC3

acteristic horizontal pressure perturbations $\Delta p_s$ coincide in the order of magnitude (10 hPa) with the partial pressure of water vapor at the planetary surface – and are independent of the horizontal scale of the circulation. Indeed, in the words of Holton (2004), apart from the synoptic scale of $10^3$ km "pressure fluctuations of similar magnitudes occur in other motion systems of vastly different scale such as tornadoes, squall lines, and hurricanes". Second, condensation dynamics also explains the observed relationship between the circulation *power* and precipitation in phenomena as diverse as hurricanes (e.g., Makarieva and Gorshkov, 2011) and global atmospheric circulation (Makarieva et al., 2013b,c, and our present article).

Put simply, in the conventional picture the temperature-induced pressure gradient pushes the air away from the warmer air column in the upper atmosphere (Fig. 2b). This creates a relative pressure deficit in the upper atmosphere and initiates the upward motion. Our position, on the other hand, is that the pressure gradient in the upper atmosphere cannot ensure the necessary pressure deficit in the vertical (because in a rotating atmosphere a steady state is the geostrophic air flow with no cross-isobaric motion). It is a dynamic (not a thermodynamic) limitation[3]. The ultimate cause of the circulation is the ascent induced by the condensation pressure perturbations (Fig. 2c).

Since condensation usually occurs in warm rising air (although there are exceptions like the Ferrel cell), turbulence parameterizations fitted to support the temperature-driven model often produce a realistic output. However, if, as we argue, the real driver of the circulation is not temperature but condensation, then such models will fail to predict what happens when the considered area gets warmer but drier (i.e. condensation and warmth do not coincide in space and time). This situation is especially relevant for the prediction of monsoons and the effects of deforestation. If one assumes that it is temperature that drives winds (while it is not), we underestimate the danger of deforestation for the atmospheric transport of moisture from ocean inland (Makarieva
* * *
[3]Another dynamic limitation, as we discussed in the previous section, does not allow deep convection to form unless there are negatively buoyant descending air parcels.

and Gorshkov, 2007; Makarieva et al., 2013a).

To summarize, condensation dynamics appears to have been neglected without a serious assessment of its crucial aspects. We believe that without accounting for these aspects the challenges currently faced by the meteorological science when describing moist atmospheric dynamics will persist. We thus welcome this discussion, thank Dr. Garrett once again for having made is possible and look forward to its continuation in the future, in one form or another.

**References**

de Rooy, W. C., et al., 2013: Entrainment and detrainment in cumulus convection: an overview. *Q. J. R. Meteorol. Soc.*, **139**, 1–19, doi:10.1002/qj.1959.

Fiori, E., L. Ferraris, L. Molini, F. Siccardi, D. Kranzlmueller, and A. Parodi, 2017: Triggering and evolution of a deep convective system in the Mediterranean Sea: modelling and observations at a very fine scale. *Q. J. R. Meteorol. Soc.*, **143**, 927–941, doi:10.1002/qj.2977.

Heath, N. K., 2015: WRF nested large-eddy simulations of deep convection during SEAC4RS. Ph.D. thesis, Florida State University.

Heus, T., C. F. J. Pols, H. J. J. Jonker, H. E. A. Van den Akker, and D. H. Lenschow, 2009: Observational validation of the compensating mass flux through the shell around cumulus clouds. *Quarterly Journal of the Royal Meteorological Society*, **135**, 101–112, doi:10.1002/qj.358.

Holton, J. R., 2004: *An Introduction to Dynamic Meteorology. Fourth Edition*. Elsevier, 535 pp.

Jonas, P., 1990: Observations of cumulus cloud entrainment. *Atmospheric Research*, **25**, 105–127, doi:10.1016/0169-8095(90)90008-Z.

Jonker, H. J. J., T. Heus, and P. P. Sullivan, 2008: A refined view of vertical mass transport by cumulus convection. *Geophysical Research Letters*, **35**, n/a–n/a, doi:10.1029/2007GL032606, l07810.

Katzwinkel, J., H. Siebert, T. Heus, and R. A. Shaw, 2014: Measurements of turbulent mixing and subsiding shells in trade wind cumuli. *J. Atmos. Sci.*, **71**, 2810–2822, doi:10.1175/JAS-D-13-0222.1.

Khairoutdinov, M. and D. Randall, 2006: High-resolution simulation of shallow-to-deep convection transition over land. *Journal of the Atmospheric Sciences*, **63**, 3421–3436, doi: 10.1175/JAS3810.1.

Makarieva, A. M. and V. G. Gorshkov, 2007: Biotic pump of atmospheric moisture as driver of the hydrological cycle on land. *Hydrol. Earth Syst. Sci.*, **11**, 1013–1033.

Makarieva, A. M. and V. G. Gorshkov, 2011: Radial profiles of velocity and pressure for condensation-induced hurricanes. *Phys. Lett. A*, **375**, 1053–1058, doi:10.1016/j.physleta.2011.01.005.

Makarieva, A. M., V. G. Gorshkov, and B.-L. Li, 2013a: Revisiting forest impact on atmospheric water vapor transport and precipitation. *Theor. Appl. Climatol.*, **111**, 79–96.

Makarieva, A. M., V. G. Gorshkov, A. V. Nefiodov, D. Sheil, A. D. Nobre, P. Bunyard, and B.-L. Li, 2013b: The key physical parameters governing frictional dissipation in a precipitating atmosphere. *J. Atmos. Sci.*, **70**, 2916–2929, doi:10.1175/JAS-D-12-0231.1.

Makarieva, A. M., V. G. Gorshkov, A. V. Nefiodov, D. Sheil, A. D. Nobre, P. L. Shearman, and B.-L. Li, 2017: Kinetic energy generation in heat engines and heat pumps: The relationship between surface pressure, temperature and circulation cell size. *Tellus A*, **69**, 1272 752, doi: 10.1080/16000870.2016.1272752.

Makarieva, A. M., V. G. Gorshkov, D. Sheil, A. D. Nobre, and B.-L. Li, 2013c: Where do winds come from? A new theory on how water vapor condensation influences atmospheric pressure and dynamics. *Atmos. Chem. Phys.*, **13**, 1039–1056, doi:10.5194/acp-13-1039-2013.

Morrison, H., A. Morales, and C. Villanueva-Birriel, 2015: Concurrent sensitivities of an idealized deep convective storm to parameterization of microphysics, horizontal grid resolution, and environmental static stability. *Mon. Wea. Rev.*, **143**, 2082–2104, doi:10.1175/MWR-D-14-00271.1.

Neggers, R. A. J., P. G. Duynkerke, and S. M. A. Rodts, 2003: Shallow cumulus convection: A validation of large-eddy simulation against aircraft and Landsat observations. *Q. J. R. Meteorol. Soc.*, **129**, 2671–2696, doi:10.1256/qj.02.93.

Potvin, C. K., E. M. Murillo, M. L. Flora, and D. M. Wheatley, 2017: Sensitivity of supercell simulations to initial-condition resolution. *J. Atmos. Sci.*, **74**, 5–26, doi:10.1175/JAS-D-16-0098.1.

Rodts, S. M. A., P. G. Duynkerke, and H. J. J. Jonker, 2003: Size distributions and dynamical properties of shallow cumulus clouds from aircraft observations and satellite data. *Journal of the Atmospheric Sciences*, **60**, 1895–1912, doi:10.1175/1520-0469(2003)060<1895:SDADPO>2.0.CO;2.

[Figure]

Vallis, G. K., 2006: *Atmospheric and Oceanic Fluid Dynamics: Fundamentals and Large-Scale Circulation*. Cambridge University Press, 745 pp.

[Figure]

[Figure]

Figure 1: Scales relevant for condensation dynamics, see text for details. (a) Shallow non-precipitating cloud, (b) a large area occupied by deep convection. Red solid arrows indicate the direction of air motion during the pressure adjustment induced by condensation.

**Fig. 1.**

[Figure]

Figure 2: Different views on what drives the large-scale atmospheric circulation. (a) Vertical profile of the pressure difference between two hydrostatic air columns of which one is warmer by 10 deg K but has a pressure deficit of 10 hPa at the surface (thick solid curve), see Fig. 1d of Makarieva et al. (2017). (b) Temperature-driven circulation: the necessary condition for the circulation to occur is the temperature-induced horizontal pressure gradient in the upper atmosphere (thick horizontal arrow), see panel (a). (c) Condensation-driven circulation: the necessary condition is the vertical pressure perturbation associated with condensation in the ascending air (thick vertical arrow). "Higher" and "Lower" at horizontal arrows refer to air pressure relative to the average at that height (cf. Fig. 2.6 of Vallis (2006)); "Higher" and "Lower" at vertical arrows refer to the corresponding perturbations of the hydrostatic pressure distribution.

**Fig. 2.**

[Figure]

---

## Author Comment (AC5) · 4 Aug 2017

Dr. Tailleux states on p. C5 [doi:10.5194/acp-2017-17-RC1]: "If the equations considered were formulated in terms of the full barycentric velocity, I would agree that this term is non-zero, and physically related to the exchange of freshwater between the land/ocean and the atmosphere, whereby the atmosphere gains freshwater at a higher temperature than it returns it to the land/ocean in the form of precipitation."

Here we, first, clarify the meaning of this statement and, second, explain why it is incorrect. It is related to one of the main points of our article (see abstract): *... confusion between gaseous air velocity and mean velocity of air and condensate ... results in gross errors despite the observed magnitudes of these velocities are very close.*" The

caveat, which has apparently escaped the referee's attention despite our repeated emphasis, pertains to the definition of the material derivative: depending on the velocity used in this definition, one obtains drastically different results. The choice is not arbitrary: in the atmospheric context using barycentric velocity as suggested by the referee violates the first law of thermodynamics.

1. Consider air and condensate with velocities $\mathbf{v}$ and $\mathbf{v}_c$ and densities $\rho$ and $\rho_c$ obeying the following steady-state continuity equations:

$$\nabla \cdot (\rho \mathbf{v}) = \dot{\rho}, \ \ \nabla \cdot (\rho_c \mathbf{v}_c) = -\dot{\rho}, \ \ \nabla \cdot (\rho_m \mathbf{v}_m) = 0. \tag{1}$$

Here $\rho_m \equiv \rho + \rho_c$ and the mean velocity of air and condensate $\mathbf{v}_m \equiv (\mathbf{v}\rho + \mathbf{v}_c\rho_c)/\rho_m$ is the so-called barycentric velocity.

If we define the material derivative of enthalpy via barycentric velocity as $dh/dt_m \equiv (\mathbf{v}_m \cdot \nabla)h$, then, using the third equation in (1) and the divergence theorem, for the integral of $dh/dt_m$ over total atmospheric mass $\mathcal{M}$ we have

$$I_m \equiv \int_{\mathcal{M}} \frac{dh}{dt_m} d\mathcal{M} = \int_{\mathcal{V}} \frac{dh}{dt_m} \rho_m d\mathcal{V} = \int_{\mathcal{S}} h\rho_m \mathbf{v}_m \cdot \mathbf{n} d\mathcal{S}. \tag{2}$$

Interpreting $\rho_m \mathbf{v}_m$ in (2) as the mass flux of air and condensate across the planetary boundary and approximating the atmosphere as having a precipitating part where $\rho_m \mathbf{v}_m \cdot \mathbf{n} = P > 0$ and an evaporating part where $\rho_m \mathbf{v}_m \cdot \mathbf{n} = -E < 0$, one obtains that $I_m$ is proportional to the difference in enthalpy of air at the surface between the two regions, $I_m = Ph_P - Eh_E$, where $E = P$ (kg year$^{-1}$) is the evaporation/precipitation flux[1].
* * *
[1]This result, albeit mistakenly with an opposite sign and $h$ (enthalpy of moist air) replaced by the enthalpy of liquid water, was obtained by Dr. Tailleux during an earlier evaluation of our work, see www.bioticregulation.ru/offprint/he3-r2.pdf.

Note that if locally we have $\rho_m \mathbf{v}_m = 0$ (interpreted as local precipitation equals local evaporation), then $I_m = 0$. This runs counter to the reviewer's interpretation of $I_m$ being related to the difference in temperatures of precipitating and evaporating water.

In contrast to $I_m$, the enthalpy integral $I_h$ calculated in our article is not zero when local evaporation and precipitation coincide. Rather than being proportional to the difference in surface air enthalpies between regions of positive net evaporation and positive net precipitation, $I_h$ is proportional to the difference in enthalpies of moist air at the surface and at the mean height where condensation occurs (see Eq. (60) on page 21).

This discrepancy alone should already provoke some thought. Indeed, as the enthalpy change is constrained by the first law of thermodynamics, apparently only one expression, either $I_m$ or $I_h$, is correct.

2. The first law of thermodynamics relates mechanical work, heat increment and change in internal energy. Mechanical work in the atmosphere is expressed as $pd\tilde{V}$, where $p$ is ideal gas pressure and $d\tilde{V}$ is the macroscopic expansion/contraction of the considered air parcel (control volume). Therefore, as discussed in Section 2 of our article, work per unit volume per unit time, $(p/\tilde{V})(d\tilde{V}/dt)$, can be expressed as $p\nabla \cdot \mathbf{v}$, where $\mathbf{v}$ is *the velocity of gas which performs work*. The relative change of the control volume occupied by condensate particles, $\nabla \cdot \mathbf{v}_c$, is not related to production of work by ideal gas with pressure $p$, thus $p\nabla \cdot \mathbf{v}_m$ *is not production of work*.

Since Laliberté et al. (2015) apply the first law of thermodynamics to the atmosphere considered as a mixture of ideal gases, their expression for atmospheric power output – the mass integral of $-\alpha dp/dt$ – must define the material derivative of $p$ using the same velocity as in the expression for work, i.e. the velocity of ideal gas, $dp/dt \equiv (\mathbf{v}\cdot\nabla)p$. The other material derivatives in the first law of thermodynamics, including the material derivative of enthalpy $dh/dt \equiv (\mathbf{v}\cdot\nabla)h$, must be consistently defined using the same, i.e. gaseous, velocity. Using barycentric velocity for the material derivative of $h$ simultaneously with gaseous velocity for work violates the first law of thermodynamics.

(We note in passing that while the referee attempts to defend the neglect of $dh/dt$ by Laliberté et al. (2015), no derivation supporting this conclusion, i.e. that $\int(dh/dt)d\mathcal{M} = 0$, has been presented. The few arguments defending the neglect of $dh/dt$ appear somewhat confusing and controversial. In particular, Eq. (6) on page c6 of the referee's comment could indeed yield $\int(dh/dt)d\mathcal{M} = 0$, but if and only if $\mathbf{v} \cdot \mathbf{n} = 0$. The latter (correct) boundary condition is, however, what the referee objects to, so he apparently cannot have this derivation in mind. Furthermore, as we pointed out in our earlier reply, see page c5 of doi:10.5194/acp-2017-17-AC1, equations (6) and (7) on page c6 of the referee's comment are not the equations actually used by Laliberté et al. (2015) (who applied a velocity correction to Eq. (6)). Ultimately, the referee avoids to specify whether $v$ in his review is the gaseous or barycentric velocity.)

In the physics literature the proposition that the velocity associated with production of mechanical work is not necessarily identical to the mean flow velocity recently stimulated a rigorous discussion which is still on-going, see references in, and Google Scholar citations of, Brenner (2009). That the choice of an appropriate velocity crucially matters for the analysis of the atmospheric power budget is likewise a non-trivial issue, which apparently has never been discussed in the meteorological literature.

**References**

Brenner, H., 2009: Bi-velocity hydrodynamics. *Physica A Statistical Mechanics and its Applications*, **388**, 3391–3398, doi:10.1016/j.physa.2009.04.029.

Laliberté, F., J. Zika, L. Mudryk, P. J. Kushner, J. Kjellsson, and K. Döös, 2015: Constrained work output of the moist atmospheric heat engine in a warming climate. *Science*, **347**, 540–543, doi:10.1126/science.1257103.

---

## Author Comment (AC6) · 13 Aug 2017

Here we address concerns of Referee 2 about the need to consider two bodies for a consistent definition of work. The referee commented as follows: *"In physics, work is defined by force times length, with force being an interaction between TWO objects. Similarly, in thermodynamics, power implicitly involves TWO objects: an engine and an end-user that "dissipates the mechanical energy" and it is easy to define power through the forces acting between these TWO objects. The situation is obviously less clear when considering only ONE object. ... In any case, this is a choice. I strongly disagree with the author's point of view that "power" is a quantity defined a priori from the equations of the fluid. Power can only be defined as the mechanical energetic output of a "power system". Obviously, concerning the atmosphere, the "power system"*

[Figure]

*is only an abstract part of the atmosphere, that needs to be specified first. I believe this misunderstanding largely explains why the introduction and the definition of "power" is so confusing in this manuscript."*

There is a simple and well-studied example of work and power defined for one object – it is a spring. A compressed spring performs work as it expands. If another body is attached to the spring, the spring can perform work on that body. If no other bodies are attached, the spring performs work *on itself*. In this case the potential energy contained in the compressed spring is converted to the kinetic energy of the spring parts. In either case work performed by the spring is the same.

Likewise, when gas expands into vacuum it performs work *on itself* – its potential energy associated with pressure is converted to the kinetic energy of the macroscopic motion. If the expanding air parcel is surrounded by other air parcels, then a certain part of its work can go to compress and/or accelerate and/or warm these surrounding parcels. But, as with the compressed spring, the work itself remains the same – governed by gas pressure and the relative change of the parcel's volume.

We have added this explanation of the physical meaning of the obtained expression for work to Section 2.

For a quantitative insight, consider a compressed spring obeying Hooke's law. For such a spring the normal component of force $F = -ES\Delta l/l$ is proportional to spring extension $\Delta l < 0$, with $E$ being Young's modulus, $S$ the area of the spring's cross-section and $l$ is spring length. Since $\Delta l \equiv l - l_0$, where $l_0$ is length of uncompressed spring, we have $d(\Delta l)/dt = dl/dt$. As the spring begins to expand, work performed due to the changing spring's length (taken per unit time per unit spring volume $V = Sl$) is $W_s = (F/V)dl/dt = -E(\Delta l/l)(1/l)dl/dt$.

Length $l$ of the spring changes as the velocity difference $\Delta v = v_2 - v_1$ between the two ends of the spring, $dl/dt = \Delta v$. With $l \to 0$ we have $\Delta v/l \to \nabla \cdot \mathbf{v}$ and $W_s = -E(\Delta l/l)\nabla \cdot \mathbf{v}$, where $-E(\Delta l/l) = F/S = p$ is pressure (for details see, e.g., Feynman

et al., 1964, Chapter 38). Thus the resulting formula for $W_s$ is analogous to $W_a = p\nabla \cdot \mathbf{v}$ we obtained for work performed per unit time per unit volume by an expanding air parcel (Eq. (8) in Section 2).

Notably, in our formulation of work the referee's requirement that *"the power system is only an abstract part of the atmosphere that needs to be specified first"* is fulfilled – it is specified by choosing the scale at which an air parcel – the power system – is considered, i.e. the scale at which the divergence of velocity is defined. We have emphasized in our article that the definition of work is scale-specific (see paragraph below Eq. (9) in Section 2 and the fifth paragraph "The reason for this contradiction..." after Eq. (15) in Section 3.1) and quantified how its magnitude changes with the changing scale (see Section 5.3).

**References**

Feynman, R., R. Leighton, M. Sands, and E. Hafner, 1964: *The Feynman Lectures on Physics*, Vol. II. Addison-Wesley.

---

## Author Comment (AC7) · 14 Aug 2017

Some referees opined that our article appears lengthy and/or confusing.[1] We admit that, despite our continuing efforts to make it as clear as possible, our text is not easy reading. At the same time, we feel that we are not entirely to blame. Since a prerequisite to resolving confusions is to describe them, a confusing field with many subtleties cannot be presented in a brief and straightforward manner. When readers are unaware of the controversies surrounding the formulation of atmospheric power and are suddenly exposed to them while reading our article, they may attribute some of their resulting confusion to our presentation.

[Figure]
* * *
[1]http://dx.doi.org/10.5194/acp-2016-203-RC1 p. C1, http://dx.doi.org/10.5194/acp-2016-203-RC2 p. C4, http://dx.doi.org/10.5194/acp-2017-17-RC1 p. C1, http://dx.doi.org/10.5194/acp-2017-17-RC2 p. C2

Our interest in atmospheric power grew out of the idea that we can use its magnitude to test predictions of the condensation-induced dynamics – a theoretical approach we have been developing for the last few years. At first we had only a rough idea of how the atmospheric power could be formulated for a moist atmosphere. But, since it was nearly seventy years ago that Lorenz proclaimed quantifying atmospheric power as a major challenge for atmospheric physics, we had expected this information to be readily available from the literature. This turned out not to be the case. Below we present a list of controversies that challenge any researcher trying to do what we did – to derive a physically consistent and comprehensive formulation of atmospheric power and its budget in the presence of phase transitions.

1. A major issue is that, contrary to what one can read in textbooks (e.g., Vallis, 2006), in the presence of phase transitions the mass-specific rate of doing work (J s$^{-1}$ kg$^{-1}$) is not $pd\alpha/dt$. Mechanical work is proportional to the change of an air parcel's volume $\tilde{V}$ (m$^3$), while $\alpha \equiv 1/\rho$ (m$^3$ kg$^{-1}$) is the volume of a constant unit mass of gas. When mass of the air parcel is not constant, $d\alpha/dt \neq (d\tilde{V}/dt)/(\rho\tilde{V})$ (see Section 2 in our article). In simple words, if gas is contained in a box and a certain part of the gas condenses, but the volume of the box does not change, no work is performed. In this case $d\alpha > 0$ but $d\tilde{V} = 0$. To our knowledge, the correct formula for mass-specific rate of doing work (see Eq. (11) in Section 2) cannot be found in any textbooks. That this misconception is wide-spread was illustrated by Referee 2 and Referee 4 (Dr. Tailleux) of our first submission, who both used $pd\alpha/dt$ in their comments.[2]

2. Another major controversy, closely related to the first one, pertains to the correct choice of velocity in the definition of material derivatives of variables related to atmospheric power (the rate of doing work). Using either the velocity of gaseous air or the mean velocity of air and condensate produces drastically different estimates. As the discussion of our two submissions demonstrated[3], there is profound confusion as
* * *
[2]http://dx.doi.org/10.5194/acp-2016-203-RC2 p. C2, http://dx.doi.org/10.5194/acp-2016-203-RC3 p. C3
[3]http://dx.doi.org/10.5194/acp-2017-17-AC5

to which expressions for atmospheric power are valid. In the revised manuscript we summarize these caveats in a questionnaire, see Fig. 1 in the revised manuscript (also shown in the end of this comment).

In the presence of phase transitions for a correctly defined material derivative of, say, mass-specific enthalpy of moist air $h$ (J kg$^{-1}$) the following relationship *does not* hold: $d(\int_{\mathcal{M}} h d\mathcal{M})/dt = \int_{\mathcal{M}}(dh/dt)d\mathcal{M}$, where $\mathcal{M}$ is atmospheric mass[4]. The reason is, again, that the air parcel for which the material derivative is defined does not have a constant mass. Thus, for a given air parcel the change, per unit mass, of its enthalpy $\tilde{h} = h\tilde{m}$ (J) *is not equal* to the change of its mass-specific enthalpy: $d\tilde{h}/\tilde{m} \neq d(\tilde{h}/\tilde{m}) \equiv dh$ (see Eq. (57) in Section 4 of our article).

Again, to our knowledge, nowhere in the meteorological literature the importance of discriminating between gaseous air velocity and mean velocity of air and condensate has been highlighted. Elsewhere in physics such a recognition does exist (e.g., Brenner, 2009).

3. Likewise, nowhere in the meteorological literature can one find a discussion of the boundary conditions for velocity and how they matter for the atmospheric power budget (they do!). Hoping to gain some insight from the studies of Pauluis et al. (2000) and Pauluis and Held (2002), which are the only attempts to derive an expression for the atmospheric power budget for a moist atmosphere, one immediately notices that the boundary conditions adopted in their approach yield nonphysical results, with total atmospheric power exceeding solar power (see Section 3.1 and Fig. 3 in our article).

This inconsistency is due to the neglect of the scale-dependence of the definition of atmospheric power – another important subtlety that we emphasize (Sections 2 and 5.3). A non-zero vertical velocity for the evaporating water vapor exists at a microscopic scale of about one free path length from the evaporating surface. At a greater distance

[4]Cf. comment 4 on p. C2 of Referee 1 of our first submission, https://www.atmos-chem-phys-discuss.net/acp-2016-203/acp-2016-203-RC1.pdf

due to collisions with dry air molecules any appreciable difference in gas velocities cease to exist. To get a consistent account of macroscopic atmospheric power one has to put this velocity equal to zero at the evaporating surface and introduce surface sources (e.g., of moist air enthalpy) as Dirac's delta functions, such that the total flow of a considered property into the atmosphere is finite.

4. Furthermore, if one now turns to the numerous publications evaluating the Lorenz energy cycle – for a clue to understanding moist atmospheric power – there, too, one finds *two contrasting formulations yielding different results (!)*. These are the so-called $v \cdot grad\ z$ and $\omega \cdot \alpha$ formulations (Kim and Kim, 2013). While Peixoto and Oort explained that the two formulations must be identical, they are not. Most surprisingly, there appears to be little concern about why this is so.

In our revised text we explain that this contradiction is due to the long tradition of using mathematically inconsistent continuity equations, with a zero mass sink for air as a whole and a non-zero mass sink for the mixing ratio of water vapor. This approach, uncritically adopted by Pauluis et al. (2000) and Laliberté et al. (2015), introduces relatively minor errors in the evaluation of the rate of kinetic energy generation. But it profoundly undermines assessments of total atmospheric power and the gravitational power of precipitation, since both depend significantly on the intensity of the mass sink (see Section 3.5 in the revised text for details).

5. Next, even if one has made one's way through all these contradictions, formulated a valid expression for atmospheric power and is willing to explore the power *budget* involving the equations of motion and continuity, one is challenged with another controversy. One suddenly finds out that while many atmospheric models explicitly proclaim a zero source/sink in the continuity equation, they then implicitly re-introduce a non-zero sink via a technical velocity correction, of which many analysts, including Dr. Tailleux and Referee 3 of our first submission, appear to remain entirely unaware. Again, how this correction matters for the atmospheric power budget (it does!) appears previously unexplored.

6. Ultimately, turning to the equations of motion – i.e. to the atmospheric dynamics, which Referee 2 of our present submission has characterised as *"rather well studied and understood"* – one discovers that there co-exist two mutually inconsistent equations of motion for a moist atmosphere, of Ooyama (2001) and Bannon (2002), and that the seventeen years that passed since those formulations got published saw no attempts to resolve the controversy (but see Makarieva et al., 2017).

Thus, to be able to generate a reliable numerical estimate of atmospheric power for comparison with our theoretical predictions, we did our best to disentangle all these confusions, including the recent controversy involving the study of Laliberté et al. (2015). Our formulations simultaneously resolve all the six controversies in a self-consistent manner. Actually, we formulated an approach to handle atmospheric power in a moist atmosphere from scratch – to find that the obtained numbers are compatible with the predictions of the condensation-induced dynamics.

In the view of all the above, in our opinion, it would not be constructive to divide the paper in several parts as Referee 2 of our first submission and Referee 2 of our present submission would advise – we believe that there is an advantage in having a wholistic picture addressing all the existing inconsistencies in a single framework. In particular, cutting the quantitative MERRA – NCAR/NCEP part from the theoretical part would be unproductive. The comparison between MERRA (with its velocity correction to the continuity equations) and NCAR/NCEP (no velocity correction) is a direct illustration of how the form of the continuity equation matters for the atmospheric power estimate. In NCAR/NCEP, but not in MERRA, the total atmospheric power is negative! We would have hoped that some of our referees might be acquainted with these re-analyses well enough to comment on this result (Referee 2 from our first submission could perhaps do that but unfortunately he never showed up again).

**Acknowledgements.** We want to use this opportunity to thank our referees once again for their time and thought-provoking comments. We are particularly grateful to Dr. Tailleux (Referee 4 of the first submission and Referee 2 of the present submission)

who spent a lot of time evaluating several versions of our text. Dr. Tailleux' objections helped us better understand how the atmospheric and oceanic power budgets and dynamics differ. We are grateful to Referee 1 of our first submission who urged us to look at other datasets beyond MERRA: for us NCAR/NCEP with its negative power was an eye-opener. We are grateful to Referee 2 of our first submission, who attempted to verify our quantitative results and even attributed to us *"an original treatment of re-analyses data"*. Comments of Referee 3 of our first submission helped us a lot in clarifying how our work relates to that of Pauluis et al.

Referee 2 of our present submission paid attention to the key Section 2, which led to a more comprehensive understanding of the expression of global atmospheric power that we obtained.[5] Moreover, Referee 2 appeared sympathetic to our main idea – the key role of water vapor in driving atmospheric circulation. In this context, we want to briefly outline one more major controversy to which condensation-induced dynamics might provide a clue – monsoons.

7. People working with monsoons know that GCMs do not properly account for these abrupt seasonal shifts in the ocean-to-land atmospheric moisture transport (see, e.g., Acharya et al., 2011). To get around this problem, some time ago people proposed the so-called "moisture advection feedback" – a proposition that the atmosphere over land gets warmer and warmer as more and more latent heat is transported from the ocean. This warming, assumed to be proportional to the inflow of latent heat, is supposed to furter facilitate the ocean-to-land transport thus providing a positive feedback on the circulation intensity. This concept was invoked to explain modern monsoonal climates (e.g., Levermann et al., 2009), make projections for Amazonian deforestation (e.g., Wright et al., 2017; Boers et al., 2017) and explain monsoon shifts in the geological past (Herzschuh et al., 2014).

However, this mechanism does not appear to be valid, because, in simple words, la-

[5]http://dx.doi.org/10.5194/acp-2017-17-AC6

tent heat does not warm. That is to say, if the atmosphere is already moist adiabatic, a more rapid release of latent heat will not make it any warmer. Explained from a different angle, as we clarified in our response to Dr. Garrett, the release of latent heat per se cannot accelerate air circulation (and can even surpress it)[6]. A "moisture advection feedback" based on latent heat does not exist. This was recently clearly pointed out in an exchange of opinions in PNAS (Boos and Storelvmo, 2016a; Levermann et al., 2016; Boos and Storelvmo, 2016b). The abrupt monsoon transitions remain unexplained.

Condensation-induced dynamics, even if it is simplistically understood as a hydrostatic mass sink (but see p. C7 in http://dx.doi.org/10.5194/acp-2017-17-AC4), is principally different. Since the removal of gas by condensation lowers surface pressure, this pressure drop initiates an inflow of air towards the condensation area. If the arriving air ascends and is easily ventilated away from the condensation area in the upper atmosphere (i.e. when neither ascent nor ventilation is the limiting process of the circulation), then enhancing the inflow of water vapor will intensify the rate at which pressure is lowered by condensation thus providing a positive feedback to the circulation intensity. The intensity of condensation-induced air circulation is directly proportional to water vapor inflow and condensation rate (see Section 6 in our article) and thus provides the true "moisture advection feedback" necessary to explain monsoons. Forests moistening the atmosphere more efficiently than does the ocean, are able to win the "tug-of-war" with the ocean for atmospheric moisture (Makarieva et al., 2013, 2014).

Finally, we thank our Editor for making these discussions possible, for supervising them and for encouraging us to extend and re-submit our work after our first submission. We have learnt a lot while working on this topic.

**Technical note.** We will post the revised manuscript to arxiv at https://arxiv.org/abs/1603.03706 as version 4 (v4). All other relevant information
* * *
[6]http://dx.doi.org/10.5194/acp-2017-17-AC3 p. C3

will be available at www.bioticregulation.ru/ab.php?id=he. Interested readers might prefer to read the article at arxiv.org (the present submission is version 3) and not in ACPD. For some reason, the ACPD publishing process eliminates all the hyperlinks from the text (hyperlinks are supported by the ACPD Latex template and are present in the original PDF file that we submit to the journal). This makes navigation between the many equations extremely complicated (here we do understand Referee 2 who complained about the large number of equations in our article).

**References**

Acharya, N., S. C. Kar, U. C. Mohanty, M. A. Kulkarni, and S. K. Dash, 2011: Performance of GCMs for seasonal prediction over India–a case study for 2009 monsoon. *Theor. Appl. Climatol.*, **105**, 505–520.

Bannon, P. R., 2002: Theoretical foundations for models of moist convection. *J. Atmos. Sci.*, **59**, 1967–1982, doi:10.1175/1520-0469(2002)059<1967:TFFMOM>2.0.CO;2.

Boers, N., N. Marwan, H. M. J. Barbosa, and J. Kurths, 2017: A deforestation-induced tipping point for the South American monsoon system. *Scientific Reports*, **7 (41489)**, doi:doi:10.1038/srep41489.

Boos, W. R. and T. Storelvmo, 2016a: Near-linear response of mean monsoon strength to a broad range of radiative forcings. *Proc. Natl. Acad. Sci.*, **113 (6)**, 1510–1515, doi:10.1073/pnas.1517143113.

Boos, W. R. and T. Storelvmo, 2016b: Reply to Levermann et al.: Linear scaling for monsoons based on well-verified balance between adiabatic cooling and latent heat release. *Proc. Natl. Acad. Sci.*, **113 (17)**, E2350–E2351, doi:10.1073/pnas.1603626113.

Brenner, H., 2009: Bi-velocity hydrodynamics. *Physica A Statistical Mechanics and its Applications*, **388**, 3391–3398, doi:10.1016/j.physa.2009.04.029.

Herzschuh, U., J. Borkowski, J. Schewe, S. Mischke, and F. Tian, 2014: Moisture-advection feedback supports strong early-to-mid Holocene monsoon climate on the eastern Tibetan Plateau as inferred from a pollen-based reconstruction. *Palaeogeography, Palaeoclimatology, Palaeoecology*, **402**, 44 – 54, doi:https://doi.org/10.1016/j.palaeo.2014.02.022.

Kim, Y.-H. and M.-K. Kim, 2013: Examination of the global lorenz energy cycle using MERRA and NCEP-reanalysis 2. *Climate Dynamics*, **40**, 1499–1513, doi:10.1007/s00382-012-1358-4.

Laliberté, F., J. Zika, L. Mudryk, P. J. Kushner, J. Kjellsson, and K. Döös, 2015: Constrained work output of the moist atmospheric heat engine in a warming climate. *Science*, **347**, 540–543, doi:10.1126/science.1257103.

Levermann, A., V. Petoukhov, J. Schewe, and H. J. Schellnhuber, 2016: Abrupt monsoon transitions as seen in paleorecords can be explained by moisture-advection feedback. *Proc. Natl. Acad. Sci.*, **113 (17)**, E2348–E2349, doi:10.1073/pnas.1603130113.

Levermann, A., J. Schewe, V. Petoukhov, and H. Held, 2009: Basic mechanism for abrupt monsoon transitions. *Proc. Natl. Acad. Sci.*, **106 (49)**, 20 572–20 577.

Makarieva, A. M., V. G. Gorshkov, and B.-L. Li, 2013: Revisiting forest impact on atmospheric water vapor transport and precipitation. *Theor. Appl. Climatol.*, **111**, 79–96.

Makarieva, A. M., V. G. Gorshkov, A. V. Nefiodov, D. Sheil, A. D. Nobre, P. Bunyard, P. Nobre, and B.-L. Li, 2017: The equations of motion for moist atmospheric air. *Journal of Geophysical Research: Atmospheres*, doi:10.1002/2017JD026773, 2017JD026773.

Makarieva, A. M., V. G. Gorshkov, D. Sheil, A. D. Nobre, P. Bunyard, and B.-L. Li, 2014: Why does air passage over forest yield more rain? Examining the coupling between rainfall, pressure, and atmospheric moisture content. *J. Hydrometeor.*, **15**, 411–426, doi:10.1175/JHM-D-12-0190.1.

Ooyama, K. V., 2001: A dynamic and thermodynamic foundation for modeling the moist atmosphere with parameterized microphysics. *J. Atmos. Sci.*, **58**, 2073–2102, doi:10.1175/1520-0469(2001)058<2073:ADATFF>2.0.CO;2.

Pauluis, O., V. Balaji, and I. M. Held, 2000: Frictional dissipation in a precipitating atmosphere. *J. Atmos. Sci.*, **57**, 989–994, doi:10.1175/1520-0469(2000)057<0989:FDIAPA>2.0.CO;2.

Pauluis, O. and I. M. Held, 2002: Entropy budget of an atmosphere in radiative-convective equilibrium. Part I: Maximum work and frictional dissipation. *J. Atmos. Sci.*, **59**, 125–139, doi:10.1175/1520-0469(2002)059<0125:EBOAAI>2.0.CO;2.

Vallis, G. K., 2006: *Atmospheric and Oceanic Fluid Dynamics: Fundamentals and Large-Scale Circulation*. Cambridge University Press, 745 pp.

Wright, J. S., R. Fu, J. R. Worden, S. Chakraborty, N. E. Clinton, C. Risi, Y. Sun, and L. Yin, 2017: Rainforest-initiated wet season onset over the southern Amazon. *Proceedings of the National Academy of Sciences*, **114 (32)**, 8481–8486, doi:10.1073/pnas.1621516114.
* * *
[Figure]

**Figure 1.** A questionnaire overviewing possible issues with the formulation of atmospheric power. Here $p$ is the ideal gas pressure, $\rho$ and $\rho_c$ are the densitites of gaseous air and condensate particles, respectively; $\mathbf{v}$ is velocity of gaseous air, $\mathbf{v}_m \equiv (\rho\mathbf{v} + \rho_c\mathbf{v}_c)/(\rho + \rho_c)$ is the mean velocity of air and condensate (sometimes called "barycentric velocity"), $\mathbf{u}$ and $\mathbf{u}_m$ are the horizontal components of, respectively, $\mathbf{v}$ and $\mathbf{v}_m$, $\alpha \equiv 1/\rho$, $\mathcal{M}$ is the mass of the gaseous atmosphere $\mathcal{M} = \int \rho d\mathcal{V}$, $\mathcal{V}$ is the total atmospheric volume. "Moist atmosphere" implies $\dot{\rho} \neq 0$ and $\mathbf{v}_m \neq \mathbf{v}$, "dry atmosphere" implies $\dot{\rho} = 0$ and $\mathbf{v}_m = \mathbf{v}$. Note that in the physics literature it has been recognized that the choice between $\mathbf{v}$ and $\mathbf{v}_m$ is not trivial (Brenner, 2009). Our response to questions A-C is given in the beginning of Section 7.

**Fig. 1.** New figure added to revised Introduction.